# Spectral Neural Networks: Approximation Theory and Optimization Landscape

## Abstract

There is a large variety of machine learning methodologies that are based on the extraction of spectral geometric information from data. However, the implementations of many of these methods often depend on traditional eigensolvers, which present limitations when applied in practical online big data scenarios. To address some of these challenges, researchers have proposed different strategies for training neural networks as alternatives to traditional eigensolvers, with one such approach known as Spectral Neural Network (SNN). In this paper, we investigate key theoretical aspects of SNN. First, we present quantitative insights into the tradeoff between the number of neurons and the amount of spectral geometric information a neural network learns. Second, we initiate a theoretical exploration of the optimization landscape of SNN's objective to shed light on the training dynamics of SNN. Unlike typical studies of convergence to global solutions of NN training dynamics, SNN presents an additional complexity due to its non-convex ambient loss function.

## 1 Introduction

In the past decades, researchers from a variety of disciplines have studied the use of spectral geometric methods to process, analyze, and learn from data. These methods have been used in supervised learning (Ando & Zhang, 2006; Belkin et al., 2006; Smola & Kondor, 2003), clustering (Ng et al., 2001; Von Luxburg, 2007), dimensionality reduction (Belkin & Niyogi, 2001; Coifman et al., 2005), and contrastive learning (HaoChen et al., 2021). While the aforementioned methods have strong theoretical foundations, their algorithmic implementations often depend on traditional eigensolvers. These eigensolvers tend to underperform in practical big data scenarios due to high computational demands and memory constraints. Moreover, they are particularly vulnerable in online settings since the introduction of new data typically necessitates a full computation from scratch.

To overcome some of the drawbacks of traditional eigensolvers, new frameworks for learning from spectral geometric information that are based on the training of neural networks have emerged. A few examples are Eigensolver net (See in Appendix A.2), Spectralnet (Shaham et al., 2018), and Spectral Neural Network (SNN) (HaoChen et al., 2021). In the aforementioned approaches, the goal is to find neural networks that can approximate the spectrum of a large target matrix, and the differences among these approaches lie mostly in the specific loss functions used for training; here we focus on SNN, and provide some details on Eigensolver net and Spectralnet in Appendix A.2 for completeness. To explain the training process in SNN, consider a data set $\mathcal{X} = \{x_1, \ldots, x_n\}$ in $\mathbb{R}^d$ and a $n \times n$ adjacency matrix $\mathcal{A}_{\mathbf{n}}$ describing similarity among points in $\mathcal{X}_n$. A NN is trained by minimizing the *spectral constrastive* loss function:

$$\min_{\theta \in \Theta} \ L(\theta) \stackrel{\text{def}}{=} \ell(\mathbf{Y}_\theta), \quad \text{where} \quad \ell(\mathbf{Y}) \stackrel{\text{def}}{=} \left\| \mathbf{Y}\mathbf{Y}^\top - \mathcal{A}_{\mathbf{n}} \right\|_{\text{F}}^2, \quad \mathbf{Y} \in \mathbb{R}^{n \times r}, \tag{1.1}$$

through first-order optimization methods; see more details in Appendix A.1. In the above and in the sequel, $\theta$ represents the vector of parameters of the neural network $f_\theta : \mathbb{R}^d \to \mathbb{R}^r$, here a multi-layer ReLU neural network –see a detailed definition in Appendix C–, which can be interpreted as a feature or representation map for the input data; the matrix $\mathbf{Y}_\theta$ is the $n \times r$ matrix whose rows are the outputs $f_\theta(x_1), \ldots, f_\theta(x_n)$; $\|\cdot\|_{\text{F}}$ is the Frobenius norm.

Compared with plain eigensolver approaches, SNN has the following advantages:

1. **Training:** the spectral contrastive loss $\ell$ lends itself to minibatch training. Moreover, each iteration in the mini-batch training is cheap and only requires knowing the local structure of the adjacency matrix around a given point, making this approach suitable for online settings; see Appendix A.1 for more details.
2. **Memory:** when the number data points is large, storing an eigenvector of $\mathcal{A}_{\mathbf{n}}$ may be costly, while SNN can trade-off between accuracy and memory by selecting the dimension of the space of parameters of the neural network.
3. **Out-of-sample extensions:** A natural out-of-sample extension is built by simple evaluation of the trained neural network at an arbitrary input point.

Motivated by these algorithmic advantages, in this paper we investigate some of SNN's theoretical underpinnings. In concrete terms, we explore the following three questions:

> Q1 Are there theoretical guarantees that a neural network can approximate the eigenvectors of large adjacency matrices? How large does the neural network need to be to achieve a certain degree of approximation?
>
> Q2 Is it possible to use Equation 1.1 to build an approximating neural network?
>
> Q3 What can be said about the landscape of the objective function in 1.1?

**Contributions**  We provide answers to the above three questions in a specific setting to be described shortly. We also formulate and discuss open problems that, while motivated by our current investigation, we believe are of interest in their own right.

To make our setting more precise, through our discussion we adopt the *manifold hypothesis* and assume the data set $\mathcal{X} = \{x_1, \ldots, x_n\}$ to be supported on a low dimensional manifold $\mathcal{M}$ embedded in $\mathbb{R}^d$; see precise assumptions in Assumptions 2.1. We also assume that $\mathcal{X}$ is endowed with a similarity matrix $\boldsymbol{G}^\varepsilon$ with entries

$$\boldsymbol{G}_{ij}^\varepsilon = \eta\left(\frac{\|x_i - x_j\|}{\varepsilon}\right), \tag{1.2}$$

where $\|x - y\|$ denotes the Euclidean distance between $x$ and $y$, $\varepsilon$ is a proximity parameter, and $\eta$ is a decreasing, non-negative function. In short, $\boldsymbol{G}^\varepsilon$ measures the similarity between points according to their proximity. From $\boldsymbol{G}^\varepsilon$ we define the adjacency matrix $\mathcal{A}_{\mathbf{n}}$ appearing in Equation 1.1 by

$$\mathcal{A}_{\mathbf{n}} \stackrel{\text{def}}{=} \mathbf{D}_{\boldsymbol{G}}^{-\frac{1}{2}} \boldsymbol{G} \mathbf{D}_{\boldsymbol{G}}^{-\frac{1}{2}} + a\mathbf{I}, \tag{1.3}$$

where $\mathbf{D}_{\boldsymbol{G}}$ is the degree matrix associated to $\boldsymbol{G}$ as in Equation A.6 and $a > 1$ is a fixed quantity. Here we distance ourselves slightly from the choice made in the original SNN paper (HaoChen et al., 2021), where $\mathcal{A}_{\mathbf{n}}$ is taken to be $\boldsymbol{G}$ itself, and instead consider a normalized version. This is due to the following key properties satisfied by our choice of $\mathcal{A}_{\mathbf{n}}$ (see also Remark D.1 in Appendix D) that make it more suitable for theoretical analysis.

**Proposition 1.** *The matrix $\mathcal{A}_{\mathbf{n}}$ defined in Equation 1.1 satisfies the following properties:*

1. *$\mathcal{A}_{\mathbf{n}}$ is symmetric positive definite.*

2. *$\mathcal{A}_{\mathbf{n}}$'s $r$ top eigenvectors (the ones corresponding to the $r$ largest eigenvalues) coincide with the eigenvectors of the $r$ smallest eigenvalues of the symmetric normalized graph Laplacian matrix (see (Von Luxburg, 2007)):*

$$\Delta_n \stackrel{\text{def}}{=} \mathbf{I} - \mathbf{D}_{\boldsymbol{G}}^{-1/2} \boldsymbol{G} \mathbf{D}_{\boldsymbol{G}}^{-1/2}. \tag{1.4}$$

The above two properties, proved in Appendix D, are useful when combined with recent results on the regularity of graph Laplacian eigenvectors over proximity graphs (Calder et al., 2022) (see Appendix E.1) and some results on the approximation of Lipschitz functions on manifolds using neural networks (Chen et al., 2022) (see Appendix E.2). In particular, we answer question Q1, which belongs to the realm of approximation theory, by providing a concrete bound on the number of neurons in a multi-layer ReLU NN that are necessary to approximate the $r$ smallest eigenvectors of the normalized graph Laplacian matrix $\Delta_n$ (as defined in 1.4) and thus also the $r$ largest eigenvectors of $\mathcal{A}_{\mathbf{n}}$; this is the content of Theorem 2.1.

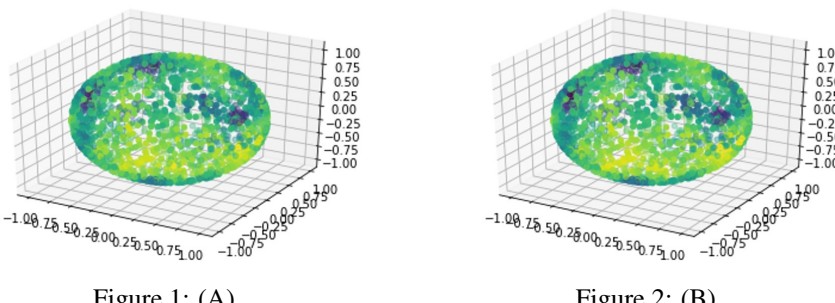

Figure 1: (A)  Figure 2: (B)

(B) shows the first eigenvector for the Laplacian of a proximity graph from data points sampled from $S^2$ obtained using an eigensolver. (A) shows the same eigenvector but obtained using SNN. The difference between the two figures is minor, showing that the neural network learns the eigenvector of the graph Laplacian well. See details in Appendix B.1.

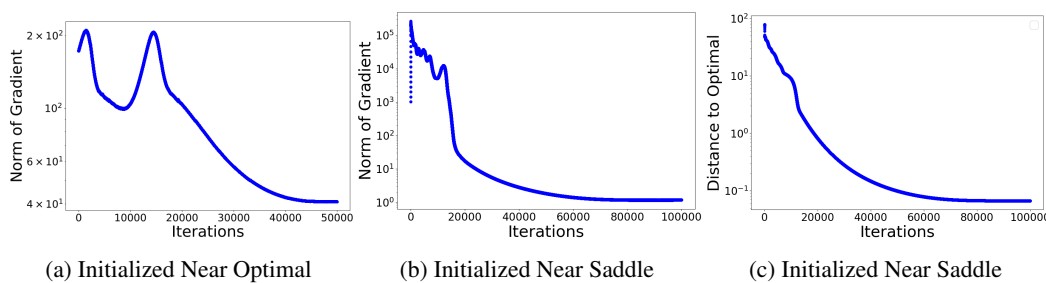

(a) Initialized Near Optimal  (b) Initialized Near Saddle  (c) Initialized Near Saddle

Figure 3: (a) and (b) Sum of the norms of the gradients for a two-layer ReLU Neural Network. In (a), the network is initialized near the global optimal solution and in (b) the network is initialized near a saddle point. (c) shows the distance between the current outputs of the neural network and the optimal solution for the case when it was initialized near a saddle point. More details are presented in Appendix B.2.

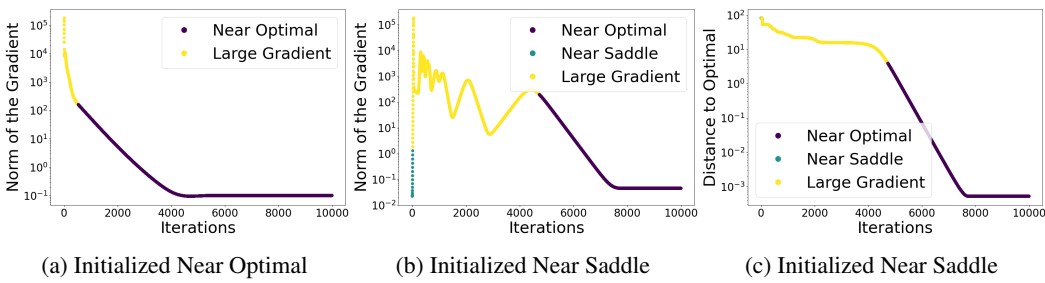

(a) Initialized Near Optimal  (b) Initialized Near Saddle  (c) Initialized Near Saddle

Figure 4: Norms of the gradients for the ambient problem and the distance to the optimal solution. In (a), $\mathbf{Y}$ is initialized near the global optimal solution, and in (b) $\mathbf{Y}$ is initialized near a saddle point. c) shows the distance between $\mathbf{Y}$ and the optimal solution for the case when it was initialized near a saddle point.

While our answer to question Q1 addresses the existence of a neural network approximating the spectrum of $\mathcal{A}_\mathbf{n}$, it does not provide a *constructive* way to find one such approximation. We thus address question Q2 and prove that an approximating NN can be constructed by solving the optimization problem 1.1, i.e., by finding a global minimizer of SNN's objective function. A precise statement can be found in Theorem 2.2. To prove this theorem, we rely on our estimates in Theorem 2.1 and on some auxiliary computations involving a global optimizer $\mathbf{Y}^*$ of the "ambient space problem":

$$\min_{\mathbf{Y} \in \mathbb{R}^{n \times r}} \ell(\mathbf{Y}). \tag{1.5}$$

For that we also make use of property 1 in Proposition 1, which allows us to guarantee, thanks to the Eckart–Young–Mirsky theorem (see (Eckart & Young, 1936) ), that solutions $\mathbf{Y}$ to Equation 1.5 coincide, up to multiplication on the right by a $r \times r$ orthogonal matrix, with a $n \times r$ matrix $\mathbf{Y}^*$ whose columns are scaled versions of the top $r$ normalized eigenvectors of the matrix $\mathcal{A}_\mathbf{n}$; see a detailed description of $\mathbf{Y}^*$ in Appendix D.2.

After discussing our spectral approximation results, we move on to discussing question Q3, which is related to the hardness of optimization problem 1.1. Notice that, while $\mathbf{Y}_{\theta^*}$ is a good approximator for $\mathcal{A}_\mathbf{n}$'s spectrum according to our theory, it is unclear whether $\theta^*$ can be reached through a standard training scheme. In fact, question Q3, as stated, is a challenging problem. This is not only due to the non-linearities in the neural network, but also because, in contrast to more standard theoretical studies of training dynamics of over-parameterized NNs (e.g., (Chizat & Bach, 2018; Wojtowytsch, 2020)), the spectral contrastive loss function $\ell$ is **non-convex** in the "ambient space" variable $\mathbf{Y}$. Specifically, $\ell(\mathbf{Y}) = \ell(\mathbf{Y}O)$ when $O$ is an orthogonal matrix. Despite this additional difficulty, numerical experiments —see Figure 3 for an illustration— suggest that first order optimization methods can find global solutions to Equation 1.1, and our goal here is to take a first step in the objective of understanding this behavior mathematically.

To begin, we present some numerical experiments where we consider different initializations for the training of SNN. Here we take 100 data points from MNIST and let $\mathcal{A}_n$ be the $n \times n$ gram matrix for the data points for simplicity. We remark that while we care about a $\mathcal{A}_n$ with a specific form for our approximation theory results, our analysis of the loss landscape described below holds for an arbitrary positive semi-definite matrix. In Figure 3, we plot the norm of the gradient during training when initialized in two different regions of parameter space. Concretely, in a region of parameters for which $\mathbf{Y}_\theta$ is close to a solution $\mathbf{Y}^*$ to problem 1.5 and a region of parameters for which $\mathbf{Y}_\theta$ is close to a saddle point of the ambient loss $\ell$. We compare these plots to the ones we produce from the gradient descent dynamics for the ambient problem 1.5, which are shown in Figure 4. We notice a similar qualitative behavior with the training dynamics of the NN, suggesting that the landscape of problem 1.1, if the NN is properly overparameterized, inherits properties of the landscape of $\ell$.

Motivated by the previous observation, in section 3 we provide a careful landscape analysis of the loss function $\ell$ introduced in Equation 1.1. We deem this landscape to be "benign", in the sense that it can be fully covered by the union of three regions described as follows: 1) the region of points close to global optimizers of Equation 1.5, where one can prove (Riemannian) strong convexity under a suitable quotient geometry; 2) the region of points close to saddle points, where one can find escape directions; and, finally, 3) the region where the gradient of $\ell$ is large. Points in these regions are illustrated in Figures 4b and 4c. The relevance of this global landscape characterization is that it implies convergence of most first-order optimization methods, or slight modifications thereof, toward global minimizers of the ambient space problem 1.5. This characterization is suggestive of analogous properties for the NN training problem in an overparameterized regime, but a full theoretical analysis of this is left as an open problem.

In summary, the main contributions of our work are the following:

- We show that we can approximate the eigenvectors of a large adjacency matrix with a NN, provided that the NN has sufficiently many neurons; see Theorem 2.1. Moreover, we show that by solving 1.1 one can *construct* such approximation provided the parameter space of the NN is rich enough; see Theorem 2.2.
- We provide precise error bounds for the approximation of eigenfunctions of a Laplace-Beltrami operator with NNs; see Corollary 1. In this way, we present an example of a setting where we can rigorously quantify the error of approximation of a solution to a PDE on a manifold with NNs.

- Motivated by numerical evidence, we begin an exploration of the optimization landscape of SNN and in particular provide a full description of SNN's associated ambient space optimization landscape. This landscape is shown to be benign; see discussion in Section 3.

## 1.1 RELATED WORK

**Spectral clustering and manifold learning**  Several works have attempted to establish precise mathematical connections between the spectra of graph Laplacian operators over proximity graphs and the spectrum of weighted Laplace-Beltrami operators over manifolds. Some examples include (Tao & Shi, 2020; Burago et al., 2014; García Trillos et al., 2020; Lu, 2022; Calder & García Trillos, 2022; Calder et al., 2022; Dunson et al., 2021; Wormell & Reich, 2021). In this paper we use adaptations of the results in (Calder et al., 2022) to infer that, with very high probability, the eigenvectors of the normalized graph Laplacian matrix $\Delta_n$ defined in Equation 1.4 are essentially Lipschitz continuous functions. These regularity estimates are one of the crucial tools for proving our Theorem 2.1.

**Contrastive Learning**  Contrastive learning is a self-supervised learning technique that has gained considerable attention in recent years due to its success in computer vision, natural language processing, and speech recognition (Chen et al., 2020a;b;c; He et al., 2020). Theoretical properties of contrastive representation learning were first studied by (Arora et al., 2019; Tosh et al., 2021; Lee et al., 2021) where they assumed conditional independence. (HaoChen et al., 2021) relaxes the conditional independence assumption by imposing the manifold assumption. With the spectral contrastive loss Equation 1.1 crucially in use, (HaoChen et al., 2021) provides an error bound for downstream tasks. In this work, we analyze how the neural network can approximate and optimize the spectral loss function Equation 1.1, which is the pertaining step of (HaoChen et al., 2021).

**Neural Network Approximations.**  Given a function $f$ with certain amount of regularity, many works have studied the tradeoff between width, depth, and total number of neurons needed and the approximation (Petersen, 2020; Lu et al., 2021). Specifically, (Shen et al., 2019) looks at the problem Holder continuous functions on the unit cube, (Yarotsky, 2018; Shen et al., 2020) for continuous functions on the unit cube, and (Petersen, 2020; Schmidt-Hieber, 2019; HaoChen et al., 2021) consider the case when the function is defined on a manifold. A related area is that of neural network memorization of a finite number of data points (Yun et al., 2019). In this paper, we use these results to show that for our specific type of regularity, we can prove similar results.

**Neural Networks and Partial Differential Equations**  (Raissi et al., 2019) introduced Physics Inspired Neural Networks as a method for solving PDEs using neural networks. Specifically, (Weinan & Yu, 2017; Bhatnagar et al., 2019; Raissi et al., 2019) use neural networks to parameterize the solution as use the PDE as the loss function. Other works such as (Guo et al., 2016; Zhu & Zabaras, 2018; Adler & Öktem, 2017; Bhatnagar et al., 2019) use neural networks to parameterize the solution operator on a given mesh on the domain. Finally, we have that eigenfunctions of operators on function spaces have a deep connection to PDEs. Recent works such as (Kovachki et al., 2021; Li et al., 2020a;b) demonstrate how to learn these operators. In this work we show that we can approximate eigenfunctions to a weighted Laplace-Beltrami operator using neural networks.

**Shallow Linear Networks and Non-convex Optimization in Linear Algebra Problems**  One of the main objects of study is the ambient problem 1.1. This formulation of the problem is related to linear networks. Linear networks are neural networks with identity activation. A variety of prior works have studied many different aspects of shallow linear networks such as the loss landscape and optimization dynamics (Baldi & Hornik, 1989; Tarmoun et al., 2021; Min et al., 2021; Bréchet et al., 2023), and generalization for one layer networks (Dobriban & Wager, 2018; Hastie et al., 2022; Bartlett et al., 2020; Kausik et al., 2023). Of relevance are also other works in the literature studying optimization problems very closely related to 1.5. For example, in Section 3 in (Li & Tang, 2017), there is a landscape analysis for 1.5 when the matrix $\mathcal{A}_\mathbf{n}$ is assumed to have rank smaller than or equal to $r$. That setting is typically referred to as overparameterized or exactly parameterized, whereas here our focus is on the underparameterized setting. On the other hand, the case studied in section 3 in (Chi et al., 2019) is the simplest case we could consider for our problem and corresponds to $r = 1$. In this simpler case, the non-convexity of the objective is completely due to a sign ambiguity, which makes the analysis more straightforward and the need to introduce quotient geometries less pressing. Luo & García Trillos (2022) describes the global optimization landscape of 1.5 under the assumption that $\mathcal{A}_\mathbf{n}$ is rank $r$; see a comparison in Remark 3.1.

## 2 SPECTRAL APPROXIMATION WITH NEURAL NETWORKS

Through this section we make the following assumption on the generation process of the data $\mathcal{X}_n$.

**Assumption 2.1.** *The points $x_1, \ldots, x_n$ are assumed to be sampled from a distribution supported on an $m$-dimensional manifold $\mathcal{M}$ that is assumed to be smooth, compact, orientable, connected, and without a boundary. We assume that this sampling distribution has a smooth density $\rho : \mathcal{M} \to \mathbb{R}_+$ with respect to $\mathcal{M}'s$ volume form, and assume that $\rho$ is bounded away from zero and also bounded above by a constant.*

### 2.1 SPECTRAL APPROXIMATION WITH MULTILAYER RELU NNS

**Theorem 2.1** (Spectral approximation of normalized Laplacians with neural networks). *Let $r \in \mathbb{N}$ be fixed. Under Assumptions 2.1, there are constants $c, C$ that depend on $\mathcal{M}, \rho$, and the embedding dimension $r$, such that, with probability at least*

$$1 - C\varepsilon^{-6m} \exp\left(-cn\varepsilon^{m+4}\right)$$

*for every $\delta \in (0, 1)$ there are $\kappa, L, p, N$ and a ReLU neural network $f_\theta \in \mathcal{F}(r, \kappa, L, p, N)$ (defined in Equation C.2), such that:*

1. *$\sqrt{n}\|\mathbf{Y}_\theta - \mathbf{Y}^*\|_{\infty,\infty} \leq C(\delta + \varepsilon^2)$, and thus also $\|\mathbf{Y}_\theta - \mathbf{Y}^*\|_{\mathrm{F}} \leq C\sqrt{r}(\delta + \varepsilon^2)$ .*
2. *The depth of the network, $L$, satisfies: $L \leq C\left(\log\frac{1}{\delta} + \log d\right)$, and its width, $p$, satisfies $p \leq C\left(\delta^{-m} + d\right)$.*
3. *The number of neurons of the network, $N$, satisfies: $N \leq Cr\left(\delta^{-m}\log\frac{1}{\delta} + d\log\frac{1}{\delta} + d\log d\right)$, and the range of weights, $\kappa$, satisfies $\kappa \leq \frac{C}{n^{1/(2L)}}$.*

Theorem 2.1 uses regularity properties of graph Laplacian eigenvectors and a NN approximation theory result for functions on manifolds. A summary of important auxiliary results needed to prove Theorem 2.1 is presented in Appendix E and the proof of the theorem itself is presented in Appendix F.

**Remark 2.1.** *We prove Theorem 2.1 by combining, in a non-trivial way, results from Chen et al. (2022) and Calder et al. (2022). More details are presented in the appendix. From our proof it follows that improvements in those works could be used to improve our estimates in Theorem 2.1.*

So far we have discussed approximations of the eigenvectors of $\mathcal{A}_{\mathbf{n}}$ (and thus also of $\Delta_n$) with neural networks, but more can be said about generalization of these NNs. In particular, the NN in our proof of Theorem 2.1 can be shown to approximate eigenfunctions of the weighted Laplace-Beltrami operator $\Delta_\rho$ defined in Appendix E.1. Precisely, we have the following result.

**Corollary 1.** *Under the same setting, notation, and assumptions as in Theorem 2.1, the neural network $f_\theta : \mathbb{R}^d \to \mathbb{R}^r$ can be chosen to satisfy*

$$\left\|\sqrt{\frac{n}{1+a}}f_\theta^i - f_i\right\|_{L^\infty(\mathcal{M})} \leq C(\delta + \varepsilon), \quad \forall i = 1, \ldots, r.$$

*In the above, $f_\theta^1, \ldots, f_\theta^r$ are the coordinate functions of the vector-valued neural network $f_\theta$, and the functions $f_1, \ldots, f_r$ are the normalized eigenfunctions of the Laplace-Beltrami operator $\Delta_\rho$ that are associated to $\Delta_\rho$'s $r$ smallest eigenvalues.*

**Remark 2.2.** *The $\varepsilon^2$ term that appears in the bound for $\|\mathbf{Y}_\theta - \mathbf{Y}^*\|_{\mathrm{F}}$ in Theorem 2.1 cannot be obtained simply from convergence of eigenvectors of $\Delta_n$ toward eigenfunctions of $\Delta_\rho$ in $L^\infty$. It turns out that we need to use a stronger notion of convergence (almost $C^{0,1}$) that in particular implies sharper regularity estimates for eigenvectors of $\Delta_n$ (see Corollary 2 in Appendix E.1 and Remark E.2 below it). In turn, the sharper $\varepsilon^2$ term is essential for our proof of Theorem 2.2 below to work; see the discussion starting in Remark E.2.*

### 2.2 SPECTRAL APPROXIMATION WITH GLOBAL MINIMIZERS OF SNN'S OBJECTIVE

After discussing the *existence* of approximating NNs, we turn our attention to *constructive* ways to approximate $\mathbf{Y}^*$ using neural networks. We give a precise answer to question Q2.

**Theorem 2.2** (Optimizing SNN approximates eigenvectors up to rotation). *Let $r \in \mathbb{N}$ be fixed and suppose that $\Delta_\rho$ is such that $\Delta_\rho$ has a spectral gap between its $r$ and $r+1$ smallest eigenvalues, i.e., in the notation in Appendix E.1, assume that $\lambda_r^{\mathcal{M}} < \lambda_{r+1}^{\mathcal{M}}$. For given $\kappa, L, p, N$ (to be chosen below), let $f_{\theta^*}$ be such that*

$$f_{\theta^*} \in \underset{f_\theta \in \mathcal{F}(r, \kappa, L, p, N)}{\arg \min} \|\mathbf{Y}_\theta \mathbf{Y}_\theta^\top - \mathcal{A}_\mathbf{n}\|_{\mathrm{F}}^2. \tag{2.1}$$

*Under Assumptions 2.1, there are constants $c, C$ that depend on $\mathcal{M}, \rho$, and the embedding dimension $r$, such that, with probability at least $1 - C\varepsilon^{-6m} \exp\left(-cn\varepsilon^{m+4}\right)$, for every $\tilde{\delta} \in (0, c)$ (i.e., $\tilde{\delta}$ sufficiently small) and for $\kappa = \frac{C}{n^{1/(2L)}}$ , $L = C\left(\log \frac{1}{\tilde{\delta}\varepsilon} + \log d\right)$, $p = C\left((\tilde{\delta}\varepsilon)^{-m} + d\right)$ and $N = \infty$, we have*

$$\min_{\mathbf{O} \in \mathbb{O}_r} \|\mathbf{Y}_{\theta^*} - \mathbf{Y}^* \mathbf{O}\|_{\mathrm{F}} \le C\varepsilon(\tilde{\delta} + \varepsilon). \tag{2.2}$$

**Remark 2.3.** *Equation 2.2 says that $\mathbf{Y}_{\theta^*}$ approximates a minimizer of the ambient problem 1.5 and that $\mathbf{Y}_{\theta^*}$ can be recovered but only up to rotation. This is unavoidable, since the loss function $\ell$ is invariant under multiplication on the right by a $r \times r$ orthogonal matrix. On the other hand, to set $N = \infty$ means we do not enforce sparsity constraints in the optimization of the NN parameters. This is convenient in practical settings and this is the reason why we state the theorem in this way. However, we can also set $N = r\left((\tilde{\delta}\varepsilon)^{-m} \log \frac{1}{\tilde{\delta}\varepsilon} + d \log \frac{1}{\tilde{\delta}\varepsilon} + d \log d\right)$ without affecting the conclusion of the theorem.*

## 3 LANDSCAPE OF SNN'S AMBIENT OPTIMIZATION PROBLEM

While in prior sections we considered a specific $\mathcal{A}_n$, the analysis in this section only relies on $\mathcal{A}_\mathbf{n}$ being positive definite with an eigengap between its $r$-th and $(r+1)$th top eigenvalues. We analyze the global optimization landscape of the non-convex Problem 1.5 under a suitable Riemannian *quotient geometry* (Absil et al., 2009; Boumal, 2023). The need for a quotient geometry comes from the fact that if $\mathbf{Y}$ is a stationary point of 1.5, then $\mathbf{YO}$ is also a stationary point for any $r \times r$ orthogonal matrix $\mathbf{O} \in \mathbb{O}_r$. This implies that the loss function $\ell$ is non-convex in any neighborhood of a stationary point (Li et al., 2019, Proposition 2). Despite the non-convexity of $\ell$, we show that under this geometry, Equation 1.5 is geodesically convex in a local neighborhood around the optimal solution.

Let $\overline{\mathcal{N}}_{r+}^n$ be the space of $n \times r$ matrices with full column rank. To define the quotient manifold, we encode the invariance mapping, i.e., $\mathbf{Y} \to \mathbf{YO}$, by defining the equivalence classes $[\mathbf{Y}] = \{\mathbf{YO} : \mathbf{O} \in \mathbb{O}_r\}$. From (Lee, 2018), we have $\mathcal{N}_{r+}^n \stackrel{\text{def}}{=} \overline{\mathcal{N}}_{r+}^n / \mathbb{O}_r$ is a quotient manifold of $\overline{\mathcal{N}}_{r+}^n$. See a detailed introduction to Riemannian optimization in (Boumal, 2023). Since the loss function in 1.5 is invariant along the equivalence classes of $\overline{\mathcal{N}}_{r+}^n$, $\ell$ induces the following optimization problem on the quotient manifold $\mathcal{N}_{r+}^n$:

$$\min_{[\mathbf{Y}] \in \mathcal{N}_{r+}^n} H([\mathbf{Y}]) \stackrel{\text{def}}{=} \frac{1}{2} \left\|\mathbf{YY}^\top - \mathcal{A}_\mathbf{n}\right\|_{\mathrm{F}}^2 \tag{3.1}$$

To analyze the landscape for Equation 3.1, we need expressions for the Riemannian gradient, the Riemannian Hessian, as well as the geodesic distance $d$ on this quotient manifold. By Lemma 2 from (Luo & García Trillos, 2022), we have that

$$d\left([\mathbf{Y}_1], [\mathbf{Y}_2]\right) = \min_{\mathbf{Q} \in \mathbb{O}_r} \|\mathbf{Y}_2 \mathbf{Q} - \mathbf{Y}_1\|_{\mathrm{F}}$$

and from Lemma 3 from (Luo & García Trillos, 2022), we have that

$$\overline{\text{grad } H([\mathbf{Y}])} = 2\left(\mathbf{YY}^\top - \mathcal{A}_\mathbf{n}\right)\mathbf{Y},$$
$$\overline{\text{Hess } H([\mathbf{Y}])}[\theta_\mathbf{Y}, \theta_\mathbf{Y}] = \left\|\mathbf{Y}\theta_\mathbf{Y}^\top + \theta_\mathbf{Y}\mathbf{Y}^\top\right\|_{\mathrm{F}}^2 + 2\left\langle\mathbf{YY}^\top - \mathcal{A}_\mathbf{n}, \theta_\mathbf{Y}\theta_\mathbf{Y}^\top\right\rangle. \tag{3.2}$$

Finally, by the classical theory on low-rank approximation (Eckart–Young–Mirsky theorem (Eckart & Young, 1936)), $[\mathbf{Y}^*]$ is the unique global minimizer of Equation 3.1. Let $\kappa^* = \sigma_1(\mathbf{Y}^*)/\sigma_r(\mathbf{Y}^*)$ be the condition number of $\mathbf{Y}^*$. Here, $\sigma_i(A)$ is the $i^{\text{th}}$ largest singular value of $A$, and $\|A\| = \sigma_1(A)$ is its spectral norm. Our precise assumption on the matrix $\mathcal{A}_n$ for this section is as follows.

**Assumption 3.1** (Eigengap). *$\sigma_{r+1}(\mathcal{A}_{\mathbf{n}})$ is strictly smaller than $\sigma_r(\mathcal{A}_{\mathbf{n}})$.*

See Remark H.1 for a discussion of the potential relaxation of the Eigengap assumption.

Let $\mu, \alpha, \beta, \gamma \geqslant 0$. We then split the landscape of $H([\mathbf{Y}])$ into the following five regions (not necessarily non-overlapping).

$$\mathcal{R}_1 \stackrel{\text{def}}{=} \left\{ \mathbf{Y} \in \mathbb{R}_*^{n \times r} \big| d\left([\mathbf{Y}], [\mathbf{Y}^*]\right) \leqslant \mu \sigma_r\left(\mathbf{Y}^*\right) / \kappa^* \right\},$$

$$\mathcal{R}_2 \stackrel{\text{def}}{=} \left\{ \mathbf{Y} \in \mathbb{R}_*^{n \times r} \left| \begin{array}{l} d\left([\mathbf{Y}], [\mathbf{Y}^*]\right) > \mu \sigma_r\left(\mathbf{Y}^*\right) / \kappa^*, \|\overline{\text{grad } H([\mathbf{Y}])}\|_{\mathrm{F}} \leqslant \alpha \mu \sigma_r^3\left(\mathbf{Y}^*\right) / (4\kappa^*), \\ \|\mathbf{Y}\| \leqslant \beta \|\mathbf{Y}^*\|, \left\|\mathbf{Y}\mathbf{Y}^\top\right\|_{\mathrm{F}} \leqslant \gamma \left\|\mathbf{Y}^*\mathbf{Y}^{*\top}\right\|_{\mathrm{F}} \end{array} \right. \right\},$$

$$\mathcal{R}_3' \stackrel{\text{def}}{=} \left\{ \mathbf{Y} \in \mathbb{R}_*^{n \times r} \left| \begin{array}{l} \|\overline{\text{grad } H([\mathbf{Y}])}\|_{\mathrm{F}} > \alpha \mu \sigma_r^3\left(\mathbf{Y}^*\right) / (4\kappa^*), \|\mathbf{Y}\| \leqslant \beta \|\mathbf{Y}^*\|, \\ \left\|\mathbf{Y}\mathbf{Y}^\top\right\|_{\mathrm{F}} \leqslant \gamma \left\|\mathbf{Y}^*\mathbf{Y}^{*\top}\right\|_{\mathrm{F}} \end{array} \right. \right\},$$

$$\mathcal{R}_3'' \stackrel{\text{def}}{=} \left\{ \mathbf{Y} \in \mathbb{R}_*^{n \times r} \big| \|\mathbf{Y}\| > \beta \|\mathbf{Y}^*\|, \|\mathbf{Y}\mathbf{Y}^\top\|_{\mathrm{F}} \leqslant \gamma \|\mathbf{Y}^*\mathbf{Y}^{*\top}\|_{\mathrm{F}} \right\},$$

$$\mathcal{R}_3''' \stackrel{\text{def}}{=} \left\{ \mathbf{Y} \in \mathbb{R}_*^{n \times r} \big| \left\|\mathbf{Y}\mathbf{Y}^\top\right\|_{\mathrm{F}} > \gamma \left\|\mathbf{Y}^*\mathbf{Y}^{*\top}\right\|_{\mathrm{F}} \right\},$$

$$\tag{3.3}$$

We show that for small values of $\mu$, the *loss function is geodesically convex* in $\mathcal{R}_1$. $\mathcal{R}_2$ is then defined as the region outside of $\mathcal{R}_1$ such that the Riemannian gradient is small relative to $\mu$. Hence this is the region in which we are close to the saddle points. We show that for this region there is *always an escape direction* (i.e., directions where the Hessian is strictly negative). $\mathcal{R}_3'$, $\mathcal{R}_3''$, and $\mathcal{R}_3'''$ are the remaining regions. We show that the *Riemannian gradient is large* (relative to $\mu$) in these regions. Finally, it is easy to see that $\mathcal{R}_1 \bigcup \mathcal{R}_2 \bigcup \mathcal{R}_3' \cup \mathcal{R}_3'' \bigcup \mathcal{R}_3''' = \mathbb{R}_*^{n \times r}$.

We are now ready to state the first of our main results from this section.

**Theorem 3.1** (Local Geodesic Strong Convexity and Smoothness of Equation 3.1). *Suppose $0 \leqslant \mu \leqslant \kappa^*/3$. Given that Assumption 3.1 holds, for any $\mathbf{Y} \in \mathcal{R}_1$ defined in Equation 3.3.*

$$\sigma_{\min}(\overline{\text{Hess } H([\mathbf{Y}])}) \geqslant \left(2\left(1 - \mu/\kappa^*\right)^2 - (14/3)\mu\right) \sigma_r\left(\mathcal{A}_{\mathbf{n}}\right) - 2\sigma_{r+1}(\mathcal{A}_{\mathbf{n}}),$$

$$\sigma_{\max}(\overline{\text{Hess } H([\mathbf{Y}])}) \leqslant 4\left(\sigma_1\left(\mathbf{Y}^*\right) + \mu\sigma_r\left(\mathbf{Y}^*\right)/\kappa^*\right)^2 + 14\mu\sigma_r^2\left(\mathbf{Y}^*\right)/3$$

*In particular, if $\mu$ is further chosen such that $\left(2\left(1 - \mu/\kappa^*\right)^2 - (14/3)\mu\right)\sigma_r\left(\mathcal{A}_{\mathbf{n}}\right) - 2\sigma_{r+1}(\mathcal{A}_{\mathbf{n}}) > 0$, we have $H([\mathbf{Y}])$ is geodesically strongly convex and smooth in $\mathcal{R}_1$.*

Theorem 3.1 guarantees that the optimization problem Equation 3.1 is geodesically strongly convex and smooth in a neighborhood of $[\mathbf{Y}^*]$. It also shows that if $\mathbf{Y}$ is close to the global minimizer, then Riemannian gradient descent converges to the global minimizer of the quotient space linearly.

Next, to analyze $\mathcal{R}_2$, we need to understand the other first-order stationary points (FOSP).

**Theorem 3.2** (FOSP of Equation 3.1). *Let $\overline{\mathbf{U}}\boldsymbol{\Sigma}\overline{\mathbf{U}}^\top$ be $\mathcal{A}_{\mathbf{n}}$'s SVD factorization, and let $\boldsymbol{\Lambda} = \boldsymbol{\Sigma}^{1/2}$. Then for any $S$ subset of $[n]$, we have that $\left[\overline{\mathbf{U}}_S\boldsymbol{\Lambda}_S\right]$ is a Riemannian FOSPs of Equation 3.1. Further, these are the only Riemannian FOSPs.*

Theorem 3.2 shows that the linear combinations of eigenvectors can be used to construct Riemannian first-order stationary points (FOSP) of Equation 3.1. This theorem also shows that there are many FOSPs of Equation 3.2. This is quite different from the regime studied in (Luo & García Trillos, 2022). In general, gradient descent is known to converge to a FOSP. Hence one might expect that if we initialized near one of the saddle points, then we might converge to that saddle point. However, our next main result of the section shows that even if we initialize near the saddle, there always exist escape directions.

**Theorem 3.3** (Escape Directions). *Assume that Assumption 3.1 holds. Then for sufficiently small $\alpha$ and any $\mathbf{Y} \in \mathcal{R}_2$ that is not an FOSP, there exists $C_1(\mathcal{A}_{\mathbf{n}}) > 0$ and $\theta_{\mathbf{Y}}$ such that*

$$\overline{\text{Hess } H([\mathbf{Y}])}\left[\theta_{\mathbf{Y}}, \theta_{\mathbf{Y}}\right] \leqslant -C_1\left(\mathcal{A}_{\mathbf{n}}\right)\|\theta_{\mathbf{Y}}\|_{\mathrm{F}}^2.$$

In particular, it is possible to exactly quantify the size of $\alpha$ and then explicitly construct the escape direction $\theta_{\mathbf{Y}}$. See Theorem H.1 in the appendix for more details.

Theorem 3.3 guarantees that, if $\mathbf{Y}$ is close to a saddle point, then $\theta_{\mathbf{Y}}$ will make its escape from the saddle point linearly.

Finally, the next result says that if we are not close to a FOSP, then we have large gradients.

**Theorem 3.4** ((Regions with Large Riemannian Gradient of Equation 1.5)**.**

1. $\left\|\overline{\operatorname{grad} H([\mathbf{Y}])}\right\|_{\mathrm{F}} > \alpha\mu\sigma_r^3\left(\mathbf{Y}^*\right)/\left(4\kappa^*\right), \forall \mathbf{Y} \in \mathcal{R}_3';$
2. $\left\|\overline{\operatorname{grad} H([\mathbf{Y}])}\right\|_{\mathrm{F}} \geqslant 2\left(\|\mathbf{Y}\|^3 - \|\mathbf{Y}\|\,\|\mathbf{Y}^*\|^2\right) > 2\left(\beta^3 - \beta\right)\|\mathbf{Y}^*\|^3, \quad \forall \mathbf{Y} \in \mathcal{R}_3'';$
3. $\left\langle \overline{\operatorname{grad} H([\mathbf{Y}])}, \mathbf{Y} \right\rangle > 2(1 - 1/\gamma)\left\|\mathbf{Y}\mathbf{Y}^\top\right\|_{\mathrm{F}}^2, \quad \forall \mathbf{Y} \in \mathcal{R}_3'''.$

*In particular, if $\beta > 1$ and $\gamma > 1$, we have the Riemannian gradient of $H([\mathbf{Y}])$ has large magnitude in all regions $\mathcal{R}_3', \mathcal{R}_3''$ and $\mathcal{R}_3'''$.*

The behavior, implied by our theorems, of gradient descent dynamics as it goes through the regions $\mathcal{R}_1, \mathcal{R}_2, \mathcal{R}_3$ is illustrated in Figures 3 and 4. See a discussion in Appendix B.2.

**Remark 3.1.** *These results can be seen as an under-parameterized generalization to the regression problem of Section 5 in (Luo & García Trillos, 2022). The proof in (Luo & García Trillos, 2022) is simpler because in their setting there are no saddle points or local minima that are not global in $\mathbb{R}_*^{n\times r}$. Conceptually, (Tarmoun et al., 2021) proves that in the setting $r \geq n$, the gradient flow for Equation 1.5 converges to a global minimum linearly. We complement this result by studying the case $r < n$.*

**Remark 3.2.** *In the specific case of $\mathcal{A}_{\mathbf{n}}$ as in Equation 1.3, and under Assumptions 2.1, Assumption 3.1 should be interpreted as $\lambda_r^{\mathcal{M}} < \lambda_{r+1}^{\mathcal{M}}$, as suggested by Remark E.1. Also, $\mu$ must be taken to be in the order $\varepsilon^2$. The scale $\varepsilon^2$ is actually a natural scale for this problem, since, as discussed in Remark G.3, the energy gap between saddle points and the global minimizer $[\mathbf{Y}^*]$ is $O(\varepsilon^2)$.*

## 4 Conclusions

We have explored some theoretical aspects of Spectral Neural Networks (SNN), a framework that substitutes the use of traditional eigensolvers with suitable neural network parameter optimization. Our emphasis has been on approximation theory, specifically identifying the minimum number of neurons of a multilayer NN required to capture spectral geometric properties in data, and investigating the optimization landscape of SNN, even in the face of its non-convex ambient loss function.

For our approximation theory results we have assumed a specific proximity graph structure over data points that are sampled from a distribution over a smooth low-dimensional manifold. A natural future direction worth of study is the generalization of these results to settings where data points, and their similarity graph, are sampled from other generative models, e.g., as in the application to contrastive learning in (HaoChen et al., 2021). To carry out this generalization, an important first step is to study the regularity properties of eigenvectors of an adjacency matrix/graph Laplacian generated from other types of probabilistic models.

At a high level, our approximation theory results have sought to bridge the extensive body of research on graph-based learning methods, their ties to PDE theory on manifolds, and the approximation theory for neural networks. While our analysis has focused on eigenvalue problems, such as those involving graph Laplacians or Laplace Beltrami operators, we anticipate that this overarching objective can be extended to develop new provably consistent methods for solving a larger class of PDEs on manifolds with neural networks, such as Schrödinger equation as in Hermann et al. (2020); Lu & Lu (2022). We believe this represents a significant and promising research avenue.

On the optimization front, we have focused on studying the landscape of the ambient space problem 1.5. This has been done anticipating the use of our estimates in a future analysis of the training dynamics of SNN. We reiterate that the setting of interest here is different from other settings in the literature that study the dynamics of neural network training in an appropriate scaling limit —leading to either a neural tangent kernel (NTK) or to a mean field limit. This difference is mainly due to the fact that the spectral contrastive loss $\ell$ (see 1.1) of SNN is non-convex, and even local strong convexity around a global minimizer does not hold in a standard sense and instead can only be guaranteed when considered under a suitable quotient geometry.

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

# A    TRAINING OF NEURAL NETWORKS FOR SPECTRAL APPROXIMATIONS

## A.1    TRAINING

Two of the main issues of standard eigensolvers are the need to store large matrices in memory and the need to redo computations from scratch if new data points are added. As mentioned, SNN can overcome this issue using mini-batch training. Specifically, the loss function $\ell(\mathbf{Y})$ can be written as,

$$\ell(\mathbf{Y}_\theta) = \sum_{i=1}^n \sum_{j=1}^n \left( (\mathcal{A}_\mathbf{n})_{ij} - (\mathbf{Y}_\theta \mathbf{Y}_\theta^\top)_{ij} \right) = \sum_{i=1}^n \sum_{j=1}^n \left( (\mathcal{A}_\mathbf{n})_{ij} - \left\langle f_\theta(x_i), f_\theta(x_j) \right\rangle \right)^2 \qquad \text{(A.1)}$$

where $(\mathcal{A}_\mathbf{n})_{ij}$ represents the $(i, j)$ entry of $\mathcal{A}_\mathbf{n}$ and $f_\theta$ is the neural network. Hence, in every iteration, one can randomly generate 1 index $(i, j)$ from $[n] \times [n]$, compute the loss and gradient for that term in the summation, and then perform one iteration of gradient descent.

## A.2    OTHER TRAINING APPROACHES

Besides SNN, there are two alternative ways of training spectral neural networks: *Eigensolver Net* and *SpectralNet* (Shaham et al., 2018). We compare these three different tools of neural network training and highlight the relative advantages and disadvantages of SNN.

**Eigensolver Net:** Given the matrix $\Delta_n$, one option could be to compute the eigendecomposition of $\Delta_n$ using traditional eigensolvers to get eigenvectors $\mathbf{v}_1, \ldots, \mathbf{v}_r$. Then, to learn an eigenfunction (that is, the function that maps data points to the corresponding entries of an eigenvector), we can minimize the following $\ell_2$ loss:

$$\min_\theta \|f_\theta(\mathcal{X}_n) - \mathbf{v}\|^2, \qquad \text{(A.2)}$$

where $\mathbf{v} = [\mathbf{v}_1, \mathbf{v}_2 \ldots, \mathbf{v}_r]$ and $\mathcal{X}_n$ is the data.

In general, the Eigensolver net is a natural way to extend to out-of-sample data and can be used to learn the eigenvector for matrices that are not PSD. On the other hand, the Eigensolver net has some drawbacks. Specifically, one still needs to compute the eigendecomposition using traditional eigensolvers.

**SpectralNet:** SpectralNet aims at minimizing the *SpectralNet loss*,

$$\mathcal{L}_{\text{SpectralNet}}(\theta) = \frac{1}{n^2} \sum_{i=1}^n \sum_{j=1}^n \eta \left( \frac{|x_i - x_j|}{\varepsilon} \right) \|f_\theta(x_i) - f_\theta(x_j)\|^2 \qquad \text{(A.3)}$$

where $f_\theta : \mathbb{R}^d \to \mathbb{R}^r$ encodes the spectral embedding of $x_i$ while satisfying the constraint

$$\mathbf{Y}_\theta^\top \mathbf{Y}_\theta = n\mathbf{I}_r, \qquad \text{(A.4)}$$

where $\mathbf{Y}_\theta = [f_\theta(x_1), \ldots, f_\theta(x_n)]$. This constraint is used to avoid a trivial solution. Note that Equation A.4 is a global constraint. (Shaham et al., 2018) have established a stochastic coordinate descent fashion to efficiently train SpectralNets. However, the stochastic training process in (Shaham et al., 2018) can only guarantee Equation A.4 holds approximately.

Conceptually, the SpectralNet loss Equation A.3 can also be written as

$$\mathcal{L}_{\text{SpectralNet}}(\theta) = \frac{2}{n^2} \text{trace} \left( \mathbf{Y}_\theta^\top \left( \mathbf{D}_{\boldsymbol{G}} - \boldsymbol{G} \right) \mathbf{Y}_\theta \right) \qquad \text{(A.5)}$$

where $\boldsymbol{G} \in \mathbb{R}^{n \times n}$ such that $\boldsymbol{G}_{ij} = \eta \left( \frac{\|x_i - x_j\|}{\varepsilon} \right)$, and $\mathbf{D}_{\boldsymbol{G}}$ is a diagonal matrix where

$$(\mathbf{D}_{\boldsymbol{G}})_{ii} = \sum_{j=1}^n \boldsymbol{G}_{ij}. \qquad \text{(A.6)}$$

The symmetric and positive semi-definite matrix $\mathbf{D}_{\boldsymbol{G}} - \boldsymbol{G}$ encodes the unnormalized graph Laplacian. Since $\mathbf{D}_{\boldsymbol{G}} - \boldsymbol{G}$ is positive semi-definite, the ambient problem of Equation A.5 is a constrained convex optimization problem. However, the parametrization and hard constraint A.4 make understanding SpectralNet's training process from a theoretical perspective challenging.

**Other NN-based Eigensolvers:** Other types of NN-based Eigensolvers have been considered in Pfau et al. (2019) and Deng et al. (2022).

Pfau et al. (2019) uses a bi-level optimization algorithm to solve a constrained optimization problem. This algorithm's computational complexity is typically higher than the one of SNN training and it requires keeping certain covariance matrices in memory during updates.

Deng et al. (2022) takes a similar approach as Pfau et al. (2019), but it can avoid the bi-level optimization in Pfau et al. (2019). This, however, comes at the expense of having an intractable theoretical computational complexity.

## B NUMERICAL DETAILS

### B.1 FOR EIGENVECTOR ILLUSTRATION

We sample 2000 data points $x_i$ uniformly from a 2-dimensional sphere embedded in $\mathbb{R}^3$, and then construct a 30 nearest neighbor graph among these points. Figure 1 shows a 1-hidden layer neural network evaluated at $x_i$, with 10000 hidden neurons to learn the first eigenvector of the graph Laplacian. The Network is trained for 5000 epochs using the full batch *Adam* in *Pytorch* and a learning rate of $2 * 10^{-5}$.

### B.2 AMBIENT VS PARAMETERIZED PROBLEM

We took 100 data points from MNIST. We normalized the pixel values to live in $[0, 1]$ and then computed $\mathcal{A}_n$ as the gran matrix.

The neural network has one hidden layer with a width of 1000. To initialize the neural network near a saddle point, we randomly pick a saddle point and then pretrain the network to approach this saddle. We used full batch gradient descent with an initial learning rate of 3e-6. We trained the network for 10000 iterations and used Cosine annealing as the learning rate scheduler.

After pretraining the network, we trained the network with the true objective. We used full batch gradient descent with an initial learning rate of 3e-6. We trained the network for 10000 iterations and used Cosine annealing as the learning rate scheduler.

When we initialized the network near the optimal solution, we followed the same procedure but pretrained the network for 1250 iterations.

For the ambient problem, we used full batch gradient descent with a learning rate 3e-6. We trained the network for 5000 iterations and again used Cosine annealing for the learning rate scheduler.

We remark that the sublinearity convergence rate in Figures 3 and 4 is due to the step size decaying in the optimizer. In $\mathcal{R}_1$, $H([\mathbf{Y}])$ has been shown to be strongly convex, so keeping the same step size should guarantee a linear rate. In this work, we don't focus on the optimization problem of SNN, but use this to illustrate Theorem 3.1, 3.3 and 3.4.

## C MULTI-LAYER RELU NEURAL NETWORKS

For concreteness, in this work we use multi-layer ReLU neural networks. To be precise, our neural networks are parameterized functions $f : \mathbb{R}^d \to \mathbb{R}^r$ of the form:

$$f(\mathbf{x}) = \mathbf{W}_L \cdot \text{ReLU}\left(\mathbf{W}_{L-1} \cdots \text{ReLU}\left(\mathbf{W}_1\mathbf{x} + \mathbf{b}_1\right) \cdots + \mathbf{b}_{L-1}\right) + \mathbf{b}_L, \quad \mathbf{x} \in \mathbb{R}^d. \quad \text{(C.1)}$$

More specifically, for a given choice of parameters $r, \kappa, L, p, N$ we will consider the family of functions:

$$\mathcal{F}(r, \kappa, L, p, N) = \left\{ f \mid f(\mathbf{x}) \text{ has the form } C.1, \text{ where:} \right.$$

$$\mathbf{W}_l \in \mathbb{R}^{p \times p}, \mathbf{b}_l \in \mathbb{R}^p \text{ for } l = 2, \dots, L - 1,$$
$$\mathbf{W}_1 \in \mathbb{R}^{p \times d}, \mathbf{b}_1 \in \mathbb{R}^p, \mathbf{W}_L \in \mathbb{R}^{r \times p}, \mathbf{b}_L \in \mathbb{R}^r. \tag{C.2}$$
$$\|\mathbf{W}_l\|_{\infty, \infty} \le \kappa, \|\mathbf{b}_l\|_{\infty} \le \kappa \text{ for } l = 1, \dots, L,$$
$$\left. \sum_{l=1}^{L} \|W_l\|_0 + \|\mathbf{b}_l\|_0 \le N \right\}$$

where $\| \cdot \|_0$ denotes the number of nonzero entries in a vector or a matrix, $\|\cdot\|_{\infty}$ denotes the $\ell_{\infty}$ norm of a vector. For a matrix $M$, we use $\|M\|_{\infty, \infty} = \max_{i,j} |M_{ij}|$.

For convenience, after specifying the quantities $r, \kappa, L, p, N$, we denote by $\Theta$ the space of admissible parameters $\theta = (\mathbf{W}_1, \mathbf{b}_1, \dots, \mathbf{W}_L, \mathbf{b}_L)$ in the function class $\mathcal{F}(r, \kappa, L, p, N)$, and we use $f_\theta$ to represent the function in Equation C.1.

## D   PROPERTIES OF THE MATRIX $\mathcal{A}_{\mathbf{n}}$ IN EQUATION 1.1

### D.1   PROOF OF PROPOSITION 1

*Proof of Proposition 1.*  Notice that

$$\mathcal{A}_{\mathbf{n}} = -\Delta_n + (a + 1)\mathbf{I}_n, \tag{D.1}$$

from where it follows that the eigenvectors of $\mathcal{A}_{\mathbf{n}}$ associated to its $r$ largest eigenvalues coincide with the eigenvectors of $\Delta_n$ associated to its $r$ smallest eigenvalues. Since $\mathcal{A}_{\mathbf{n}}$ is obviously symmetric, it remains to show that its eigenvalues are non-negative. In turn, from the definition of $\mathcal{A}_{\mathbf{n}}$ in Equation 1.3 and the fact that $a > 1$, it is sufficient to argue that all eigenvalues of $\mathbf{D}_G^{-1/2} \mathbf{G} \mathbf{D}_G^{-1/2}$ have absolute value less than or equal to 1. This, however, follows from the following two facts: 1) the matrix $\mathbf{D}_G^{-1/2} \mathbf{G} \mathbf{D}_G^{-1/2}$ is similar to the matrix $\mathbf{D}_G^{-1} \mathbf{G}$, given that

$$\mathbf{D}_G^{1/2} (\mathbf{D}_G^{-1} \mathbf{G}) \mathbf{D}_G^{-1/2} = \mathbf{D}_G^{-1/2} \mathbf{G} \mathbf{D}_G^{-1/2},$$

implying that $\mathbf{D}_G^{-1/2} \mathbf{G} \mathbf{D}_G^{-1/2}$ and $\mathbf{D}_G^{-1} \mathbf{G}$ have the same eigenvalues; and 2) all the eigenvalues of $\mathbf{D}_G^{-1} \mathbf{G}$ have norm less than one, since $\mathbf{D}_G^{-1} \mathbf{G}$ is a transition probability matrix. $\square$

**Remark D.1.** *While one could set $\mathcal{A}_{\mathbf{n}}$ to be $\Delta_n$ itself (since $\Delta_n$ is PSD), solving the resulting problem 1.5 would return the eigenvectors of $\Delta_n$ with the largest eigenvalues, which would not constitute a desirable output for data analysis, as the tail of the spectrum of $\Delta_n$ has little geometric information about the data set $\mathcal{X}_n$. It is interesting that we can still recover the relevant part of the spectrum of $\Delta_n$ indirectly, by studying the spectrum of the matrix $\mathcal{A}_{\mathbf{n}}$ that we use in this paper. Finally, it is worth mentioning that we add the term $a\mathbf{I}_n$ in the definition of $\mathcal{A}_{\mathbf{n}}$ in 1.3 to guarantee that $\mathcal{A}_{\mathbf{n}}$ is always PSD, in this way simplifying the statements and proofs of our main results.*

### D.2   FORM OF $\mathbf{Y}^*$ AND SOME NOTATION

Since $\mathcal{A}_{\mathbf{n}}$ is a PSD matrix, the Eckart–Young–Mirsky theorem (see (Eckart & Young, 1936)) implies that the global optimizers of 1.5 are the matrices $\mathbf{Y}$ of the form $\mathbf{Y} = \mathbf{Y}^* O$, where $O \in \mathbb{O}_r$ and

$$\mathbf{Y}^* \stackrel{\text{def}}{=} \begin{bmatrix} | & & | \\ \sqrt{\sigma_1(\mathcal{A}_{\mathbf{n}})} v_1 & \dots & \sqrt{\sigma_r(\mathcal{A}_{\mathbf{n}})} v_r \\ | & & | \end{bmatrix}.$$

In the above, $\sigma_l(\mathcal{A}_{\mathbf{n}})$ represents the $l$-th largest eigenvalue of $\mathcal{A}_{\mathbf{n}}$ and $v_l$ is a corresponding eigenvector with Euclidean norm one. In case there are repeated eigenvalues, the corresponding $v_l$ need to be chosen as being orthogonal to each other.

For convenience, we rescale the vectors $v_l$ as follows:

$$u_l \stackrel{\text{def}}{=} \sqrt{n} v_l.$$

In this way we guarantee that

$$\|u_l\|_{L^2(\mathcal{X}_n)}^2 \overset{\text{def}}{=} \frac{1}{n} \sum_{i=1}^{n} (u_l(x_i))^2 = 1,$$

i.e., the rescaled eigenvectors $u_l$ are normalized in the $L^2$-norm with respect to the empirical measure $\frac{1}{n} \sum_{i=1}^{n} \delta_{x_i}$. In terms of the rescaled eigenvectors $u_l$, we can rewrite $\mathbf{Y}^*$ as follows:

$$\mathbf{Y}^* = \begin{bmatrix} | & & | \\ \sqrt{\frac{\sigma_1(\mathcal{A}_\mathbf{n})}{n}} u_1 & \cdots & \sqrt{\frac{\sigma_r(\mathcal{A}_\mathbf{n})}{n}} u_r \\ | & & | \end{bmatrix}. \tag{D.2}$$

**Remark D.2.** *As discussed in Remark E.1 below, under Assumptions 2.1 we can assume that all the $\sigma_s(\mathcal{A}_\mathbf{n})$ are quantities of order one.*

# E   AUXILIARY APPROXIMATION RESULTS

## E.1   GRAPH-BASED SPECTRAL APPROXIMATION OF WEIGHTED LAPLACE-BELTRAMI OPERATORS

In this section, we discuss two important results characterizing the behavior of the spectrum of the normalized graph Laplacian matrix $\Delta_n$ defined in Equation 2 when $n$ is large and $\varepsilon$ scales with $n$ appropriately. In particular, $\Delta_n$'s spectrum is seen to be closely connected to that of the weighted Laplace-Beltrami operator $\Delta_\rho$ defined as

$$\Delta_\rho f \overset{\text{def}}{=} -\frac{1}{\rho^{3/2}} \operatorname{div} \left( \rho^2 \nabla \left( \frac{f}{\sqrt{\rho}} \right) \right)$$

for all smooth enough $f : \mathcal{M} \to \mathbb{R}$; see section 1.4 in (García Trillos & Slepčev, 2018). In the above, div stands for the divergence operator on $\mathcal{M}$, and $\nabla$ for the gradient in $\mathcal{M}$. $\Delta_\rho$ can be easily seen to be a positive semi-definite operator with respect to the $L^2(\mathcal{M}, \rho)$ inner product and its eigenvalues (repeated according to multiplicity) can be listed in increasing order as

$$0 = \lambda_1^{\mathcal{M}} \leq \lambda_2^{\mathcal{M}} \leq \dots$$

We will use $f_1, f_2, \dots$ to denote associated normalized (in the $L^2(\mathcal{M}, \rho)$-sense) eigenfuntions of $\Delta_\rho$.

The first result, whose proof we omit as it is a straightforward adaptation of the proof of Theorem 2.4 in (Calder & García Trillos, 2022) –which considers the *unnormalized* graph Laplacian case–, relates the eigenvalues of $\Delta_n$ and $\Delta_\rho$.

**Theorem E.1** (Convergence of eigenvalues of graph Laplacian; Adapted from Theorem 2.4 in (Calder & García Trillos, 2022))**.** *Let $l \in \mathbb{N}$ be fixed. Under Assumptions 2.1, with probability at least $1 - Cn \exp\left(-cn\varepsilon^{m+4}\right)$ over the sampling of the $x_i$, we have:*

$$\left| \sigma_\eta \lambda_s^{\mathcal{M}} - \frac{\hat{\lambda}_s}{\varepsilon^2} \right| \leq C_r \varepsilon, \quad \forall s = 1, \dots, l.$$

*In the above, $\hat{\lambda}_1 \leq \dots \leq \hat{\lambda}_l$ are the first eigenvalues of $\Delta_n$ in increasing order, $C_l$ is a deterministic constant that depends on $\mathcal{M}$'s geometry and on $l$, and $\sigma_\eta$ is a constant that depends on the kernel $\eta$ determining the graph weights (see Equation 1.2). We also recall that $m$ denotes the intrinsic dimension of the manifold $\mathcal{M}$.*

**Remark E.1.** *From Theorem E.1 and Equation D.1 we see that the top $l$ eigenvalues of $\mathcal{A}_\mathbf{n}$ (for $l$ fixed), i.e., $\sigma_1(\mathcal{A}_\mathbf{n}), \dots, \sigma_l(\mathcal{A}_\mathbf{n})$, can be written as*

$$\sigma_s(\mathcal{A}_\mathbf{n}) = 1 + a - \sigma_\eta \sigma_s^{\mathcal{M}} \varepsilon^2 + O(\varepsilon^3)$$

*with very high probability.*

*In particular, although each individual $\sigma_s(\mathcal{A}_\mathbf{n})$ is an order one quantity, the difference between any two of them is an order $\varepsilon^2$ quantity.*

Next we discuss the convergence of eigenvectors of $\Delta_n$ toward eigenfunctions of $\Delta_\rho$. For the purposes of this paper (see some discussion below) we follow a strong, almost $C^{0,1}$ convergence result established in (Calder et al., 2022) for the case of unnormalized graph Laplacians. A straightforward adaptation of Theorem 2.6 in (Calder et al., 2022) implies the following.

**Theorem E.2** (Almost $C^{0,1}$ convergence of graph Laplacian eigenvectors; Adapted from Theorem 2.6 in (Calder et al., 2022)). *Let $r \in \mathbb{N}$ be fixed and let $u_1, \ldots, u_r$ be normalized eigenvectors of $\Delta_n$ as in Appendix D.2. Under Assumptions 2.1, with probability at least $1 - C\varepsilon^{-6m} \exp\left(-cn\varepsilon^{m+4}\right)$ over the sampling of the $x_i$, we have:*

$$\|f_s - u_s\|_{L^\infty(\mathcal{X}_n)} + [f_s - u_s]_{\varepsilon,\mathcal{X}_n} \leq C_r\varepsilon. \quad \forall s = 1, \ldots, r, \tag{E.1}$$

*for normalized eigenfunctions $f_i : \mathcal{M} \to \mathbb{R}$ of $\Delta_\rho$, as introduced at the beginning of this section. In the above, $\|\cdot\|_{L^\infty(\mathcal{X}_n)}$ is the norm $\|v\|_{L^\infty(\mathcal{X}_n)} \overset{def}{=} \max_{x_i \in \mathcal{X}_n} |v(x_i)|$, and $[\cdot]_{\varepsilon,\mathcal{X}_n}$ is the seminorm*

$$[v]_{\varepsilon,\mathcal{X}_n} \overset{def}{=} \max_{x_i,x_j \in \mathcal{X}_n} \frac{|v(x_i) - v(x_j)|}{d_\mathcal{M}(x_i, x_j) + \varepsilon}.$$

$d_\mathcal{M}(\cdot, \cdot)$ *denotes the geodesic distance on $\mathcal{M}$.*

An essential corollary of the above theorem is the following set of regularity estimates satisfied by eigenvectors of the normalized graph Laplacian $\Delta_n$.

**Corollary 2.** *Under the same setting, notation, and assumptions as in Theorem E.2, the functions $u_s$ satisfy*

$$|u_s(x_i) - u_s(x_j)| \leq L_s(d_\mathcal{M}(x_i, x_j) + \varepsilon^2), \quad \forall x_i, x_j \in \mathcal{X}_n \tag{E.2}$$

*for some deterministic constant $L_s$.*

*Proof.* From Equation E.1 we have

$$|(u_s(x_i) - f_s(x_i)) - (u_s(x_j) - f_s(x_j))| \leq C_s\varepsilon(d_\mathcal{M}(x_i, x_j) + \varepsilon), \quad \forall x_i, x_j \in \mathcal{X}_n.$$

It follows from the triangle inequality that

$$\begin{aligned}
|u_s(x_i) - u_s(x_j)| &\leq |u_s(x_i) - f_s(x_i) - (u_s(x_j) - f_s(x_j))| + |f_s(x_i) - f_s(x_j)| \\
&\leq C_s\varepsilon(d_\mathcal{M}(x_i, x_j) + \varepsilon) + C'_s d_\mathcal{M}(x_i, x_j) \\
&\leq L_s(d_\mathcal{M}(x_i, x_j) + \varepsilon^2).
\end{aligned}$$

In the above, the second inequality follows from inequality E.1 and the fact that $f_s$, being a normalized eigenfunction of the elliptic operator $\Delta_\rho$, is Lipschitz continuous with some Lipschitz constant $C'_s$.

$\square$

**Remark E.2.** *We observe that the $\varepsilon^2$ term on the right hand side of Equation E.2 is strictly better than the $\varepsilon$ term that appears in the explicit regularity estimates in Remark 2.4 in (Calder et al., 2022). It turns out that in the proof of Theorem 2.2 it is essential to have a correction term for the distance that is $o(\varepsilon)$; see more details in Remark G.1 below.*

### E.2 NEURAL NETWORK APPROXIMATION OF LIPSCHITZ FUNCTIONS ON MANIFOLDS

(Chen et al., 2022) shows that Lipschitz functions $f$ defined over an $m$-dimensional smooth manifold $\mathcal{M}$ embedded in $\mathbb{R}^d$ can be approximated with a ReLU neural network with a number of neurons that doesn't grow exponentially with the ambient space dimension $d$. Precisely:

**Theorem E.3** (Theorem 1 in (Chen et al., 2022)). *Let $f : \mathcal{M} \to \mathbb{R}$ be a Lipschitz function with Lipschitz constant less than $K$. Given any $\delta \in (0, 1)$, there are $\kappa, L, p, N$ satisfying:*

1. *$L \leq C_K \left(\log \frac{1}{\delta} + \log d\right)$, and $p \leq C_K \left(\delta^{-m} + d\right)$,*

2. *$N \leq C_K \left(\delta^{-m} \log \frac{1}{\delta} + d \log \frac{1}{\delta} + d \log d\right)$, and $\kappa \leq C_K$,*

*such that there is a neural network $f_\theta \in \mathcal{F}(1, \kappa, L, p, N)$ (as defined in Equation C.2), for which*

$$\|f_\theta - f\|_{L^\infty(\mathcal{M})} \leq \delta.$$

*In the above, $C_K$ is a constant that depends on $K$ and on the geometry of the manifold $\mathcal{M}$.*

In this paper we utilize the results from (Chen et al., 2022) due to the fact that in their estimates the ambient space dimension $d$ does not appear in any exponent.

## F  PROOFS OF THEOREM 2.1 AND COROLLARY 1

**Lemma F.1.** *Let $u : \mathcal{X}_n \to \mathbb{R}$ be a function satisfying*

$$|u(x) - u(\tilde{x})| \leq L(d_\mathcal{M}(x, \tilde{x}) + \varepsilon^2), \quad \forall x, \tilde{x} \in \mathcal{X}_n \tag{F.1}$$

*for some $L$ and $\varepsilon > 0$. Then there exists a $3L$-Lipschitz function $\tilde{g} : \mathcal{M} \to \mathbb{R}$ such that*

$$\|u - \tilde{g}\|_{L^\infty(\mathcal{X}_n)} \leq 5L\varepsilon^2. \tag{F.2}$$

*Proof.* We start by constructing a subset $\mathcal{X}'_n$ of $\mathcal{X}_n$ satisfying the following properties:

1. Any two points $x, \tilde{x} \in \mathcal{X}'_n$ (different from each other) satisfy $d_\mathcal{M}(x, \tilde{x}) \geq \frac{1}{2}\varepsilon^2$.

2. For any $x \in \mathcal{X}_n$ there exists $\tilde{x} \in \mathcal{X}'_n$ such that $d_\mathcal{M}(x, \tilde{x}) \leq \varepsilon^2$.

The set $\mathcal{X}'_n$ can be constructed inductively, as we explain next. First, we enumerate the points in $\mathcal{X}_n$ as $x_1, \ldots, x_n$. After having decided whether to include or not in $\mathcal{X}'_n$ the first $s$ points in the list, we decide to include $x_{s+1}$ as follows: if the ball of radius $\varepsilon^2/2$ centered at $x_{s+1}$ intersects any of the balls of radius $\varepsilon^2/2$ centered around the points already included in $\mathcal{X}'_n$, then we do not include $x_{s+1}$ in $\mathcal{X}'_n$, otherwise we include it. It is clear from this construction that the resulting set $\mathcal{X}'_n$ satisfies the desired properties (property 2 follows from the triangle inequality).

Now, notice that the function $u : \mathcal{X}'_n \to \mathbb{R}$ (i.e., $u$ restricted to $\mathcal{X}'_n$) is $3L$-Lipschitz, since

$$|u(x) - u(\tilde{x})| \leq L(d_\mathcal{M}(x, \tilde{x}) + \varepsilon^2) \leq 3Ld_\mathcal{M}(x, \tilde{x})$$

for any pair of points $x, \tilde{x}$ in $\mathcal{X}'_n$. Using McShane-Whitney theorem we can extend the function $u : \mathcal{X}'_n \to \mathbb{R}$ to a $3L$-Lipschitz function $\tilde{g} : \mathcal{M} \to \mathbb{R}$. It remains to prove Equation F.2. To see this, let $x \in \mathcal{X}_n$ and let $\tilde{x} \in \mathcal{X}_n$ be as in property 2 of $\mathcal{X}'_n$. Then

$$\begin{aligned}
|u(x) - \tilde{g}(x)| &\leq |u(x) - u(\tilde{x})| + |u(\tilde{x}) - g(x)| \\
&= |u(x) - u(\tilde{x})| + |g(\tilde{x}) - g(x)| \\
&\leq L(d_\mathcal{M}(x, \tilde{x}) + \varepsilon^2) + 3Ld_\mathcal{M}(x, \tilde{x}) \\
&\leq 5L\varepsilon^2.
\end{aligned}$$

This completes the proof. $\qquad\square$

**Remark F.1.** *Lemma F.1 guarantees that if a function $u$, defined in any given metric space, is $(L, \varepsilon^2)$-almost Lipschitz, then we can find a function $\tilde{g}$ that is $L$-Lipschitz continuous in the same space and is within uniform distance $\varepsilon^2$ from $u$.*

We are ready to prove Theorem 2.1, which here we restate for convenience.

**Theorem 2.1** (Spectral approximation of normalized Laplacians with neural networks)**.** *Let $r \in \mathbb{N}$ be fixed. Under Assumptions 2.1, there are constants $c, C$ that depend on $\mathcal{M}, \rho$, and the embedding dimension $r$, such that, with probability at least*

$$1 - C\varepsilon^{-6m} \exp\left(-cn\varepsilon^{m+4}\right)$$

*for every $\delta \in (0, 1)$ there are $\kappa, L, p, N$ and a ReLU neural network $f_\theta \in \mathcal{F}(r, \kappa, L, p, N)$ (defined in Equation C.2), such that:*

1. $\sqrt{n}\|\mathbf{Y}_\theta - \mathbf{Y}^*\|_{\infty,\infty} \leq C(\delta + \varepsilon^2)$, *and thus also* $\|\mathbf{Y}_\theta - \mathbf{Y}^*\|_{\mathrm{F}} \leq C\sqrt{r}(\delta + \varepsilon^2)$.
2. *The depth of the network, $L$, satisfies:* $L \leq C\left(\log\frac{1}{\delta} + \log d\right)$, *and its width, $p$, satisfies* $p \leq C\left(\delta^{-m} + d\right)$.
3. *The number of neurons of the network, $N$, satisfies:* $N \leq Cr\left(\delta^{-m}\log\frac{1}{\delta} + d\log\frac{1}{\delta} + d\log d\right)$, *and the range of weights, $\kappa$, satisfies* $\kappa \leq \frac{C}{n^{1/(2L)}}$.

*Proof.* Let $s \leq r$. As in the discussion of section D.2 we let $u_s$ be a $\|\cdot\|_{L^2(\mathcal{X}_n)}$-normalized eigenvector of $\Delta_n$ corresponding to its $s$-th smallest eigenvalue. Thanks to Corollary 2, we know that, with very high probability, the function $u_s : \mathcal{X}_n \to \mathbb{R}$ satisfies

$$|u_s(x_i) - u_s(x_j)| \leq L_s(d_{\mathcal{M}}(x_i, x_j) + \varepsilon^2), \quad \forall x_i, x_j \in \mathcal{X}_n, \tag{F.3}$$

for some deterministic constant $L_s$. Using the fact that $\sqrt{\sigma_s(\mathcal{A}_{\mathbf{n}})}$ is an order one quantity (according to Remark E.1) in combination with Lemma F.1, we deduce the existence of a $CL_s$-Lipschitz function $g_s : \mathcal{M} \to \mathbb{R}$ satisfying

$$\|g_s - \sqrt{\sigma_s(\mathcal{A}_{\mathbf{n}})}u_s\|_{L^\infty(\mathcal{X}_n)} \leq 5CL_s\varepsilon^2. \tag{F.4}$$

In turn, Theorem E.3 implies the existence of parameters $\kappa, L, p, N$ as in the statement of the theorem and a (scalar-valued) neural network $f_{\tilde{\theta}}$ in the class $\mathcal{F}(1, \kappa, L, p, N)$ such that

$$\|f_{\tilde{\theta}}(x) - g_s(x)\|_{L^\infty(\mathcal{M})} \leq \delta. \tag{F.5}$$

Using the fact that the ReLU is a homogeneous function of degree one, we can deduce that

$$\frac{1}{\sqrt{n}}f_{\tilde{\theta}} = f_\theta,$$

where $\theta \overset{\text{def}}{=} \frac{1}{n^{1/(2L)}}\tilde{\theta}$ and thus $f_\theta \in \mathcal{F}(1, \frac{\kappa}{n^{1/(2L)}}, L, p, N)$. It follows that the neural network $f_\theta$ satisfies

$$\sqrt{n}\|f_\theta - \frac{1}{\sqrt{n}}g_s\|_{L^\infty(\mathcal{M})} \leq \delta,$$

and also, thanks to Equation F.4,

$$\sqrt{n}\left\|f_\theta - \sqrt{\frac{\sigma_s(\mathcal{A}_{\mathbf{n}})}{n}}u_s\right\|_{L^\infty(\mathcal{X}_n)} \leq (5CL_s + 1)(\delta + \varepsilon^2).$$

Stacking the scalar neural networks constructed above to approximate each of the functions $u_s$ for $s = 1, \ldots r$, and using Equation D.2, we obtain the desired vector valued neural network approximating $\mathbf{Y}^*$.

$\square$

**Remark F.2.** *Notice that the term $\sqrt{n}\|\mathbf{Y}^*\|_{\infty,\infty}$ is of order one. Consequently, the estimate in Theorem 2.1 is a non-trivial error bound.*

The bound in $\|\cdot\|_{\infty,\infty}$ between $\mathbf{Y}_\theta$ and $\mathbf{Y}^*$ in Theorem 2.1 can be used to bound the difference between $\mathbf{Y}_\theta\mathbf{Y}_\theta^\top$ and $\mathbf{Y}^*\mathbf{Y}^{*\top}$ in $\|\cdot\|_{\infty,\infty}$.

**Corollary 3.** *For $f_\theta$ as in Theorem 2.1 we have*

$$\sqrt{n}\|\mathbf{Y}_\theta\mathbf{Y}_\theta^\top - \mathbf{Y}^*\mathbf{Y}^{*\top}\|_{\infty,\infty} \leq C_r(\delta + \varepsilon^2), \tag{F.6}$$

*and thus also*

$$\|\mathbf{Y}_\theta\mathbf{Y}_\theta^\top - \mathbf{Y}^*\mathbf{Y}^{*\top}\|_{\mathrm{F}} \leq \sqrt{r}C_r(\delta + \varepsilon^2),$$

*for some deterministic constant $C_r$.*

*Proof.*

$$\sqrt{n}\|\mathbf{Y}_\theta\mathbf{Y}_\theta^\top - \mathbf{Y}^*\mathbf{Y}^{*\top}\|_{\infty,\infty} = \sqrt{n}\|\mathbf{Y}_\theta\left(\mathbf{Y}_\theta^\top - \mathbf{Y}^{*\top}\right) + \left(\mathbf{Y}_\theta - \mathbf{Y}^*\right)\mathbf{Y}^{*\top}\|_{\infty,\infty}$$
$$\leq \sqrt{n}\|\mathbf{Y}_\theta\left(\mathbf{Y}_\theta^\top - \mathbf{Y}^{*\top}\right)\|_{\infty,\infty} + \sqrt{n}\|\left(\mathbf{Y}_\theta - \mathbf{Y}^*\right)\mathbf{Y}^{*\top}\|_{\infty,\infty}$$
$$\leq \sqrt{nr}\|\mathbf{Y}_\theta\|_{\mathrm{F}}\|\mathbf{Y}_\theta^\top - \mathbf{Y}^{*\top}\|_{\infty,\infty} + \sqrt{nr}\|\mathbf{Y}_\theta - \mathbf{Y}^*\|_{\infty,\infty}\|\mathbf{Y}^{*\top}\|_{\mathrm{F}}$$
$$\leq \sqrt{r}(C_r(\delta + \varepsilon^2) + 2\|\mathbf{Y}^*\|_{\mathrm{F}})C_r(\delta + \varepsilon^2)$$
$$\leq C_r(\delta + \varepsilon^2),$$

where the second to last inequality follows from our estimate for $\sqrt{n}\|\mathbf{Y}_\theta - \mathbf{Y}^*\|_{\infty,\infty} \leq C_r(\delta + \varepsilon^2)$ in Theorem 2.1, and the last inequality follows from the fact that $\|\mathbf{Y}^*\|_{\mathrm{F}}^2 = \sum_{s=1}^{r} \sigma_s(\mathcal{A}_\mathbf{n}) = \mathcal{O}(r)$. □

## F.1 EIGENFUNCTION APPROXIMATION

The neural network $f_\theta$ constructed in the proof of Theorem 2.1 can be used to approximate eigenfunctions of $\Delta_\rho$. We restate Corollary 1 for the convenience of the reader.

**Corollary 1.** *Under the same setting, notation, and assumptions as in Theorem 2.1, the neural network $f_\theta : \mathbb{R}^d \to \mathbb{R}^r$ can be chosen to satisfy*

$$\left\|\sqrt{\frac{n}{1+a}}f_\theta^i - f_i\right\|_{L^\infty(\mathcal{M})} \leq C(\delta + \varepsilon), \quad \forall i = 1,\ldots,r.$$

*In the above, $f_\theta^1, \ldots, f_\theta^r$ are the coordinate functions of the vector-valued neural network $f_\theta$, and the functions $f_1, \ldots, f_r$ are the normalized eigenfunctions of the Laplace-Beltrami operator $\Delta_\rho$ that are associated to $\Delta_\rho$'s $r$ smallest eigenvalues.*

*Proof.* Let $g_s : \mathcal{M} \to \mathbb{R}$ be the Lipschitz function appearing in Equation F.4 and recall that the scalar neural network $f_\theta$ constructed in the proof of Theorem 2.1 satisfies

$$\sqrt{n}\|f_\theta - \frac{1}{\sqrt{n}}g_s\|_{L^\infty(\mathcal{M})} \leq \delta. \tag{F.7}$$

It can be shown that except on an event with probability less than $n\exp(-n\varepsilon^m)$, for any $x \in \mathcal{M}$, there exists $x_i \in \mathcal{X}_n$ such that $d_\mathcal{M}(x_i, x) \leq \varepsilon$. From the triangle inequality, it thus follows that

$$|f_s(x) - \sqrt{n/(1+a)}f_\theta(x)| \leq |f_s(x) - f_s(x_i)| + |f_s(x_i) - u_s(x_i)|$$
$$+ |u_s(x_i) - \frac{1}{\sqrt{\sigma_s(\mathcal{A}_\mathbf{n})}}g_s(x_i)| + |\frac{1}{\sqrt{\sigma_s(\mathcal{A}_\mathbf{n})}}g_s(x_i) - \frac{1}{\sqrt{1+a}}g_s(x_i)|$$
$$+ |\frac{1}{\sqrt{1+a}}g_s(x_i) - \frac{1}{\sqrt{1+a}}g_s(x)|$$
$$+ |\frac{1}{\sqrt{1+a}}g_s(x) - \sqrt{\frac{n}{1+a}}f_\theta(x)|$$
$$\leq C_s(\delta + \varepsilon), \tag{F.8}$$

where we have used the Lipschitz continuity of $f_s$ and $g_s$, Theorem E.2, Remark E.1, and Equation F.7.

□

**Remark F.3.** *We notice that, while one could use existing memorization results (e.g., Theorem 3.1 in (Yun et al., 2019)) to show that there is a neural network with ReLU activation function and $\mathcal{O}(\sqrt{n})$ neurons that fits $\mathbf{Y}^*$ perfectly, this does not constitute an improvement over our results in Theorem 2.1 and Corollary 1. Indeed, by using this type of memorization result, we can not state any bounds on the size of the parameters the network, and none of the out-of-sample generalization properties that we have discussed before (i.e., approximation of eigenfunctions of $\Delta_\rho$) can be guaranteed.*

## G  PROOF OF THEOREM 2.2

Recall that that $f_{\theta^*} \in \arg\min_{f_\theta \in \mathcal{F}(r,\kappa,L,p,N)} \|\mathbf{Y}_\theta \mathbf{Y}_\theta^\top - \mathcal{A}_\mathbf{n}\|_\mathrm{F}^2$. We start our proof with a lemma from linear algebra.

**Lemma G.1.** *For any* $\mathbf{Y} \in \mathbb{R}^{n \times r}$ *we have*

$$\|\mathbf{Y}\mathbf{Y}^\top - \mathcal{A}_\mathbf{n}\|_\mathrm{F}^2 - \|\mathbf{Y}^*\mathbf{Y}^{*\top} - \mathcal{A}_\mathbf{n}\|_\mathrm{F}^2 \le \|\mathbf{Y}\mathbf{Y}^\top - \mathbf{Y}^*\mathbf{Y}^{*\top}\|_\mathrm{F}^2.$$

*Proof.* A straightforward computation reveals that

$$
\begin{aligned}
&\|\mathbf{Y}\mathbf{Y}^\top - \mathcal{A}_\mathbf{n}\|_\mathrm{F}^2 - \|\mathbf{Y}^*\mathbf{Y}^{*\top} - \mathcal{A}_\mathbf{n}\|_\mathrm{F}^2 \\
&= \|(\mathbf{Y}\mathbf{Y}^\top - \mathbf{Y}^*\mathbf{Y}^{*\top}) + (\mathbf{Y}^*\mathbf{Y}^{*\top} - \mathcal{A}_\mathbf{n})\|_\mathrm{F}^2 - \|\mathbf{Y}^*\mathbf{Y}^{*\top} - \mathcal{A}_\mathbf{n}\|_\mathrm{F}^2 \\
&= \|\mathbf{Y}\mathbf{Y}^\top - \mathbf{Y}^*\mathbf{Y}^{*\top}\|_\mathrm{F}^2 + 2\langle \mathbf{Y}\mathbf{Y}^\top - \mathbf{Y}^*\mathbf{Y}^{*\top}, \mathbf{Y}^*\mathbf{Y}^{*\top} - \mathcal{A}_\mathbf{n}\rangle \qquad\text{(G.1)} \\
&= \|\mathbf{Y}\mathbf{Y}^\top - \mathbf{Y}^*\mathbf{Y}^{*\top}\|_\mathrm{F}^2 + 2\langle \mathbf{Y}\mathbf{Y}^\top, \mathbf{Y}^*\mathbf{Y}^{*\top} - \mathcal{A}_\mathbf{n}\rangle \\
&\le \|\mathbf{Y}\mathbf{Y}^\top - \mathbf{Y}^*\mathbf{Y}^{*\top}\|_\mathrm{F}^2,
\end{aligned}
$$

where the last inequality follows thanks to the fact that $\mathbf{Y}\mathbf{Y}^\top$ is positive semi-definite and the fact that $\mathbf{Y}^*\mathbf{Y}^{*\top} - \mathcal{A}_\mathbf{n}$ is negative semi-definite, as can be easily deduced from the form of $\mathbf{Y}^*$ discussed in section D.2. $\qquad\square$

Invoking Corollary 3 with $\delta = \tilde{\delta}\varepsilon$ we immediately obtain the following approximation estimate.

**Corollary 4.** *With probability at least*

$$1 - C\varepsilon^{-6m}\exp\left(-cn\varepsilon^{m+4}\right),$$

*for every* $\tilde{\delta} \in (0,1)$ *there is* $f_\theta \in \mathcal{F}(r,\kappa,L,p,N)$ *with* $\kappa, L, p, N$ *as specified in Theorem 2.2 such that*

$$\|\mathbf{Y}_\theta \mathbf{Y}_\theta^\top - \mathbf{Y}^*\mathbf{Y}^{*\top}\|_\mathrm{F} \le C_r\varepsilon(\tilde{\delta} + \varepsilon). \qquad\text{(G.2)}$$

**Corollary 5.** *Let* $f_\theta$ *be as in Corollary 4. Then*

$$\|\mathbf{Y}_\theta \mathbf{Y}_\theta^\top - \mathcal{A}_\mathbf{n}\|_\mathrm{F}^2 - \|\mathbf{Y}^*\mathbf{Y}^{*\top} - \mathcal{A}_\mathbf{n}\|_\mathrm{F}^2 \le C_r\varepsilon^2(\tilde{\delta} + \varepsilon)^2.$$

*Proof.* Let $\theta$ be as in Corollary 4. Then

$$\|\mathbf{Y}_\theta \mathbf{Y}_\theta^\top - \mathcal{A}_\mathbf{n}\|_\mathrm{F}^2 - \|\mathbf{Y}^*\mathbf{Y}^{*\top} - \mathcal{A}_\mathbf{n}\|_\mathrm{F}^2 \le \|\mathbf{Y}_\theta \mathbf{Y}_\theta^\top - \mathbf{Y}^*\mathbf{Y}^{*\top}\|_\mathrm{F}^2 \le C_r^2\varepsilon^2(\tilde{\delta} + \varepsilon)^2,$$

where the second to last inequality follows from Lemma G.1. $\qquad\square$

In what follows we will write the SVD (eigendecomposition) of $\mathcal{A}_\mathbf{n}$ as $\overline{\mathbf{U}}\mathbf{\Sigma}\overline{\mathbf{U}}^\top$. Using the fact that $\overline{\mathbf{U}}$ is invertible (since it is an ortogonal matrix), we can easily see that $\mathbf{Y}_{\theta^*}$ can be written as $\mathbf{Y}_{\theta^*} = \overline{\mathbf{U}}(\mathbf{E}^1 + \mathbf{E}^2)$ where $\mathbf{E}^1, \mathbf{E}^2 \in \mathbb{R}^{n \times r}$ are such that the $i^\mathrm{th}$ row $\mathbf{E}_i^1 = \mathbf{0}$ for $i \ge r+1$, and $i^\mathrm{th}$ row $\mathbf{E}_i^2 = \mathbf{0}$ for $i \le r$. Indeed, it suffices to select $\mathbf{E}_1$ and $\mathbf{E}_2$ so as to have $\mathbf{E}^1 + \mathbf{E}^2 = \overline{\mathbf{U}}^{-1}\mathbf{Y}_{\theta^*}$. We thus have $(\mathbf{E}^2)^\top \mathbf{E}^1 = \mathbf{0}$.

In what follows we will make the following assumption.

**Assumption G.1.** $\varepsilon$ *and* $\tilde{\delta}$ *in Corollary 4 satisfy the following condition:*

$$\varepsilon^2 E < \sigma_r^2(\mathcal{A}_\mathbf{n}) - \sigma_{r+1}^2(\mathcal{A}_\mathbf{n}), \qquad\text{(G.3)}$$

*where* $E \overset{def}{=} C_r(\tilde{\delta} + \varepsilon)^2$.

**Remark G.1.** *Assumption G.1 is satisfied under the assumptions in the statement of Theorem 2.2. To see this, notice that* $\sigma_r^2(\mathcal{A}_\mathbf{n}) - \sigma_{r+1}^2(\mathcal{A}_\mathbf{n}) \sim \varepsilon^2$ *according to Remark E.1 and the fact that* $\lambda_r^\mathcal{M} < \lambda_{r+1}^\mathcal{M}$. *Thus, taking* $\tilde{\delta}$ *to be sufficiently small, we can guarantee that indeed* $\varepsilon^2 E < \sigma_r^2(\mathcal{A}_\mathbf{n}) - \sigma_{r+1}^2(\mathcal{A}_\mathbf{n})$.

**Remark G.2.** *Returning to Remark E.2, if the correction term in the Lipschitz estimate for graph Laplacian eigenvectors had been $\varepsilon$, and not $\varepsilon^2$, the term $\varepsilon^2 E$ would have to be replaced with the term $(C_r \varepsilon \tilde{\delta} + C_r \varepsilon)^2$, but the latter cannot be guaranteed to be smaller than $\sigma_r^2(\mathcal{A}_\mathbf{n}) - \sigma_{r+1}^2(\mathcal{A}_\mathbf{n})$.*

**Remark G.3.** *The energy gap between $\mathbf{Y}^*$ and the constructed $\mathbf{Y}_\theta$ is, according to Corollary 5, $\varepsilon^2 E$, whereas the energy gap between $\mathbf{Y}^*$ and any other critical point of $\ell$ that is not a global optimizer is in the order of $\varepsilon^2$, as it follows from Remark E.1. Continuing the discussion from Remark G.2, it was thus relevant to use estimates that could guarantee that, at least energetically, our constructed $\mathbf{Y}_\theta$ was closer to $\mathbf{Y}^*$ than any other saddle of $\ell$.*

*Proof of Theorem 2.2.* Due to the definition of $\theta^*$, we have

$$\|\mathbf{Y}^*\mathbf{Y}^{*\top} - \mathcal{A}_\mathbf{n}\|_{\mathrm{F}}^2 \le \|\mathbf{Y}_{\theta^*}\mathbf{Y}_{\theta^*}^\top - \mathcal{A}_\mathbf{n}\|_{\mathrm{F}}^2 \le \|\mathbf{Y}_\theta \mathbf{Y}_\theta^\top - \mathcal{A}_\mathbf{n}\|_{\mathrm{F}}^2. \tag{G.4}$$

Also,

$$
\begin{aligned}
0 &\ge \|\mathbf{Y}_{\theta^*}\mathbf{Y}_{\theta^*}^\top - \mathcal{A}_\mathbf{n}\|_{\mathrm{F}}^2 - \|\mathbf{Y}_\theta \mathbf{Y}_\theta^\top - \mathcal{A}_\mathbf{n}\|_{\mathrm{F}}^2 \\
&= \|(\mathbf{Y}_{\theta^*}\mathbf{Y}_{\theta^*}^\top - \mathbf{Y}^*\mathbf{Y}^{*\top}) + (\mathbf{Y}^*\mathbf{Y}^{*\top} - \mathcal{A}_\mathbf{n})\|_{\mathrm{F}}^2 - \|\mathbf{Y}_\theta \mathbf{Y}_\theta^\top - \mathcal{A}_\mathbf{n}\|_{\mathrm{F}}^2 \\
&= \|\mathbf{Y}_{\theta^*}\mathbf{Y}_{\theta^*}^\top - \mathbf{Y}^*\mathbf{Y}^{*\top}\|_{\mathrm{F}}^2 + \|\mathbf{Y}^*\mathbf{Y}^{*\top} - \mathcal{A}_\mathbf{n}\|_{\mathrm{F}}^2 + 2\langle \mathbf{Y}_{\theta^*}\mathbf{Y}_{\theta^*}^\top - \mathbf{Y}^*\mathbf{Y}^{*\top}, \mathbf{Y}^*\mathbf{Y}^{*\top} - \mathcal{A}_\mathbf{n} \rangle - \|\mathbf{Y}_\theta \mathbf{Y}_\theta^\top - \mathcal{A}_\mathbf{n}\|_{\mathrm{F}}^2 \\
&= \|\mathbf{Y}_{\theta^*}\mathbf{Y}_{\theta^*}^\top - \mathbf{Y}^*\mathbf{Y}^{*\top}\|_{\mathrm{F}}^2 + \|\mathbf{Y}^*\mathbf{Y}^{*\top} - \mathcal{A}_\mathbf{n}\|_{\mathrm{F}}^2 + 2\langle \mathbf{Y}_{\theta^*}\mathbf{Y}_{\theta^*}^\top, \mathbf{Y}^*\mathbf{Y}^{*\top} - \mathcal{A}_\mathbf{n} \rangle - \|\mathbf{Y}_\theta \mathbf{Y}_\theta^\top - \mathcal{A}_\mathbf{n}\|_{\mathrm{F}}^2
\end{aligned}
\tag{G.5}
$$

where the third equality follows from the fact that $\langle \mathbf{Y}^*\mathbf{Y}^{*\top}, \mathbf{Y}^*\mathbf{Y}^{*\top} - \mathcal{A}_\mathbf{n} \rangle = 0$. Notice that

$$\|\mathbf{Y}_{\theta^*}\mathbf{Y}_{\theta^*}^\top - \mathbf{Y}^*\mathbf{Y}^{*\top}\|_{\mathrm{F}}^2 + 2\langle \mathbf{Y}_{\theta^*}\mathbf{Y}_{\theta^*}^\top, \mathbf{Y}^*\mathbf{Y}^{*\top} - \mathcal{A}_\mathbf{n} \rangle = \|\mathbf{Y}_{\theta^*}\mathbf{Y}_{\theta^*}^\top\|_{\mathrm{F}}^2 + \|\mathbf{Y}^*\mathbf{Y}^{*\top}\|_{\mathrm{F}}^2 - 2\langle \mathbf{Y}_{\theta^*}\mathbf{Y}_{\theta^*}^\top, \mathcal{A}_\mathbf{n} \rangle \tag{G.6}$$

By combining Equation G.5, Lemma 5 and Equation G.6, we have

$$\|\mathbf{Y}_{\theta^*}\mathbf{Y}_{\theta^*}^\top\|_{\mathrm{F}}^2 + \|\mathbf{Y}^*\mathbf{Y}^{*\top}\|_{\mathrm{F}}^2 - 2\langle \mathbf{Y}_{\theta^*}\mathbf{Y}_{\theta^*}^\top, \mathcal{A}_\mathbf{n} \rangle \le \varepsilon^2 E \tag{G.7}$$

From $(\mathbf{E}^1)^\top \mathbf{E}^2 = \mathbf{0}$ and $\mathrm{Tr}(AB) = \mathrm{Tr}(BA)$, we have

$$
\begin{aligned}
\langle \mathbf{E}^1(\mathbf{E}^1)^\top, \mathbf{E}^2(\mathbf{E}^2)^\top \rangle &= 0 \\
\langle \mathbf{E}^1(\mathbf{E}^2)^\top, \mathbf{E}^2(\mathbf{E}^2)^\top \rangle &= 0 \\
\langle \mathbf{E}^1(\mathbf{E}^2)^\top, \mathbf{E}^1(\mathbf{E}^1)^\top \rangle &= 0 \\
\langle \mathbf{E}^2(\mathbf{E}^1)^\top, \mathbf{E}^1(\mathbf{E}^1)^\top \rangle &= 0 \\
\langle \mathbf{E}^2(\mathbf{E}^1)^\top, \mathbf{E}^2(\mathbf{E}^2)^\top \rangle &= 0
\end{aligned}
\tag{G.8}
$$

Let $\mathbf{\Sigma}^1$ be the diagonal matrix such that $(\mathbf{\Sigma}^1)_{ii} = \mathbf{\Sigma}_{ii}$ for $i \le r$, and $(\mathbf{\Sigma}^1)_{ii} = 0$ for $i > r$; let $\mathbf{\Sigma}^2$ be the diagonal matrix such that $(\mathbf{\Sigma}^2)_{ii} = 0$ for $i \le r$, and $(\mathbf{\Sigma}^2)_{ii} = \mathbf{\Sigma}_{ii}$ for $i > r$. By plugging the

decomposition of $\mathbf{Y}_{\theta*}$ in Equation G.7, we deduce

$$
\begin{aligned}
\varepsilon^2 E \geq & \|\mathbf{Y}_{\theta*}\mathbf{Y}_{\theta*}^\top\|_{\mathrm{F}}^2 + \|\mathbf{Y}^*\mathbf{Y}^{*\top}\|_{\mathrm{F}}^2 - 2\langle\mathbf{Y}_{\theta*}\mathbf{Y}_{\theta*}^\top, \mathcal{A}_{\mathbf{n}}\rangle \\
= & \|\overline{\mathbf{U}}(\mathbf{E}^1+\mathbf{E}^2)(\mathbf{E}^1+\mathbf{E}^2)^\top\overline{\mathbf{U}}^\top\|_{\mathrm{F}}^2 + \|\mathbf{Y}^*\mathbf{Y}^{*\top}\|_{\mathrm{F}}^2 - 2\langle\overline{\mathbf{U}}(\mathbf{E}^1+\mathbf{E}^2)(\mathbf{E}^1+\mathbf{E}^2)^\top\overline{\mathbf{U}}^\top, \mathcal{A}_{\mathbf{n}}\rangle \\
= & \|(\mathbf{E}^1+\mathbf{E}^2)(\mathbf{E}^1+\mathbf{E}^2)^\top\|_{\mathrm{F}}^2 + \|\mathbf{Y}^*\mathbf{Y}^{*\top}\|_{\mathrm{F}}^2 - 2\langle(\mathbf{E}^1+\mathbf{E}^2)(\mathbf{E}^1+\mathbf{E}^2)^\top, \boldsymbol{\Sigma}\rangle \\
\overset{\text{Equation G.8}}{=} & \|\mathbf{E}^1(\mathbf{E}^1)^\top\|_{\mathrm{F}}^2 + \|\mathbf{E}^2(\mathbf{E}^2)^\top\|_{\mathrm{F}}^2 + 2\|\mathbf{E}^2(\mathbf{E}^1)^\top\|_{\mathrm{F}}^2 + 2\langle(\mathbf{E}^1)^\top\mathbf{E}^1, (\mathbf{E}^2)^\top\mathbf{E}^2\rangle \\
& + \|\mathbf{Y}^*\mathbf{Y}^{*\top}\|_{\mathrm{F}}^2 - 2\langle(\mathbf{E}^1+\mathbf{E}^2)(\mathbf{E}^1+\mathbf{E}^2)^\top, \boldsymbol{\Sigma}\rangle \\
\overset{(\mathbf{E}^1)^\top\boldsymbol{\Sigma}\mathbf{E}^2=\mathbf{0}}{=} & \|\mathbf{E}^1(\mathbf{E}^1)^\top\|_{\mathrm{F}}^2 + \|\mathbf{E}^2(\mathbf{E}^2)^\top\|_{\mathrm{F}}^2 + 2\|\mathbf{E}^2(\mathbf{E}^1)^\top\|_{\mathrm{F}}^2 + 2\langle(\mathbf{E}^1)^\top\mathbf{E}^1, (\mathbf{E}^2)^\top\mathbf{E}^2\rangle \\
& + \|\mathbf{Y}^*\mathbf{Y}^{*\top}\|_{\mathrm{F}}^2 - 2\langle\mathbf{E}^1(\mathbf{E}^1)^\top + \mathbf{E}^2(\mathbf{E}^2)^\top, \boldsymbol{\Sigma}\rangle \\
= & \|\mathbf{E}^1(\mathbf{E}^1)^\top\|_{\mathrm{F}}^2 + \|\mathbf{E}^2(\mathbf{E}^2)^\top\|_{\mathrm{F}}^2 + 2\|\mathbf{E}^2(\mathbf{E}^1)^\top\|_{\mathrm{F}}^2 + 2\langle(\mathbf{E}^1)^\top\mathbf{E}^1, (\mathbf{E}^2)^\top\mathbf{E}^2\rangle \\
& + \|\boldsymbol{\Sigma}^1\|_{\mathrm{F}}^2 - 2\langle\mathbf{E}^1(\mathbf{E}^1)^\top, \boldsymbol{\Sigma}^1\rangle - 2\langle\mathbf{E}^2(\mathbf{E}^2)^\top, \boldsymbol{\Sigma}^2\rangle \\
= & \|\mathbf{E}^1(\mathbf{E}^1)^\top - \boldsymbol{\Sigma}^1\|_{\mathrm{F}}^2 + \|\mathbf{E}^2(\mathbf{E}^2)^\top\|_{\mathrm{F}}^2 + 2\|\mathbf{E}^2(\mathbf{E}^1)^\top\|_{\mathrm{F}}^2 + 2\langle(\mathbf{E}^1)^\top\mathbf{E}^1, (\mathbf{E}^2)^\top\mathbf{E}^2\rangle \\
& - 2\langle\mathbf{E}^2(\mathbf{E}^2)^\top, \boldsymbol{\Sigma}^2\rangle \\
\geq & \|\mathbf{E}^1(\mathbf{E}^1)^\top - \boldsymbol{\Sigma}^1\|_{\mathrm{F}}^2 + \|\mathbf{E}^2(\mathbf{E}^2)^\top\|_{\mathrm{F}}^2 + 2\|\mathbf{E}^2(\mathbf{E}^1)^\top\|_{\mathrm{F}}^2 + 2\langle(\mathbf{E}^1)^\top\mathbf{E}^1, (\mathbf{E}^2)^\top\mathbf{E}^2\rangle \\
& - 2\|\mathbf{E}^2(\mathbf{E}^2)^\top\|_{\mathrm{F}} \cdot \sigma_{r+1}(\mathcal{A}_{\mathbf{n}}) \\
\geq & \|\mathbf{E}^1(\mathbf{E}^1)^\top - \boldsymbol{\Sigma}^1\|_{\mathrm{F}}^2 + \|\mathbf{E}^2(\mathbf{E}^2)^\top\|_{\mathrm{F}}^2 + (2\|\mathbf{E}^2\|_{\mathrm{F}}^2 + 2\|\mathbf{E}^2(\mathbf{E}^2)^\top\|_{\mathrm{F}}) \cdot \sigma_r^2(\mathbf{E}^1) \\
& - 2\|\mathbf{E}^2(\mathbf{E}^2)^\top\|_{\mathrm{F}} \cdot \sigma_{r+1}(\mathcal{A}_{\mathbf{n}}).
\end{aligned}
\tag{G.9}
$$

On the other hand, we have

$$
\begin{aligned}
\|\mathbf{Y}_{\theta*}\mathbf{Y}_{\theta*}^\top - \mathbf{Y}^*\mathbf{Y}^{*\top}\|_{\mathrm{F}}^2 = & \|\mathbf{Y}_{\theta*}\mathbf{Y}_{\theta*}^\top\|_{\mathrm{F}}^2 + \|\mathbf{Y}^*\mathbf{Y}^{*\top}\|_{\mathrm{F}}^2 - 2\langle\mathbf{Y}_{\theta*}\mathbf{Y}_{\theta*}^\top, \mathcal{A}_{\mathbf{n}}\rangle + 2\langle\mathbf{Y}_{\theta*}\mathbf{Y}_{\theta*}^\top, \mathcal{A}_{\mathbf{n}} - \mathbf{Y}^*\mathbf{Y}^{*\top}\rangle \\
= & \|\mathbf{Y}_{\theta*}\mathbf{Y}_{\theta*}^\top\|_{\mathrm{F}}^2 + \|\mathbf{Y}^*\mathbf{Y}^{*\top}\|_{\mathrm{F}}^2 - 2\langle\mathbf{Y}_{\theta*}\mathbf{Y}_{\theta*}^\top, \mathcal{A}_{\mathbf{n}}\rangle + 2\langle\mathbf{E}^2(\mathbf{E}^2)^\top, \boldsymbol{\Sigma}^2\rangle \\
\leq & \varepsilon^2 E + 2\|\mathbf{E}^2(\mathbf{E}^2)^\top\|_{\mathrm{F}} \cdot \sigma_{r+1}(\mathcal{A}_{\mathbf{n}}).
\end{aligned}
\tag{G.10}
$$

It remains to show that $\|\mathbf{E}^2(\mathbf{E}^2)^\top\|_{\mathrm{F}}$ can be controlled by a term of the form $C\varepsilon^2 E$. We split the following discussion into two cases. First, we assume that $\sigma_r^2(\mathbf{E}^1)$ is large compared with $\sigma_{r+1}(\mathcal{A}_{\mathbf{n}})$. In this first case $\|\mathbf{E}^2(\mathbf{E}^2)^\top\|_{\mathrm{F}}$ can be guaranteed to be small according to Equation G.9. Second, when $\sigma_r^2(\mathbf{E}^1)$ is small, we'll show that $\|\mathbf{E}^1(\mathbf{E}^1)^\top - \boldsymbol{\Sigma}^1\|_{\mathrm{F}}^2$ is large, which will contradict Equation G.9.

**Case 1: If $\sigma_r^2(\mathbf{E}^1) \geq \frac{2}{3}\sigma_{r+1}(\mathcal{A}_{\mathbf{n}})$.**

We have $3\|\mathbf{E}^2\|_{\mathrm{F}}^2 \cdot \sigma_r^2(\mathbf{E}^1) - 2\|\mathbf{E}^2(\mathbf{E}^2)^\top\|_{\mathrm{F}} \cdot \sigma_{r+1}(\mathcal{A}_{\mathbf{n}}) \geq 0$. Then, from Equation G.9 and the fact that $\|AB\|_{\mathrm{F}} \leq \|A\|_{\mathrm{F}} \cdot \|B\|_{\mathrm{F}}$, we have

$$
\|\mathbf{E}^1(\mathbf{E}^1)^\top - \boldsymbol{\Sigma}^1\|_{\mathrm{F}}^2 + \|\mathbf{E}^2(\mathbf{E}^2)^\top\|_{\mathrm{F}}^2 + 2\|\mathbf{E}^2(\mathbf{E}^2)^\top\|_{\mathrm{F}} \cdot \sigma_{r+1}(\mathcal{A}_{\mathbf{n}}) \leq \varepsilon^2 E.
\tag{G.11}
$$

This immediately implies

$$
\|\mathbf{E}^2(\mathbf{E}^2)^\top\|_{\mathrm{F}} \leq \frac{\varepsilon^2 E}{\sigma_{r+1}(\mathcal{A}_{\mathbf{n}})}.
\tag{G.12}
$$

Combining Equation G.12 and Equation G.10, we obtain

$$
\|\mathbf{Y}_{\theta*}\mathbf{Y}_{\theta*}^\top - \mathbf{Y}^*\mathbf{Y}^{*\top}\|_{\mathrm{F}}^2 \leq \varepsilon^2 E + \|\mathbf{E}^2(\mathbf{E}^2)^\top\|_{\mathrm{F}} \cdot \sigma_{r+1}(\mathcal{A}_{\mathbf{n}}) \leq 2\varepsilon^2 E.
\tag{G.13}
$$

**Case 2: If $0 \leq \sigma_r^2(\mathbf{E}^1) < \frac{2}{3}\sigma_{r+1}(\mathcal{A}_{\mathbf{n}})$.**

Invoking Equation G.9, we have

$$
\begin{aligned}
\varepsilon^2 E &\geq \|\mathbf{E}^1(\mathbf{E}^1)^\top - \mathbf{\Sigma}^1\|_{\mathrm{F}}^2 + \|\mathbf{E}^2(\mathbf{E}^2)^\top\|_{\mathrm{F}}^2 + (2\|\mathbf{E}^2\|_{\mathrm{F}}^2 + 2\|\mathbf{E}^2(\mathbf{E}^2)^\top\|_{\mathrm{F}}) \cdot \sigma_r^2(\mathbf{E}^1) - 2\|\mathbf{E}^2(\mathbf{E}^2)^\top\|_{\mathrm{F}} \cdot \sigma_{r+1}(\mathcal{A}_{\mathbf{n}}) \\
&\geq (\sigma_r^2(\mathbf{E}^1) - \sigma_r(\mathcal{A}_{\mathbf{n}}))^2 + \|\mathbf{E}^2(\mathbf{E}^2)^\top\|_{\mathrm{F}}^2 + 4\|\mathbf{E}^2(\mathbf{E}^2)^\top\|_{\mathrm{F}} \cdot \sigma_r^2(\mathbf{E}^1) - 2\|\mathbf{E}^2(\mathbf{E}^2)^\top\|_{\mathrm{F}} \cdot \sigma_{r+1}(\mathcal{A}_{\mathbf{n}}) \\
&= (\sigma_r^2(\mathbf{E}^1) - \sigma_r(\mathcal{A}_{\mathbf{n}}))^2 + \|\mathbf{E}^2(\mathbf{E}^2)^\top\|_{\mathrm{F}}^2 - 2\|\mathbf{E}^2(\mathbf{E}^2)^\top\|_{\mathrm{F}} \cdot (\sigma_{r+1}(\mathcal{A}_{\mathbf{n}}) - 2\sigma_r^2(\mathbf{E}^1)) \\
&= (\sigma_r^2(\mathbf{E}^1) - \sigma_r(\mathcal{A}_{\mathbf{n}}))^2 + \left(\|\mathbf{E}^2(\mathbf{E}^2)^\top\|_{\mathrm{F}} - (\sigma_{r+1}(\mathcal{A}_{\mathbf{n}}) - 2\sigma_r^2(\mathbf{E}^1))\right)^2 - (\sigma_{r+1}(\mathcal{A}_{\mathbf{n}}) - 2\sigma_r^2(\mathbf{E}^1))^2 \\
&\geq (\sigma_r^2(\mathbf{E}^1) - \sigma_r(\mathcal{A}_{\mathbf{n}}))^2 - (\sigma_{r+1}(\mathcal{A}_{\mathbf{n}}) - 2\sigma_r^2(\mathbf{E}^1))^2,
\end{aligned}
\tag{G.14}
$$

where the second inequality follows from Weyl's inequality (Stewart, 1998).

It is straightforward to check that $(\sigma_r^2(\mathbf{E}^1) - \sigma_r(\mathcal{A}_{\mathbf{n}}))^2 - (\sigma_{r+1}(\mathcal{A}_{\mathbf{n}}) - 2\sigma_r^2(\mathbf{E}^1))^2$ is a decreasing function with respect to $\sigma_r^2(\mathbf{E}^1)$ in the range $0 \leq \sigma_r^2(\mathbf{E}^1) < \frac{2}{3}\sigma_{r+1}(\mathcal{A}_{\mathbf{n}})$. The smallest value of $(\sigma_r^2(\mathbf{E}^1) - \sigma_r(\mathcal{A}_{\mathbf{n}}))^2 - (\sigma_{r+1}(\mathcal{A}_{\mathbf{n}}) - 2\sigma_r^2(\mathbf{E}^1))^2$ in this range is thus larger than $\frac{1}{9}(\sigma_r^2(\mathcal{A}_{\mathbf{n}}) - \sigma_{r+1}^2(\mathcal{A}_{\mathbf{n}}))$. However, the resulting inequality contradicts Assumption G.1. Case 2 is thus void.

By combining the aforementioned two cases, we conclude

$$
\|\mathbf{Y}_{\theta^*}\mathbf{Y}_{\theta^*}^\top - \mathbf{Y}^*\mathbf{Y}^{*\top}\|_{\mathrm{F}}^2 \leq 2E\varepsilon^2.
\tag{G.15}
$$

By using Equation H.3, we have

$$
d^2([\mathbf{Y}_{\theta^*}], [\mathbf{Y}^*]) \leq \frac{1}{2(\sqrt{2}-1)\sigma_r^2(\mathbf{Y}^*)}\|\mathbf{Y}_{\theta^*}\mathbf{Y}_{\theta^*}^\top - \mathbf{Y}^*\mathbf{Y}^{*\top}\|_{\mathrm{F}}^2 \leq \frac{\varepsilon^2 E}{(\sqrt{2}-1)\sigma_r^2(\mathbf{Y}^*)},
\tag{G.16}
$$

where $d([\mathbf{Y}_{\theta^*}], [\mathbf{Y}^*]) = \min_{\mathbf{O} \in \mathbb{O}_r}\|\mathbf{Y}_{\theta^*} - \mathbf{Y}^*\mathbf{O}\|_{\mathrm{F}}$. This completes the proof. $\qquad\square$

# H  AMBIENT OPTIMIZATION

This section contains the proof of the results from Section 3.

## H.1  SETUP FROM MAIN TEXT

Let us recall the quotient manifold that we are interested in. Let $\overline{\mathcal{N}}_{r+}^n$ be the space of $n \times r$ matrices with full column rank. To define the quotient manifold, we encode the invariance mapping, i.e., $\mathbf{Y} \to \mathbf{Y}\mathbf{O}$, by defining the equivalence classes $[\mathbf{Y}] = \{\mathbf{Y}\mathbf{O} : \mathbf{O} \in \mathbb{O}_r\}$. Since the invariance mapping is performed via the Lie group $\mathbb{O}_r$ smoothly, freely and properly, we have $\mathcal{N}_{r+}^n \stackrel{\text{def}}{=} \overline{\mathcal{N}}_{r+}^n / \mathbb{O}_r$ is a quotient manifold of $\overline{\mathcal{N}}_{r+}^n$ (Lee, 2018). Moreover, we equip the tangent space $T_{\mathbf{Y}}\overline{\mathcal{N}}_{r+}^n = \mathbb{R}^{n \times r}$ with the metric $\bar{g}_{\mathbf{Y}}(\eta_{\mathbf{Y}}, \theta_{\mathbf{Y}}) = \mathrm{tr}\left(\eta_{\mathbf{Y}}^\top \theta_{\mathbf{Y}}\right)$.

For convenience, we recall the following.

$$
\begin{aligned}
\overline{\mathrm{grad}\, H([\mathbf{Y}])} &= 2\left(\mathbf{Y}\mathbf{Y}^\top - \mathcal{A}_{\mathbf{n}}\right)\mathbf{Y}, \\
\overline{\mathrm{Hess}\, H([\mathbf{Y}])}\,[\theta_{\mathbf{Y}}, \theta_{\mathbf{Y}}] &= \left\|\mathbf{Y}\theta_{\mathbf{Y}}^\top + \theta_{\mathbf{Y}}\mathbf{Y}^\top\right\|_{\mathrm{F}}^2 + 2\left\langle \mathbf{Y}\mathbf{Y}^\top - \mathcal{A}_{\mathbf{n}}, \theta_{\mathbf{Y}}\theta_{\mathbf{Y}}^\top \right\rangle
\end{aligned}
\tag{H.1}
$$

$$
\begin{aligned}
\mathcal{R}_1 &\stackrel{\text{def}}{=} \left\{\mathbf{Y} \in \mathbb{R}_*^{n \times r}\,\middle|\, d([\mathbf{Y}], [\mathbf{Y}^*]) \leqslant \mu\sigma_r(\mathbf{Y}^*)/\kappa^*\right\}, \\
\mathcal{R}_2 &\stackrel{\text{def}}{=} \left\{\mathbf{Y} \in \mathbb{R}_*^{n \times r}\,\middle|\, \begin{array}{c} d([\mathbf{Y}], [\mathbf{Y}^*]) > \mu\sigma_r(\mathbf{Y}^*)/\kappa^*, \|\overline{\mathrm{grad}\, H([\mathbf{Y}])}\|_{\mathrm{F}} \leqslant \alpha\mu\sigma_r^3(\mathbf{Y}^*)/(4\kappa^*) \\ \|\mathbf{Y}\| \leqslant \beta\|\mathbf{Y}^*\|, \|\mathbf{Y}\mathbf{Y}^\top\|_{\mathrm{F}} \leqslant \gamma\|\mathbf{Y}^*\mathbf{Y}^{*\top}\|_{\mathrm{F}} \end{array}\right\}, \\
\mathcal{R}_3' &\stackrel{\text{def}}{=} \left\{\mathbf{Y} \in \mathbb{R}_*^{n \times r}\,\middle|\, \|\overline{\mathrm{grad}\, H([\mathbf{Y}])}\|_{\mathrm{F}} > \alpha\mu\sigma_r^3(\mathbf{Y}^*)/(4\kappa^*), \|\mathbf{Y}\| \leqslant \beta\|\mathbf{Y}^*\|, \|\mathbf{Y}\mathbf{Y}^\top\|_{\mathrm{F}} \leqslant \gamma\|\mathbf{Y}^*\mathbf{Y}^{*\top}\|_{\mathrm{F}}\right\}, \\
\mathcal{R}_3'' &\stackrel{\text{def}}{=} \left\{\mathbf{Y} \in \mathbb{R}_*^{n \times r}\,\middle|\, \|\mathbf{Y}\| > \beta\|\mathbf{Y}^*\|, \|\mathbf{Y}\mathbf{Y}^\top\|_{\mathrm{F}} \leqslant \gamma\|\mathbf{Y}^*\mathbf{Y}^{*\top}\|_{\mathrm{F}}\right\}, \\
\mathcal{R}_3''' &\stackrel{\text{def}}{=} \left\{\mathbf{Y} \in \mathbb{R}_*^{n \times r}\,\middle|\, \|\mathbf{Y}\mathbf{Y}^\top\|_{\mathrm{F}} > \gamma\|\mathbf{Y}^*\mathbf{Y}^{*\top}|_{\mathrm{F}}\right\},
\end{aligned}
\tag{H.2}
$$

**Remark H.1.** *To demonstrate strong geodesic convexity, the eigengap assumption is necessary as it prevents multiple global solutions. However, it is possible to relax this assumption and instead deduce a Polyak-Lojasiewicz condition, which would also imply a linear convergence rate for a first-order method.*

## H.2 SOME AUXILIARY INEQUALITIES

In this section, we collect results from prior work that will be useful for us. First, we provide the characterization of and results about the geodesic distance on $\mathcal{N}_{r+}^n$ from (Massart & Absil, 2020) and (Luo & García Trillos, 2022).

**Lemma H.1** (Lemma 2, (Luo & García Trillos, 2022)). *Let $\mathbf{Y}_1, \mathbf{Y}_2 \in \mathbb{R}_*^{n \times r}$, and $\mathbf{Q}_U \boldsymbol{\Sigma} \mathbf{Q}_V^\top$ be the SVD of $\mathbf{Y}_1^\top \mathbf{Y}_2$. Denote $\mathbf{Q}^* = \mathbf{Q}_V \mathbf{Q}_U^\top$. Then*

1. $\mathbf{Y}_2 \mathbf{Q}^* - \mathbf{Y}_1 \in \mathcal{H}_{\mathbf{Y}_1} \overline{\mathcal{N}}_{r+}^n$, $\mathbf{Q}^*$ *is one of the best orthogonal matrices aligning $\mathbf{Y}_1$ and $\mathbf{Y}_2$, i.e., $\mathbf{Q}^* \in \arg\min_{\mathbf{Q} \in \mathbb{O}_r} \|\mathbf{Y}_2 \mathbf{Q} - \mathbf{Y}_1\|_F$ and the geodesic distance between $[\mathbf{Y}_1]$ and $[\mathbf{Y}_2]$ is $d([\mathbf{Y}_1], [\mathbf{Y}_2]) = \|\mathbf{Y}_2 \mathbf{Q}^* - \mathbf{Y}_1\|_F$;*

2. *if $\mathbf{Y}_1^\top \mathbf{Y}_2$ is nonsingular, then $\mathbf{Q}^*$ is unique and the Riemannian logarithm $\log_{[\mathbf{Y}_1]}[\mathbf{Y}_2]$ is uniquely defined and its horizontal lift at $\mathbf{Y}_1$ is given by $\overline{\log_{[\mathbf{Y}_1]}[\mathbf{Y}_2]} = \mathbf{Y}_2 \mathbf{Q}^* - \mathbf{Y}_1$; moreover, the unique minimizing geodesic from $[\mathbf{Y}_1]$ to $[\mathbf{Y}_2]$ is $[\mathbf{Y}_1 + t(\mathbf{Y}_2 \mathbf{Q}^* - \mathbf{Y}_1)]$ for $t \in [0, 1]$.*

**Lemma H.2** (Lemma 12 in (Luo & García Trillos, 2022)). *For any $\mathbf{Y}_1, \mathbf{Y}_2 \in \mathbb{R}_*^{n \times r}$, we have*

$$d^2([\mathbf{Y}_1], [\mathbf{Y}_2]) \leqslant \frac{1}{2(\sqrt{2}-1)\sigma_r^2(\mathbf{Y}_2)} \|\mathbf{Y}_1 \mathbf{Y}_1^\top - \mathbf{Y}_2 \mathbf{Y}_2^\top\|_F^2 \tag{H.3}$$

*and*

$$\left\|(\mathbf{Y}_1 - \mathbf{Y}_2 \mathbf{Q})(\mathbf{Y}_1 - \mathbf{Y}_2 \mathbf{Q})^\top\right\|_F^2 \leqslant 2 \|\mathbf{Y}_1 \mathbf{Y}_1^\top - \mathbf{Y}_2 \mathbf{Y}_2^\top\|_F^2, \tag{H.4}$$

*where $\mathbf{Q} = \arg\min_{\mathbf{O} \in \mathbb{O}_r} \|\mathbf{Y}_1 - \mathbf{Y}_2 \mathbf{O}\|_F$.*

*In addition, for any $\mathbf{Y}_1, \mathbf{Y}_2 \in \mathbb{R}_*^{n \times r}$ obeying $d([\mathbf{Y}_1], [\mathbf{Y}_2]) \leqslant \frac{1}{3}\sigma_r(\mathbf{Y}_2)$, we have*

$$\|\mathbf{Y}_1 \mathbf{Y}_1^\top - \mathbf{Y}_2 \mathbf{Y}_2^\top\|_F \leqslant \frac{7}{3} \|\mathbf{Y}_2\| d([\mathbf{Y}_1], [\mathbf{Y}_2]) \tag{H.5}$$

Given any $\mathbf{Y} \in \mathbb{R}_*^{n \times r}$ and $x > 0$, let $B_x([\mathbf{Y}]) \stackrel{\text{def}}{=} \{[\mathbf{Y}_1] : d([\mathbf{Y}_1], [\mathbf{Y}]) < x\}$ be the geodesic ball centered at $[\mathbf{Y}]$ with radius $x$. For any Riemannian manifold, there exists a convex geodesic ball at every point (Chapter 3.4, (Do Carmo & Flaherty Francis, 1992)). The next result quantifies the convexity radius around a point $[\mathbf{Y}]$ in the manifold $\mathcal{N}_{r+}^n$.

**Lemma H.3** (Theorem 2, (Luo & García Trillos, 2022)). *Given any $\mathbf{Y} \in \mathbb{R}_*^{n \times r}$, the geodesic ball centered at $[\mathbf{Y}]$ with radius $x \leqslant r_{\mathbf{Y}} \stackrel{\text{def}}{=} \sigma_r(\mathbf{Y})/3$ is geodesically convex. In fact, for any two points $[\mathbf{Y}_1], [\mathbf{Y}_2] \in B_x([\mathbf{Y}])$, there is a unique shortest geodesic joining them, which is entirely contained in $B_x([\mathbf{Y}])$.*

Finally, we provide some useful inequalities.

**Lemma H.4** (Proposition 2 in (Luo et al., 2021)). *Let $\mathbf{Y} \in \mathbb{R}_*^{n \times r}$, and let $\mathbf{X} = \mathbf{YY}^\top$. Then $2\sigma_r^2(\mathbf{Y}) \|\theta_{\mathbf{Y}}\|_F^2 \leqslant \|\mathbf{Y}\theta_{\mathbf{Y}}^\top + \theta_{\mathbf{Y}}\mathbf{Y}^\top\|_F^2 \leqslant 4\sigma_1^2(\mathbf{Y}) \|\theta_{\mathbf{Y}}\|_F^2$ holds for all $\theta_{\mathbf{Y}} \in \mathcal{H}_{\mathbf{Y}} \overline{\mathcal{M}}_{r+}^q$.*

**Lemma H.5.** *For $\mathbf{A} \in \mathbb{R}^{m \times n}$, $\mathbf{B} \in \mathbb{R}^{n \times n}$ where $\mathbf{B}$ is positive semi-definite, we have*

$$\|\mathbf{A}\|_F \cdot \sigma_n(\mathbf{B}) \leq \|\mathbf{AB}\|_F \leq \|\mathbf{A}\|_F \cdot \sigma_1(\mathbf{B}) \tag{H.6}$$

*Proof.* When $m = 1$, this statement is direct by the definition of the Frobenius norm. When $m > 1$, we denote $\mathbf{A}_i$ to be the $i^{\text{th}}$ row of $\mathbf{A}$, and then

$$\|\mathbf{AB}\|_F^2 = \sum_{i=1}^m \|\mathbf{A}_i \mathbf{B}\|_F^2 \leq \sum_{i=1}^m \|\mathbf{A}_i\|_F \cdot \sigma_1(\mathbf{B}) = \|\mathbf{A}\|_F \cdot \sigma_1(\mathbf{B})$$

Similarly,

$$\|\mathbf{A}\mathbf{B}\|_{\mathrm{F}}^2 = \sum_{i=1}^{m} \|\mathbf{A}_i \mathbf{B}\|_{\mathrm{F}}^2 \geq \sum_{i=1}^{m} \|\mathbf{A}_i\|_{\mathrm{F}} \cdot \sigma_n(\mathbf{B}) = \|\mathbf{A}\|_{\mathrm{F}} \cdot \sigma_n(\mathbf{B})$$

$\square$

### H.3 Proof of Results

In this section, we provide the proofs for Theorems 3.1, 3.2, 3.3, and 3.4.

**Theorem 3.1** (Local Geodesic Strong Convexity and Smoothness of Equation 3.1). *Suppose* $0 \leqslant \mu \leqslant \kappa^*/3$. *Given that Assumption 3.1 holds, for any* $\mathbf{Y} \in \mathcal{R}_1$ *defined in Equation 3.3.*

$$\sigma_{\min}(\overline{\mathrm{Hess}\, H([\mathbf{Y}])}) \geqslant \left(2\left(1-\mu/\kappa^*\right)^2 - (14/3)\mu\right)\sigma_r(\mathcal{A}_{\mathbf{n}}) - 2\sigma_{r+1}(\mathcal{A}_{\mathbf{n}}),$$

$$\sigma_{\max}(\overline{\mathrm{Hess}\, H([\mathbf{Y}])}) \leqslant 4\left(\sigma_1(\mathbf{Y}^*) + \mu\sigma_r(\mathbf{Y}^*)/\kappa^*\right)^2 + 14\mu\sigma_r^2(\mathbf{Y}^*)/3$$

*In particular, if* $\mu$ *is further chosen such that* $\left(2\left(1-\mu/\kappa^*\right)^2 - (14/3)\mu\right)\sigma_r(\mathcal{A}_{\mathbf{n}}) - 2\sigma_{r+1}(\mathcal{A}_{\mathbf{n}}) > 0$*, we have* $H([\mathbf{Y}])$ *is geodesically strongly convex and smooth in* $\mathcal{R}_1$.

*Proof.* Denote by $\mathbf{Q}$ the best orthogonal matrix that aligns $\mathbf{Y}$ and $\mathbf{Y}^*$. Then by the assumption on $\mathbf{Y} \in \mathcal{R}_1$ as defined in Equation H.2, we have

$$\|\mathbf{Y} - \mathbf{Y}^*\mathbf{Q}\| \leqslant \|\mathbf{Y} - \mathbf{Y}^*\mathbf{Q}\|_{\mathrm{F}} = d([\mathbf{Y}], [\mathbf{Y}^*]) \leqslant \mu\sigma_r(\mathbf{Y}^*)/\kappa^*. \tag{H.7}$$

Thus

$$\sigma_r(\mathbf{Y}) = \sigma_r(\mathbf{Y} - \mathbf{Y}^*\mathbf{Q} + \mathbf{Y}^*\mathbf{Q}) \geqslant \sigma_r(\mathbf{Y}^*) - \|\mathbf{Y} - \mathbf{Y}^*\mathbf{Q}\| \overset{\mathrm{Equation\ H.7}}{\geqslant} (1 - \mu/\kappa^*)\sigma_r(\mathbf{Y}^*)$$

$$\sigma_1(\mathbf{Y}) = \sigma_1(\mathbf{Y} - \mathbf{Y}^*\mathbf{Q} + \mathbf{Y}^*\mathbf{Q}) \leqslant \sigma_1(\mathbf{Y}^*) + \|\mathbf{Y} - \mathbf{Y}^*\mathbf{Q}\| \overset{\mathrm{Equation\ H.7}}{\leqslant} \sigma_1(\mathbf{Y}^*) + \mu\sigma_r(\mathbf{Y}^*)/\kappa^* \tag{H.8}$$

where the first inequalities follow from Weyl's theorem (Stewart, 1998). Then,

$$
\begin{aligned}
\overline{\mathrm{Hess}\, H([\mathbf{Y}])}\,[\theta_{\mathbf{Y}}, \theta_{\mathbf{Y}}] &= \left\|\mathbf{Y}\theta_{\mathbf{Y}}^{\top} + \theta_{\mathbf{Y}}\mathbf{Y}^{\top}\right\|_{\mathrm{F}}^2 + 2\left\langle \mathbf{Y}\mathbf{Y}^{\top} - \mathcal{A}_{\mathbf{n}}, \theta_{\mathbf{Y}}\theta_{\mathbf{Y}}^{\top}\right\rangle && \text{[Equation H.1]}\\
&\geqslant 2\sigma_r^2(\mathbf{Y})\|\theta_{\mathbf{Y}}\|_{\mathrm{F}}^2 + 2\left\langle \mathbf{Y}\mathbf{Y}^{\top} - \mathcal{A}_{\mathbf{n}}, \theta_{\mathbf{Y}}\theta_{\mathbf{Y}}^{\top}\right\rangle && \text{[ Lemma H.4]}\\
&= 2\sigma_r^2(\mathbf{Y})\|\theta_{\mathbf{Y}}\|_{\mathrm{F}}^2 + 2\left\langle \mathbf{Y}\mathbf{Y}^{\top}, \theta_{\mathbf{Y}}\theta_{\mathbf{Y}}^{\top}\right\rangle - 2\left\langle \mathbf{Y}^*\mathbf{Y}^{*\top}, \theta_{\mathbf{Y}}\theta_{\mathbf{Y}}^{\top}\right\rangle\\
&\quad - 2\left\langle \mathbf{Z}\mathbf{Z}^{\top}, \theta_{\mathbf{Y}}\theta_{\mathbf{Y}}^{\top}\right\rangle && [\mathcal{A}_{\mathbf{n}} = \mathbf{Y}^*\mathbf{Y}^{*\top} + \mathbf{Z}\mathbf{Z}^{\top}]\\
&\geqslant 2\sigma_r^2(\mathbf{Y})\|\theta_{\mathbf{Y}}\|_{\mathrm{F}}^2 - 2\left\|\mathbf{Y}\mathbf{Y}^{\top} - \mathbf{Y}^*\mathbf{Y}^{*\top}\right\|\left\|\theta_{\mathbf{Y}}\theta_{\mathbf{Y}}^{\top}\right\|_{\mathrm{F}}\\
&\quad - 2\|\mathbf{Z}\mathbf{Z}^{\top}\|\|\theta_{\mathbf{Y}}\theta_{\mathbf{Y}}^{\top}\|_{\mathrm{F}} && [\langle A, B\rangle \leq \|A\|\|B\|_{\mathrm{F}}]\\
&\geqslant 2\sigma_r^2(\mathbf{Y})\|\theta_{\mathbf{Y}}\|_{\mathrm{F}}^2 - 2\left\|\mathbf{Y}\mathbf{Y}^{\top} - \mathbf{Y}^*\mathbf{Y}^{*\top}\right\|\|\theta_{\mathbf{Y}}\|_{\mathrm{F}}^2\\
&\quad - 2\|\mathbf{Z}\mathbf{Z}^{\top}\|\|\theta_{\mathbf{Y}}\|_{\mathrm{F}}^2 && [\|\theta_{\mathbf{Y}}\theta_{\mathbf{Y}}^{\top}\|_{\mathrm{F}} = \|\theta_{\mathbf{Y}}\|_{\mathrm{F}}^2]\\
&\geqslant 2\left(1 - \frac{\mu}{\kappa^*}\right)^2\sigma_r^2(\mathbf{Y}^*)\|\theta_{\mathbf{Y}}\|_{\mathrm{F}}^2 - 2\|\mathbf{Z}\mathbf{Z}^{\top}\|\|\theta_{\mathbf{Y}}\|_{\mathrm{F}}^2\\
&\quad - 2\left\|\mathbf{Y}\mathbf{Y}^{\top} - \mathbf{Y}^*\mathbf{Y}^{*\top}\right\|\|\theta_{\mathbf{Y}}\|_{\mathrm{F}}^2 && \text{[Equation H.8]}\\
&\geqslant 2\left(1 - \frac{\mu}{\kappa^*}\right)^2\sigma_r^2(\mathbf{Y}^*)\|\theta_{\mathbf{Y}}\|_{\mathrm{F}}^2 - 2\|\mathbf{Z}\mathbf{Z}^{\top}\|\|\theta_{\mathbf{Y}}\|_{\mathrm{F}}^2\\
&\quad - 2 \cdot \frac{7}{3}\|\mathbf{Y}^*\|\frac{\mu\sigma_r(\mathbf{Y}^*)}{\kappa^*}\|\theta_{\mathbf{Y}}\|_{\mathrm{F}}^2 && [\text{Lemma H.2}, \mathbf{Y} \in \mathcal{R}_1]\\
&= 2\left(1 - \frac{\mu}{\kappa^*}\right)^2\sigma_r^2(\mathbf{Y}^*)\|\theta_{\mathbf{Y}}\|_{\mathrm{F}}^2 - 2 \cdot \frac{7}{3}\|\mathbf{Y}^*\|\frac{\mu\sigma_r(\mathbf{Y}^*)}{\kappa^*}\|\theta_{\mathbf{Y}}\|_{\mathrm{F}}^2\\
&\quad - 2\sigma_{r+1}(\mathcal{A}_{\mathbf{n}})\|\theta_{\mathbf{Y}}\|_{\mathrm{F}}^2 && [\|\mathbf{Z}\mathbf{Z}^{\top}\| = \sigma_{r+1}(\mathcal{A}_{\mathbf{n}})]\\
&= \left(\left(2\left(1 - \frac{\mu}{\kappa^*}\right)^2 - \frac{14}{3}\mu\right)\sigma_r(\mathcal{A}_{\mathbf{n}}) - 2\sigma_{r+1}(\mathcal{A}_{\mathbf{n}})\right)\|\theta_{\mathbf{Y}}\|_{\mathrm{F}}^2 && \left[\kappa^* = \frac{\|\mathbf{Y}^*\|}{\sigma_r(\mathbf{Y}^*)}\right]
\end{aligned}
$$

Likewise,

$$\overline{\text{Hess } H([\mathbf{Y}])} [\theta_\mathbf{Y}, \theta_\mathbf{Y}] = \left\| \mathbf{Y}\theta_\mathbf{Y}^\top + \theta_\mathbf{Y}\mathbf{Y}^\top \right\|_\text{F}^2 + 2\left\langle \mathbf{YY}^\top - \mathcal{A}_\mathbf{n}, \theta_\mathbf{Y}\theta_\mathbf{Y}^\top \right\rangle \qquad [\text{Equation H.1}]$$

$$\leqslant 4\sigma_1^2(\mathbf{Y}) \left\| \theta_\mathbf{Y} \right\|_\text{F}^2 + 2\left\langle \mathbf{YY}^\top - \mathcal{A}_\mathbf{n}, \theta_\mathbf{Y}\theta_\mathbf{Y}^\top \right\rangle \qquad [\text{ Lemma } H.4]$$

$$\leqslant 4\sigma_1^2(\mathbf{Y}) \left\| \theta_\mathbf{Y} \right\|_\text{F}^2 + 2\left\langle \mathbf{YY}^\top - \mathbf{Y}^*\mathbf{Y}^{*\top}, \theta_\mathbf{Y}\theta_\mathbf{Y}^\top \right\rangle \qquad [\mathcal{A}_\mathbf{n} - \mathbf{Y}^*\mathbf{Y}^{*\top} \text{ is PSD}]$$

$$\leqslant 4\sigma_1^2(\mathbf{Y}) \left\| \theta_\mathbf{Y} \right\|_\text{F}^2 + 2\left\| \mathbf{YY}^\top - \mathbf{Y}^*\mathbf{Y}^{*\top} \right\| \left\| \theta_\mathbf{Y} \right\|_\text{F}^2$$

$$\leqslant 4\sigma_1^2(\mathbf{Y}) \left\| \theta_\mathbf{Y} \right\|_\text{F}^2 + 2\left\| \mathbf{YY}^\top - \mathbf{Y}^*\mathbf{Y}^{*\top} \right\|_\text{F} \left\| \theta_\mathbf{Y} \right\|_\text{F}^2$$

$$\leqslant 4\left( \sigma_1\left(\mathbf{Y}^*\right) + \frac{\mu\sigma_r\left(\mathbf{Y}^*\right)}{\kappa^*} \right)^2 \left\| \theta_\mathbf{Y} \right\|_\text{F}^2 + 2\left\| \mathbf{YY}^\top - \mathbf{Y}^*\mathbf{Y}^{*\top} \right\|_\text{F} \left\| \theta_\mathbf{Y} \right\|_\text{F}^2 \qquad [\text{Equation H.8}]$$

$$\leqslant \left( 4\left( \sigma_1\left(\mathbf{Y}^*\right) + \frac{\mu\sigma_r\left(\mathbf{Y}^*\right)}{\kappa^*} \right)^2 + \frac{14}{3}\mu\sigma_r^2\left(\mathbf{Y}^*\right) \right) \left\| \theta_\mathbf{Y} \right\|_\text{F}^2 \qquad [\text{Lemma H.2}]$$

From the above we conclude that when $\mu$ is chosen such that

$$\left( 2\left( 1 - \frac{\mu}{\kappa^*} \right)^2 - \frac{14}{3}\mu \right) \sigma_r\left(\mathcal{A}_\mathbf{n}\right) - 2\sigma_{r+1}(\mathcal{A}_\mathbf{n}) > 0,$$

we have $H([\mathbf{Y}])$ in Equation 3.1 is geodesically strongly convex and smooth in $\mathcal{R}_1$ as $\mathcal{R}_1$ is a geodesically convex set by (Luo & García Trillos, 2022). Note that this is equivalent to

$$\left( \left( 1 - \frac{\mu}{\kappa^*} \right)^2 - \frac{7}{3}\mu \right) > \frac{\sigma_{r+1}(\mathcal{A}_\mathbf{n})}{\sigma_r(\mathcal{A}_\mathbf{n})}.$$

Then note as $\mu \to 0$, the left hand side approaches 1 and the inequality becomes true as $\sigma_r(\mathcal{A}_\mathbf{n}) > \sigma_{r+1}(\mathcal{A}_\mathbf{n})$. $\qquad\square$

**Remark H.2.** *Compared with the bound in Theorem 8 of (Luo & García Trillos, 2022), the smoothness and geodesically strongly convexity are as follows,*

$$\sigma_{\min}(\overline{\text{Hess } H([\mathbf{Y}])}) \geqslant \left( 2\left( 1 - \mu/\kappa^* \right)^2 - (14/3)\mu \right) \sigma_r^2\left(\mathbf{Y}^*\right),$$

$$\sigma_{\max}(\overline{\text{Hess } H([\mathbf{Y}])}) \leqslant 4\left( \sigma_1\left(\mathbf{Y}^*\right) + \mu\sigma_r\left(\mathbf{Y}^*\right)/\kappa^* \right)^2 + 14\mu\sigma_r^2\left(\mathbf{Y}^*\right)/3.$$

*There is an extra term $-2\sigma_{r+1}(\mathcal{A}_\mathbf{n})$ in our lower bound of the strong convexity because even if $d([\mathbf{Y}], [\mathbf{Y}^*])$ is small, $\mathcal{A}_\mathbf{n} - \mathbf{YY}^\top$ is not close to $\mathbf{0}$, which leads to the extra error term.*

In the next three theorems, we show that for $\mathbf{Y} \notin \mathcal{R}_1$, either the Riemannian Hessian evaluated at $\mathbf{Y}$ has a large negative eigenvalue, or the norm of the Riemannian gradient is large. Let $\mathbf{Y} = \mathbf{UDV}^\top$, $\mathbf{Y}^* = \mathbf{U}^*\mathbf{\Sigma}^{*1/2}$.

**Theorem 3.2** (FOSP of Equation 3.1). *Let $\overline{\mathbf{U}}\mathbf{\Sigma}\overline{\mathbf{U}}^\top$ be $\mathcal{A}_\mathbf{n}$'s SVD factorization, and let $\mathbf{\Lambda} = \mathbf{\Sigma}^{1/2}$. Then for any $S$ subset of $[n]$, we have that $\left[\overline{\mathbf{U}}_S \mathbf{\Lambda}_S\right]$ is a Riemannian FOSPs of Equation 3.1. Further, these are the only Riemannian FOSPs.*

*Proof.* From Equation H.1, the gradient can be written down as,

$$\overline{\text{grad } H([\mathbf{Y}])} = 2\left( \mathbf{YY}^\top - \mathcal{A}_\mathbf{n} \right)\mathbf{Y} = 2\left( \mathbf{UDV}^\top(\mathbf{UDV}^\top)^\top - \mathcal{A}_\mathbf{n} \right)\mathbf{UDV}^\top$$

$$= 2\left( \mathbf{UD}^3\mathbf{V}^\top - \mathcal{A}_\mathbf{n}\mathbf{UDV}^\top \right)$$

Therefore, whenever $\overline{\text{grad } H([\mathbf{Y}])} = \mathbf{0}$, we have $\mathbf{UD}^3\mathbf{V}^\top - \mathcal{A}_\mathbf{n}\mathbf{UDV}^\top = \mathbf{0}$. Since both $\mathbf{V}$ and $\mathbf{D}$ are of full rank, the condition is equivalent to

$$\mathbf{UD}^2 - \mathcal{A}_\mathbf{n}\mathbf{U} = \mathbf{0} \qquad (\text{H.9})$$

Since $\mathbf{D}^2$ is also a diagonal matrix, to satisfy Equation H.9, the columns of $\mathbf{U}$ have to be the eigenvectors of $\mathcal{A}_\mathbf{n}$, and the diagonal of $\mathbf{D}^2$ has to be the eigenvalues of $\mathcal{A}_\mathbf{n}$. This completes the proof. $\quad\square$

Before we can prove the next main result, Theorem 3.3, we need to discuss some of the assumptions. Specifically, we want to quantify the statement $\alpha$ is sufficiently small.

**Assumption H.1** (Parameters Settings). *Denote $e_1, e_2$ and $e_3$ to be some error terms.*

$$e_1 \overset{def}{=} \frac{\alpha\mu\sigma_r^3\left(\mathbf{Y}^*\right)}{2\sqrt{2}\kappa^*\sigma_{r+1}(\mathbf{\Lambda})}, \quad e_2 = \frac{e_1}{\sqrt{2}}, \quad \textit{and} \quad e_3 = e_2 \cdot \sigma_{r+1}(\mathbf{\Lambda})$$

*Note that $e_1, e_2, e_3 \to 0$ and $\alpha \to 0$. Hence, pick $\alpha$ small enough such that the following are true.*

1. $\sigma_r^2(\mathbf{\Lambda}) - 2e_1 - \sigma_{r+1}^2(\mathbf{\Lambda}) > 0.$

2. $\sigma_r^2(\mathbf{\Lambda})\left(1 - \frac{e_1^2}{\left|\sigma_r^2(\mathbf{\Lambda}) - e_1 - \sigma_{r+1}^2(\mathbf{\Lambda})\right|^2}\right) - e_1 - \sigma_{r+1}^2(\mathbf{\Lambda}) > 0.$

3. $(\alpha - 2(\sqrt{2} - 1))\sigma_r^2\left(\mathbf{Y}^*\right) + 6\frac{\alpha^2\sigma_r^4(\mathbf{Y}^*)\sigma_{r+1}^2(\mathbf{\Lambda})/16}{\left|\sigma_r^2(\mathbf{\Lambda}) - e_2 - \sigma_{r+1}^2(\mathbf{\Lambda})\right|^2} < 0.$

Note that for the first two, we have that as $\alpha \to 0$. They converge to $\sigma_r^2(\mathbf{\Lambda}) - \sigma_{r+1}^2(\mathbf{\Lambda})$ which is positive due to the eigengap assumption. For the last condition, we have that as $\alpha \to 0$, it converges to $-2(\sqrt{2}-1)\sigma_r^2\left(\mathbf{Y}^*\right)$ which is negative.

Hence, notice that this assumption is only related to the eigengap assumption $\sigma_r(\mathbf{\Lambda})$ and $\sigma_{r+1}(\mathbf{\Lambda})$ in Assumption 3.1. As soon as $\alpha$ is small enough, Assumption H.1 is satisfied.

**Theorem H.1** (Region with Negative Eigenvalue in the Riemannian Hessian of Equation 1.5 (formal Theorem 3.3)). *Assume that Assumption 3.1 holds. Given any $\mathbf{Y} \in \mathbb{R}_*^{n\times r}$, let $\theta_{\mathbf{Y}}^1 = [\mathbf{0},\mathbf{0},\ldots,\mathbf{0},\mathbf{a},\mathbf{0},\ldots,\mathbf{0}]\mathbf{V}^\top$ where $\mathbf{a}$ such that*

$$\mathbf{a} = \underset{\mathbf{a}:\mathbf{Y}^\top\mathbf{a}=\mathbf{0}}{\arg\max} \frac{\mathbf{a}^\top\mathcal{A}_\mathbf{n}\mathbf{a}}{\|\mathbf{a}\|^2} \tag{H.10}$$

*and $[\mathbf{0},\mathbf{0},\ldots,\mathbf{0},\mathbf{a},\mathbf{0},\ldots,\mathbf{0}] \in \mathbb{R}^{n\times r}$ such that the $\tilde{i}^{\text{th}}$ columns is $\mathbf{a}$ and other columns are $\mathbf{0}$ where*

$$\tilde{i} \overset{def}{=} \underset{j\in[r]}{\arg\min} \mathbf{D}_{jj}. \tag{H.11}$$

*Denote $\theta_{\mathbf{Y}}^2 = \mathbf{Y} - \mathbf{Y}^*\mathbf{Q}$, where $\mathbf{Q} \in \mathbb{O}_r$ is the best orthogonal matrix aligning $\mathbf{Y}^*$ and $\mathbf{Y}$. We choose $\theta_{\mathbf{Y}}$ to be either $\theta_{\mathbf{Y}}^1$ or $\theta_{\mathbf{Y}}^2$. Then*

$$\overline{\mathrm{Hess}\, H([\mathbf{Y}])}\,[\theta_{\mathbf{Y}},\theta_{\mathbf{Y}}] \leqslant \min\left\{-\frac{\sigma_{r+1}^2(\mathbf{\Lambda})}{2}\|\theta_{\mathbf{Y}}\|^2,\right.$$
$$-2\left(\sigma_r^2(\mathbf{\Lambda})\left(1 - \frac{e_1^2}{\left|\sigma_r^2(\mathbf{\Lambda}) - e_1 - \sigma_{r+1}^2(\mathbf{\Lambda})\right|^2}\right) - e_1 - \sigma_{r+1}^2(\mathbf{\Lambda})\right)\|\theta_{\mathbf{Y}}\|^2,$$
$$\left.\left((\alpha - 2(\sqrt{2}-1))\sigma_r^2\left(\mathbf{Y}^*\right) + 6\frac{\alpha^2\sigma_r^4\left(\mathbf{Y}^*\right)\sigma_{r+1}^2(\Lambda)/16}{\left|\sigma_r^2(\mathbf{\Lambda}) - e_2 - \sigma_{r+1}^2(\mathbf{\Lambda})\right|^2}\right)\|\theta_{\mathbf{Y}}\|_{\mathrm{F}}^2\right\}$$

*In particular, if $\alpha$ and $\mu$ satisfies Assumption H.1, we have $\overline{\mathrm{Hess}\, H([\mathbf{Y}])}$ has at least one negative eigenvalue and $\theta_{\mathbf{Y}}$ is the escaping direction.*

*Proof.* By the definition of $\mathbf{a}$, $\mathbf{a} \in \mathrm{Span}\{\overline{\mathbf{U}}_{1,\ldots,r+1}\}$. This is because the null space of $\mathbf{Y}$ has dimension $n - r$. Hence, its intersection with a dimension $r + 1$ space has a dimension of at least 1.

Using the SVD decomposition of $\mathbf{Y}$, we have, $\mathbf{U}^\top\mathbf{a} = \mathbf{0}$. Then, by using Equation H.1, we have

$$\begin{aligned}
\overline{\mathrm{Hess}\, H([\mathbf{Y}])}\left[\theta_{\mathbf{Y}}^1,\theta_{\mathbf{Y}}^1\right] &= \left\|\mathbf{Y}(\theta_{\mathbf{Y}}^1)^\top + \theta_{\mathbf{Y}}^1\mathbf{Y}^\top\right\|_{\mathrm{F}}^2 + 2\left\langle\mathbf{Y}\mathbf{Y}^\top - \mathcal{A}_\mathbf{n}, \theta_{\mathbf{Y}}^1(\theta_{\mathbf{Y}}^1)^\top\right\rangle && \text{[Equation H.1]}\\
&= \left\|\mathbf{Y}(\theta_{\mathbf{Y}}^1)^\top + \theta_{\mathbf{Y}}^1\mathbf{Y}^\top\right\|_{\mathrm{F}}^2 - 2\left\langle\mathcal{A}_\mathbf{n}, \theta_{\mathbf{Y}}^1(\theta_{\mathbf{Y}}^1)^\top\right\rangle && [\mathbf{Y}^\top\mathbf{a} = \mathbf{0}]\\
&= 2\langle\mathbf{Y}^\top\mathbf{Y}, (\theta_{\mathbf{Y}}^1)^\top\theta_{\mathbf{Y}}^1\rangle + 2\langle\mathbf{Y}(\theta_{\mathbf{Y}}^1)^\top, \theta_{\mathbf{Y}}^1\mathbf{Y}^\top\rangle - 2\left\langle\mathcal{A}_\mathbf{n}, \theta_{\mathbf{Y}}^1(\theta_{\mathbf{Y}}^1)^\top\right\rangle && [\|A\|_{\mathrm{F}}^2 = \langle A, A\rangle]\\
&= 2\langle\mathbf{Y}^\top\mathbf{Y}, (\theta_{\mathbf{Y}}^1)^\top\theta_{\mathbf{Y}}^1\rangle - 2\left\langle\mathcal{A}_\mathbf{n}, \theta_{\mathbf{Y}}^1(\theta_{\mathbf{Y}}^1)^\top\right\rangle && [\mathbf{Y}^\top\mathbf{a} = \mathbf{0}]\\
&= 2\langle\mathbf{V}\mathbf{D}^2\mathbf{V}^\top, (\theta_{\mathbf{Y}}^1)^\top\theta_{\mathbf{Y}}^1\rangle - 2\left\langle\mathcal{A}_\mathbf{n}, \theta_{\mathbf{Y}}^1(\theta_{\mathbf{Y}}^1)^\top\right\rangle\\
&= 2\mathbf{D}_{\tilde{i}\tilde{i}}^2\|\mathbf{a}\|^2 - 2\mathbf{a}^\top\mathcal{A}_\mathbf{n}\mathbf{a}
\end{aligned}$$

where the last equality comes from the definition of $a$ and the fact that the $\mathbf{V}^\top\mathbf{V} = \mathbf{I}$ in $\theta_{\mathbf{Y}}^1(\theta_{\mathbf{Y}}^1)^\top$. Recall $\tilde{i} = \arg\min \mathbf{D}_{ii}$, then

$$\overline{\mathrm{Hess}\,H([\mathbf{Y}])}\left[\theta_{\mathbf{Y}}^1, \theta_{\mathbf{Y}}^1\right] = 2\min_i \mathbf{D}_{ii}^2\|a\|^2 - 2a^\top\mathcal{A}_{\mathbf{n}}a \tag{H.12}$$

In the following, we separate the proof into three regimes of $\min_i \mathbf{D}_{ii}^2$, corresponding to different escape directions.

**Case 1:** $\left(\textbf{When } \min_i \mathbf{D}_{ii}^2 < \frac{\sigma_{r+1}^2(\mathbf{\Lambda})}{2}\right)$. For this case we must have that

$$\overline{\mathrm{Hess}\,H([\mathbf{Y}])}\left[\theta_{\mathbf{Y}}^1, \theta_{\mathbf{Y}}^1\right] \leq -\frac{\sigma_{r+1}^2(\mathbf{\Lambda})}{2}\|\theta_{\mathbf{Y}}^1\|^2.$$

This is because $a^\top\mathcal{A}_{\mathbf{n}}a \geq \sigma_{r+1}^2(\mathbf{\Lambda})\|a\|^2$ and $\|a\| = \|\theta_{\mathbf{Y}}^1\|$.

**Case 2:** $\left(\textbf{When } \min_i \mathbf{D}_{ii}^2 \geq \frac{\sigma_{r+1}^2(\mathbf{\Lambda})}{2}\right)$.

From the proof of Theorem 3.2, the gradient condition of $\mathcal{R}_2$ can be written as

$$\begin{aligned}
\alpha\mu\sigma_r^3(\mathbf{Y}^*)/(4\kappa^*) &\geq \|\overline{\mathrm{grad}\,H([\mathbf{Y}])}\|_{\mathrm{F}} & [\mathbf{Y} \in \mathcal{R}_2]\\
&= \|2\left(\mathbf{UD}^3\mathbf{V}^\top - \mathcal{A}_{\mathbf{n}}\mathbf{UDV}^\top\right)\|_{\mathrm{F}} & [\text{Equation H.1}]\\
&= \|2\left(\mathbf{UD}^2 - \mathcal{A}_{\mathbf{n}}\mathbf{U}\right)\mathbf{D}\|_{\mathrm{F}}
\end{aligned}$$

Assume $\mathbf{U} = \overline{\mathbf{U}}C$ where $C \in \mathbb{R}^{n\times r}$. Since $\mathbf{U}^\top\mathbf{U} = \mathbf{I}_r$ and $\overline{\mathbf{U}}^\top\overline{\mathbf{U}} = \mathbf{I}_n$, we have $C^\top C = \mathbf{I}_r$. Furthermore,

$$\begin{aligned}
\|2\left(\mathbf{UD}^2 - \mathcal{A}_{\mathbf{n}}\mathbf{U}\right)\mathbf{D}\|_{\mathrm{F}} &= \|2\left(\overline{\mathbf{U}}C\mathbf{D}^2 - \mathcal{A}_{\mathbf{n}}\overline{\mathbf{U}}C\right)\mathbf{D}\|_{\mathrm{F}} & [\mathbf{U} = \overline{\mathbf{U}}C]\\
&= \|2\left(\overline{\mathbf{U}}C\mathbf{D}^2 - \overline{\mathbf{U}}\mathbf{\Sigma}C\right)\mathbf{D}\|_{\mathrm{F}} & [\mathcal{A}_{\mathbf{n}} = \overline{\mathbf{U}}\mathbf{\Sigma}\overline{\mathbf{U}}^\top]\\
&= 2\|\left(C\mathbf{D}^2 - \mathbf{\Sigma}C\right)\mathbf{D}\|_{\mathrm{F}}.
\end{aligned}$$

Here the third equality follows from $\overline{\mathbf{U}}^\top\overline{\mathbf{U}} = \mathbf{I}_n$. By a direct computation, the $i^{\text{th}}$ column of $\left(C\mathbf{D}^2 - \mathbf{\Sigma}C\right)\mathbf{D}$ is $\mathbf{D}_{ii}^3 C_i - \mathbf{D}_{ii}\mathbf{\Sigma}C_i$. Therefore, the gradient condition of $\mathcal{R}_2$ can be written as

$$\sum_{i,j}\left(\mathbf{D}_{ii}^3 C_{ji} - \mathbf{D}_{ii}\mathbf{\Sigma}_{jj}C_{ji}\right)^2 \leq \alpha^2\mu^2\sigma_r^6(\mathbf{Y}^*)/(4\kappa^*)^2 \tag{H.13}$$

We fix $i$ in the left hand side of Equation H.13, we have

$$\sum_j\left(\mathbf{D}_{ii}^2 - \mathbf{\Sigma}_{jj}\right)^2\mathbf{D}_{ii}^2 C_{ji}^2 \leq \alpha^2\mu^2\sigma_r^6(\mathbf{Y}^*)/(4\kappa^*)^2 \tag{H.14}$$

where $\sum_j C_{ji}^2 = 1$. From $\mathbf{D}_{ii}^2 \geq \frac{\sigma_{r+1}^2(\mathbf{\Lambda})}{2}$, we must have

$$\min_j |\mathbf{D}_{ii}^2 - \mathbf{\Sigma}_{jj}|^2 \leq \sum_j\left(\mathbf{D}_{ii}^2 - \mathbf{\Sigma}_{jj}\right)^2 C_{ji}^2 \leq \frac{\alpha^2\mu^2\sigma_r^6(\mathbf{Y}^*)}{(4\kappa^*)^2\frac{\sigma_{r+1}^2(\mathbf{\Lambda})}{2}}. \tag{H.15}$$

We use Equation H.14 for the second inequality. Equation H.15 is important in the proof because this essentially guarantees that $\mathbf{D}_{ii}^2$ must be close to some $\mathbf{\Sigma}_{jj}$. This is because $\frac{\alpha^2\mu^2\sigma_r^6(\mathbf{Y}^*)}{(4\kappa^*)^2\frac{\sigma_{r+1}^2(\mathbf{\Lambda})}{2}}$ is guaranteed small according to Assumption H.1.

We decompose $C_{\tilde{i}}$ into $\xi^1 + \xi^2$ where $\xi_j^1 = 0$ for all $j \geq r+1$ and $\xi_j^2 = 0$ for all $j \in [r]$. Since $\langle \xi^1, \xi^2\rangle = 0$ and $C^\top C = \mathbf{I}$,

$$\|\xi^1\|^2 + \|\xi^2\|^2 = 1 \tag{H.16}$$

In the following, we divide all the cases into different regimes based on which of the eigenvalues of $\mathbf{\Lambda}$ is close to $\mathbf{D}_{\tilde{i}\tilde{i}}$.

**Case 2.1:** $\big($When $\frac{\sigma_{r+1}^2(\mathbf{\Lambda})}{2} \leq \mathbf{D}_{\tilde{i}\tilde{i}}^2 \leq \frac{\alpha\mu\sigma_r^3(\mathbf{Y}^*)}{2\sqrt{2}\kappa^*\sigma_{r+1}(\mathbf{\Lambda})} + \sigma_{r+1}^2(\mathbf{\Lambda})\big)$.

Notice that the first assumption in Assumption H.1 essentially guarantees a small $e_1 = \frac{\alpha\mu\sigma_r^3(\mathbf{Y}^*)}{2\sqrt{2}\kappa^*\sigma_{r+1}(\mathbf{\Lambda})}$.

Hence, we have

$$\alpha^2\mu^2\sigma_r^6(\mathbf{Y}^*)/(4\kappa^*)^2 \geq \sum_j \big(\mathbf{D}_{\tilde{i}\tilde{i}}^2 - \mathbf{\Sigma}_{jj}\big)^2 \mathbf{D}_{\tilde{i}\tilde{i}}^2 C_{j\tilde{i}}^2 \qquad \text{[Equation H.13]}$$

$$\geq \sum_{j \leq r} \big|\sigma_j^2(\mathbf{\Lambda}) - \mathbf{D}_{\tilde{i}\tilde{i}}^2\big|^2 \cdot \mathbf{D}_{\tilde{i}\tilde{i}}^2 \cdot C_{j\tilde{i}}^2$$

$$\geq \big|\sigma_r^2(\mathbf{\Lambda}) - \mathbf{D}_{\tilde{i}\tilde{i}}^2\big|^2 \cdot \mathbf{D}_{\tilde{i}\tilde{i}}^2 \cdot \|\xi^1\|^2$$

$$\geq \big|\sigma_r^2(\mathbf{\Lambda}) - e_1 - \sigma_{r+1}^2(\mathbf{\Lambda})\big|^2 \cdot \frac{\sigma_{r+1}^2(\mathbf{\Lambda})}{2} \cdot \|\xi^1\|^2.$$

Where in the last two inequalities, we use the condition $\frac{\sigma_{r+1}^2(\mathbf{\Lambda})}{2} \leq \mathbf{D}_{\tilde{i}\tilde{i}}^2 \leq e_1 + \sigma_{r+1}^2(\mathbf{\Lambda})$ and that $e_1 < (\sigma_r^2(\mathbf{\Lambda}) - \sigma_{r+1}^2(\mathbf{\Lambda}))/2$ (follows from Assumption H.1).

By reordering the inequality, we have

$$\|\xi^1\| \leq \frac{e_1}{\big|\sigma_r^2(\mathbf{\Lambda}) - e_1 - \sigma_{r+1}^2(\mathbf{\Lambda})\big|} \qquad (\text{H.17})$$

Recall that $\mathbf{Y} = \mathbf{UDV}^\top$, then $\boldsymbol{a}^\top\mathbf{Y} = \mathbf{0}$ reduces to $\boldsymbol{a}^\top\mathbf{UDV}^\top = \mathbf{0}$. Since both $\mathbf{D}, \mathbf{V} \in \mathbb{R}^{r*r}$ are full rank, then we have $\boldsymbol{a}^\top\mathbf{U} = \mathbf{0}$, in turn $\boldsymbol{a}^\top\overline{\mathbf{U}}C = \mathbf{0}$ because $\mathbf{U} = \overline{\mathbf{U}}C$. Denote $\boldsymbol{b}^\top \overset{\text{def}}{=} \boldsymbol{a}^\top\overline{\mathbf{U}}$, then

$$\max_{\boldsymbol{a}:\mathbf{Y}^\top\boldsymbol{a}=\mathbf{0}} \frac{\boldsymbol{a}^\top\mathcal{A}_\mathbf{n}\boldsymbol{a}}{\|\boldsymbol{a}\|^2} = \max_{\boldsymbol{a}:\boldsymbol{a}^\top\overline{\mathbf{U}}C=\mathbf{0}} \frac{\boldsymbol{a}^\top\mathcal{A}_\mathbf{n}\boldsymbol{a}}{\|\boldsymbol{a}\|^2}$$

$$= \max_{\boldsymbol{a}:\boldsymbol{a}^\top\overline{\mathbf{U}}C=\mathbf{0}} \frac{\boldsymbol{a}^\top\overline{\mathbf{U}}\mathbf{\Lambda}\overline{\mathbf{U}}^\top\boldsymbol{a}}{\|\boldsymbol{a}\|^2} \quad [\mathcal{A}_\mathbf{n} = \overline{\mathbf{U}}\mathbf{\Lambda}\overline{\mathbf{U}}^\top] \qquad (\text{H.18})$$

$$= \max_{\boldsymbol{b}:\boldsymbol{b}^\top C=\mathbf{0}} \frac{\boldsymbol{b}^\top\mathbf{\Lambda}\boldsymbol{b}}{\|\boldsymbol{b}\|^2} \qquad [\overline{\mathbf{U}}^\top\overline{\mathbf{U}} = \mathbf{I}]$$

Since $\boldsymbol{a} \in \text{Span}\{\overline{\mathbf{U}}_{1,\dots,r+1}\}$, we have $\boldsymbol{b}_j = 0$ for $j > r+1$. From $\boldsymbol{b}^\top C = \mathbf{0}$, we have $\boldsymbol{b}^\top C_{\tilde{i}} = 0$, which can be written as $\boldsymbol{b}^\top(\xi^1 + \xi^2) = 0$. Since there are in total $r$ constraints in $\boldsymbol{b}^\top C = \mathbf{0}$, there must exist a $\boldsymbol{b}$ satisfying the constraints $\boldsymbol{b}^\top C = \mathbf{0}$, and the norm of $\boldsymbol{b}_{r+1:n}$ is relatively small compared with the norm of $\boldsymbol{b}_{1:r}$. Specifically, denote $C_{1:r}$ to be the $1^{\text{st}}$ to $r^{\text{th}}$ rows of $C$. We consider $\boldsymbol{b}$ to be $\boldsymbol{b}^1 + \boldsymbol{b}^2$ such that $\boldsymbol{b}_i^1 = 0$ for $i > r$, and $\boldsymbol{b}_i^2 = 0$ for $i \in [r]$. We discuss two cases of $C_{1:r} \in \mathbb{R}^{r*r}$ in the following.

**Case 2.1.1: If $C_{1:r}$ is not full rank.**

In this case, there exists $\tilde{\boldsymbol{b}}^1 \in \mathbb{R}^r$ such that $\|\tilde{\boldsymbol{b}}^1\| > 0$ and $(\tilde{\boldsymbol{b}}^1)^\top C_{1:r} = \mathbf{0}$. Therefore, by denoting $\bar{\boldsymbol{b}}_{1:r} = t\tilde{\boldsymbol{b}}^1 + \boldsymbol{b}_{1:r}^1$, and $\bar{\boldsymbol{b}}_{r+1:n} = \boldsymbol{b}_{r+1:n}^2$. From the definition of $\bar{\boldsymbol{b}}$ and the fact that $\boldsymbol{b}^\top C = \mathbf{0}$, we have $\bar{\boldsymbol{b}}^\top C = \mathbf{0}$. By letting $t \to \infty$, we have

$$\max_{\boldsymbol{b}^\top C=\mathbf{0}} \frac{\boldsymbol{b}^\top\mathbf{\Lambda}\boldsymbol{b}}{\|\boldsymbol{b}\|^2} \geq \frac{\bar{\boldsymbol{b}}^\top\mathbf{\Lambda}\bar{\boldsymbol{b}}}{\|\bar{\boldsymbol{b}}\|^2} \geq \sigma_r^2(\mathbf{\Lambda}) \qquad (\text{H.19})$$

Combining Equation H.19, Equation H.12 and the Assumption that $\mathbf{D}_{\tilde{i}\tilde{i}}^2 \leq e_1 + \sigma_{r+1}^2(\mathbf{\Lambda})$, this implies,

$$\overline{\text{Hess } H([\mathbf{Y}])}\big[\theta_\mathbf{Y}^1, \theta_\mathbf{Y}^1\big] \leqslant -(\sigma_r^2(\mathbf{Y}^*) - \sigma_{r+1}(\mathbf{\Lambda}) - e_1)\|\theta_\mathbf{Y}^1\|_\text{F}^2 \qquad (\text{H.20})$$

According to Assumption H.1, this satisfies the bound in Theorem H.1 with $\theta_\mathbf{Y}^1$ being a negative escaping direction.

**Case 2.1.2 : If $C_{1:r}$ is full rank.** In this case, we denote $\boldsymbol{b}^2 = \xi^2$. Since $C_{1:r}$ is full rank, there exists $\boldsymbol{b}^1$ to have $(\boldsymbol{b}_{1:r}^1)^\top C_{1:r} = -(\boldsymbol{b}^2)^\top C$; this is because $(\boldsymbol{b}_{1:r}^1)^\top C_{1:r} = -(\xi^2)^\top C$ has in total

$r$ constraints, and there are in total $r$ parameters in $\boldsymbol{b}_{1:r}^1$. Specifically, one can choose $\boldsymbol{b}^1$ to be $\boldsymbol{b}_{1:r}^1 = -\xi^2 \boldsymbol{C}(\boldsymbol{C}_{1:r})^{-1}$ to satisfy $\boldsymbol{b}^\top \boldsymbol{C} = \boldsymbol{0}$. In addition, from the specific condition $\boldsymbol{b}^\top \boldsymbol{C}_{\tilde{i}} = \boldsymbol{0}$, we know that

$$\boldsymbol{b}^1 \cdot \xi^1 + \|\xi^2\|^2 = 0 \tag{H.21}$$

By using the Cauchy inequality, this further implies that

$$\|\boldsymbol{b}^1\| \geq \frac{\|\xi^2\|^2}{\|\xi^1\|} \tag{H.22}$$

Since we only choose a specific $\boldsymbol{b}$ such that $\boldsymbol{b}^\top \boldsymbol{C} = \boldsymbol{0}$ holds, we have

$$\begin{aligned}
\max_{\boldsymbol{b}^\top \boldsymbol{C}=\boldsymbol{0}} \frac{\boldsymbol{b}^\top \boldsymbol{\Lambda} \boldsymbol{b}}{\|\boldsymbol{b}\|^2} &\geq \frac{(\boldsymbol{b}^1 + \boldsymbol{b}^2)^\top \boldsymbol{\Lambda} (\boldsymbol{b}^1 + \boldsymbol{b}^2)}{\|\boldsymbol{b}^1 + \boldsymbol{b}^2\|^2} \\
&= \frac{(\boldsymbol{b}^1)^\top \boldsymbol{\Lambda} \boldsymbol{b}^1 + (\boldsymbol{b}^2)^\top \boldsymbol{\Lambda} \boldsymbol{b}^2}{\|\boldsymbol{b}^1\|^2 + \|\boldsymbol{b}^2\|^2} \\
&\geq \frac{(\boldsymbol{b}^1)^\top \boldsymbol{\Lambda} \boldsymbol{b}^1}{\|\boldsymbol{b}^1\|^2 + \|\xi^2\|^2} \\
&\geq \frac{\|\boldsymbol{b}^1\|^2 \cdot \sigma_r^2(\boldsymbol{\Lambda})}{\|\boldsymbol{b}^1\|^2 + \|\xi^2\|^2} \\
&\geq \frac{\frac{\|\xi^2\|^4}{\|\xi^1\|^2} \cdot \sigma_r^2(\boldsymbol{\Lambda})}{\frac{\|\xi^2\|^4}{\|\xi^1\|^2} + \|\xi^2\|^2} \\
&= \|\xi^2\|^2 \cdot \sigma_r^2(\boldsymbol{\Lambda})
\end{aligned} \tag{H.23}$$

where the first equality follows from the definition of $\boldsymbol{b}^1$ and $\boldsymbol{b}^2$; the second inequality follows from the assumption that $\boldsymbol{\Lambda}$ is PSD, and $\boldsymbol{b}^2 = \xi^2$; the third inequality follows from the fact that $\boldsymbol{b}_i^1 = 0$ for $i > r$; the fourth inequality follows from Equation H.22; the last equality follows from Equation H.16. By using Equation H.18, this can be written as

$$\max_{\boldsymbol{a}:\boldsymbol{Y}^\top \boldsymbol{a}=\boldsymbol{0}} \frac{\boldsymbol{a}^\top \mathcal{A}_{\mathbf{n}} \boldsymbol{a}}{\|\boldsymbol{a}\|^2} \geq \|\xi^2\|^2 \cdot \sigma_r^2(\boldsymbol{\Lambda}) \tag{H.24}$$

By the definition in Equation H.10 and Equation H.12, we have

$$\begin{aligned}
\overline{\operatorname{Hess} H([\mathbf{Y}])} \left[\theta_{\mathbf{Y}}^1, \theta_{\mathbf{Y}}^1\right] &= 2 \min_i \mathbf{D}_{ii}^2 \cdot \|\boldsymbol{a}\|^2 - 2\boldsymbol{a}^\top \mathcal{A}_{\mathbf{n}} \boldsymbol{a} && [\text{Equation H.12}] \\
&\leq 2\mathbf{D}_{ii}^2 \cdot \|\boldsymbol{a}\|^2 - 2\sigma_r^2(\boldsymbol{\Lambda})\|\xi^2\|^2 \cdot \|\boldsymbol{a}\|^2 && [\text{Equation H.24}] \\
&= 2\mathbf{D}_{ii}^2 \cdot \|\boldsymbol{a}\|^2 - 2\sigma_r^2(\boldsymbol{\Lambda})(1 - \|\xi^1\|^2) \cdot \|\boldsymbol{a}\|^2 && [\|\xi^1\|^2 + \|\xi^2\|^2 = 1] \\
&\leq \left(-2\sigma_r^2(\boldsymbol{\Lambda})(1 - \|\xi^1\|^2) + 2e_1 + 2\sigma_{r+1}^2(\boldsymbol{\Lambda})\right) \|\theta_{\mathbf{Y}}^1\|^2
\end{aligned}$$

where the last inequality follows from $\mathbf{D}_{\tilde{i}\tilde{i}}^2 \leq e_1 + \sigma_{r+1}^2(\boldsymbol{\Lambda})$ and the fact that $\|\theta_{\mathbf{Y}}^1\| = \|\boldsymbol{a}\|$. Finally, by applying Equation H.17 to control $\|\xi^1\|$, we conclude that

$$\overline{\operatorname{Hess} H([\mathbf{Y}])} \left[\theta_{\mathbf{Y}}^1, \theta_{\mathbf{Y}}^1\right] \leq -2 \left(\sigma_r^2(\boldsymbol{\Lambda}) \left(1 - \frac{e_1^2}{\left|\sigma_r^2(\boldsymbol{\Lambda}) - e_1 - \sigma_{r+1}^2(\boldsymbol{\Lambda})\right|^2}\right) - e_1 - \sigma_{r+1}^2(\boldsymbol{\Lambda})\right) \|\theta_{\mathbf{Y}}^1\|^2 \tag{H.25}$$

According to the second assumption in Assumption H.1, Equation H.25 guarantees an escape direction.

**Case 2.2:** $\left(\textbf{When } \mathbf{D}_{\tilde{i}\tilde{i}}^2 > e_1 + \sigma_{r+1}^2(\boldsymbol{\Lambda})\right).$

Recall the first assumption in Assumption H.1, we have $e_1$ is small enough, which is viewed as an error term. In the following, we will show that $\theta_{\mathbf{Y}}^2$ is the escaping direction. We have

$$\min_j \mathbf{D}_{\tilde{i}\tilde{i}}^2 |\mathbf{D}_{\tilde{i}\tilde{i}}^2 - \boldsymbol{\Sigma}_{jj}|^2 \leq \mathbf{D}_{\tilde{i}\tilde{i}}^2 \sum_j \left(\mathbf{D}_{\tilde{i}\tilde{i}}^2 - \boldsymbol{\Sigma}_{jj}\right)^2 \boldsymbol{C}_{j\tilde{i}}^2 \leq \frac{\alpha^2 \mu^2 \sigma_r^6(\mathbf{Y}^*)}{(4\kappa^*)^2} \tag{H.26}$$

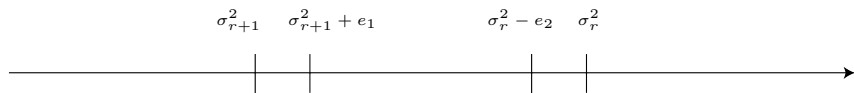

Figure 5: The value of $\mathbf{D}_{\widetilde{ii}}^2$ must be close to some $\sigma_i(\mathbf{\Lambda})$ according to Equation H.26. If $\mathbf{D}_{\widetilde{ii}}^2 > \sigma_{r+1}^2 + e_1$, then we must have $\mathbf{D}_{\widetilde{ii}}^2 \geq \sigma_r^2 - e_2$.

where we use Equation H.14 in the last inequality.

Recall that Assumption H.1 guarantees small $e_1$ and $e_2$ , by combining Equation H.26 and the assumption $\mathbf{D}_{\widetilde{ii}}^2 > \sigma_{r+1}^2(\mathbf{\Lambda}) + e_1$, we must have

$$\mathbf{D}_{\widetilde{ii}}^2 \geq \sigma_r^2(\mathbf{\Lambda}) - e_2 \tag{H.27}$$

where $e_2$ is defined in Assumption H.1. Otherwise, if $\sigma_{r+1}^2(\mathbf{\Lambda}) + e_1 < \mathbf{D}_{\widetilde{ii}}^2 < \sigma_r^2(\mathbf{\Lambda}) - e_2$, this contradicts to Equation H.26; see an illustration of this fact in Figure 5.

In this scenario, we consider the escaping direction $\theta_{\mathbf{Y}}^2$ to be $\mathbf{Y} - \mathbf{Y}^*\mathbf{Q}$. From the fact that $\mathbf{D}_{ii} \geq \mathbf{D}_{\widetilde{ii}}$, we have

$$\alpha^2\mu^2\sigma_r^6\left(\mathbf{Y}^*\right)/\left(4\kappa^*\right)^2 \geq \sum_{i=1}^n \sum_{j=1}^n \left(\mathbf{D}_{ii}^2 - \mathbf{\Sigma}_{jj}\right)^2 \mathbf{D}_{ii}^2 C_{ji}^2 \qquad \text{[Equation H.13]}$$

$$\geq \sum_{i=1}^n \sum_{j=r+1}^n \left|\sigma_j^2(\mathbf{\Lambda}) + e_2 - \sigma_r^2(\mathbf{\Lambda})\right|^2 \cdot \mathbf{D}_{ii}^2 C_{ji}^2$$

where Equation H.27 and the first assumption in Assumption H.1 guarantees the last inequality because $e_2$ is small with respect to $\sigma_r^2(\mathbf{\Lambda}) - \sigma_{r+1}^2(\mathbf{\Lambda})$. Therefore,

$$\sum_{i=1}^n \sum_{j=r+1}^n \mathbf{D}_{ii}^2 C_{ij}^2 \leq \frac{e_3^2}{\left|\sigma_r^2(\mathbf{\Lambda}) - e_2 - \sigma_{r+1}^2(\mathbf{\Lambda})\right|^2} \tag{H.28}$$

where $e_3$ is defined in Assumption H.1. Recall that $e_3$ is small enough, guaranteed in Assumption H.1. Also recall that $e_2 = \frac{e_1}{\sqrt{2}}$, which is guaranteed to be small enough as in the first assumption in Assumption H.1, so $\sigma_r(\mathbf{\Lambda})^2 - e_3 - \sigma_{r+1}^2(\mathbf{\Lambda}) > 0$.

Denote $\mathbf{\Sigma}_{(r+1):n}$ to be a diagonal matrix with only $r+1^{\text{th}}$ to $n^{\text{th}}$ eigenvalues of $\mathbf{\Sigma}$, then we have

$$\langle \mathcal{A}_{\mathbf{n}} - \mathbf{X}^*, \mathbf{Y}\mathbf{Y}^\top \rangle = \langle \mathcal{A}_{\mathbf{n}} - \mathbf{X}^*, \mathbf{U}\mathbf{D}^2\mathbf{U}^\top \rangle$$

$$= \langle \mathcal{A}_{\mathbf{n}} - \mathbf{X}^*, \overline{\mathbf{U}}C\mathbf{D}^2 C^\top \overline{\mathbf{U}}^\top \rangle$$

$$= \langle \mathbf{\Sigma}_{(r+1):n}, C\mathbf{D}^2 C^\top \rangle$$

$$\leq \sigma_{r+1}^2(\mathbf{\Lambda}) \sum_{j=r+1}^n \sum_i C_{ij}^2 \mathbf{D}_{ii}^2 \tag{H.29}$$

$$\leq \frac{e_3^2 \sigma_{r+1}^2(\mathbf{\Lambda})}{\left|\sigma_r^2(\mathbf{\Lambda}) - e_2 - \sigma_{r+1}^2(\mathbf{\Lambda})\right|^2}$$

where the last inequality follows from Equation H.28. Equation H.29 directly implies,

$$\langle \mathcal{A}_{\mathbf{n}} - \mathbf{X}^*, \theta_{\mathbf{Y}}^2(\theta_{\mathbf{Y}}^2)^\top \rangle = \langle \mathcal{A}_{\mathbf{n}} - \mathbf{X}^*, \mathbf{Y}\mathbf{Y}^\top \rangle$$

$$\leq \frac{e_3^2 \sigma_{r+1}^2(\mathbf{\Lambda})}{\left|\sigma_r^2(\mathbf{\Lambda}) - e_2 - \sigma_{r+1}^2(\mathbf{\Lambda})\right|^2} \tag{H.30}$$

because $(\mathcal{A}_{\mathbf{n}} - \mathbf{X}^*)\mathbf{Y}^* = \mathbf{0}$ and $\theta_{\mathbf{Y}}^2 = \mathbf{Y} - \mathbf{Y}^*\mathbf{Q}$.

Recall $\mathbf{X}^* = \mathbf{Y}^*\mathbf{Y}^{*\top}$. A simple calculation yields

$$\mathbf{Y}(\theta_{\mathbf{Y}}^2)^\top - \mathbf{X}^* + \theta_{\mathbf{Y}}^2(\theta_{\mathbf{Y}}^2)^\top = \mathbf{Y}(\theta_{\mathbf{Y}}^2)^\top + \theta_{\mathbf{Y}}^2\mathbf{Y}^\top \tag{H.31}$$

and by using Equation H.1,

$$\begin{aligned}
\langle \overline{\operatorname{grad} H([\mathbf{Y}])}, \theta_{\mathbf{Y}}^2 \rangle &= \langle 2\left(\mathbf{Y}\mathbf{Y}^\top - \mathcal{A}_{\mathbf{n}}\right)\mathbf{Y}, \theta_{\mathbf{Y}}^2 \rangle \\
&= \langle 2(\mathbf{Y}\mathbf{Y}^\top - \mathcal{A}_{\mathbf{n}}), \theta_{\mathbf{Y}}^2 \mathbf{Y}^\top \rangle \\
&= \langle \mathbf{Y}\mathbf{Y}^\top - \mathcal{A}_{\mathbf{n}}, \theta_{\mathbf{Y}}^2\mathbf{Y}^\top + \mathbf{Y}(\theta_{\mathbf{Y}}^2)^\top \rangle \qquad \text{[first argument is symmetric]} \\
&= \langle \mathbf{Y}\mathbf{Y}^\top - \mathcal{A}_{\mathbf{n}}, \theta_{\mathbf{Y}}^2(\theta_{\mathbf{Y}}^2)^\top + \mathbf{Y}\mathbf{Y}^\top - \mathbf{X}^* \rangle .
\end{aligned} \tag{H.32}$$

where the last equality follows from Equation H.31.

$$\begin{aligned}
\overline{\operatorname{Hess} H([\mathbf{Y}])}\left[\theta_{\mathbf{Y}}^2, \theta_{\mathbf{Y}}^2\right] &= \left\|\mathbf{Y}(\theta_{\mathbf{Y}}^2)^\top + \theta_{\mathbf{Y}}^2\mathbf{Y}^\top\right\|_{\mathrm{F}}^2 + 2\langle \mathbf{Y}\mathbf{Y}^\top - \mathcal{A}_{\mathbf{n}}, \theta_{\mathbf{Y}}^2(\theta_{\mathbf{Y}}^2)^\top \rangle & \text{[Equation H.1]} \\
&= \left\|\mathbf{Y}\mathbf{Y}^\top - \mathbf{X}^* + \theta_{\mathbf{Y}}^2(\theta_{\mathbf{Y}}^2)^\top\right\|_{\mathrm{F}}^2 + 2\langle \mathbf{Y}\mathbf{Y}^\top - \mathcal{A}_{\mathbf{n}}, \theta_{\mathbf{Y}}^2(\theta_{\mathbf{Y}}^2)^\top \rangle & \text{[Equation H.31]} \\
&= \left\|\theta_{\mathbf{Y}}^2(\theta_{\mathbf{Y}}^2)^\top\right\|_{\mathrm{F}}^2 + \left\|\mathbf{Y}\mathbf{Y}^\top - \mathbf{X}^*\right\|_{\mathrm{F}}^2 + 4\langle \mathbf{Y}\mathbf{Y}^\top - \mathbf{X}^*, \theta_{\mathbf{Y}}^2(\theta_{\mathbf{Y}}^2)^\top \rangle \\
&\quad - 2\langle \mathcal{A}_{\mathbf{n}} - \mathbf{X}^*, \theta_{\mathbf{Y}}^2(\theta_{\mathbf{Y}}^2)^\top \rangle \\
&= \left\|\theta_{\mathbf{Y}}^2(\theta_{\mathbf{Y}}^2)^\top\right\|_{\mathrm{F}}^2 - 3\left\|\mathbf{Y}\mathbf{Y}^\top - \mathbf{X}^*\right\|_{\mathrm{F}}^2 + 4\langle \mathbf{Y}\mathbf{Y}^\top - \mathbf{X}^*, \mathbf{Y}\mathbf{Y}^\top - \mathbf{X}^* + \theta_{\mathbf{Y}}^2(\theta_{\mathbf{Y}}^2)^\top \rangle \\
&\quad - 2\langle \mathcal{A}_{\mathbf{n}} - \mathbf{X}^*, \theta_{\mathbf{Y}}^2(\theta_{\mathbf{Y}}^2)^\top \rangle \\
&= \left\|\theta_{\mathbf{Y}}^2(\theta_{\mathbf{Y}}^2)^\top\right\|_{\mathrm{F}}^2 - 3\left\|\mathbf{Y}\mathbf{Y}^\top - \mathbf{X}^*\right\|_{\mathrm{F}}^2 + 4\langle \mathbf{Y}\mathbf{Y}^\top - \mathcal{A}_{\mathbf{n}}, \mathbf{Y}\mathbf{Y}^\top - \mathbf{X}^* + \theta_{\mathbf{Y}}^2(\theta_{\mathbf{Y}}^2)^\top \rangle \\
&\quad + 2\langle \mathcal{A}_{\mathbf{n}} - \mathbf{X}^*, \theta_{\mathbf{Y}}^2(\theta_{\mathbf{Y}}^2)^\top \rangle + 4\langle \mathcal{A}_{\mathbf{n}} - \mathbf{X}^*, \mathbf{Y}\mathbf{Y}^\top - \mathbf{X}^* \rangle \\
&= \left\|\theta_{\mathbf{Y}}^2(\theta_{\mathbf{Y}}^2)^\top\right\|_{\mathrm{F}}^2 - 3\left\|\mathbf{Y}\mathbf{Y}^\top - \mathbf{X}^*\right\|_{\mathrm{F}}^2 + 4\langle \mathcal{A}_{\mathbf{n}} - \mathbf{X}^*, \mathbf{Y}\mathbf{Y}^\top - \mathbf{X}^* \rangle \\
&\quad + 2\langle \mathcal{A}_{\mathbf{n}} - \mathbf{X}^*, \theta_{\mathbf{Y}}^2(\theta_{\mathbf{Y}}^2)^\top \rangle + 4\langle \overline{\operatorname{grad} H([\mathbf{Y}])}, \theta_{\mathbf{Y}}^2 \rangle & \text{[Equation H.32]}
\end{aligned}$$

This decomposes $\overline{H([\mathbf{Y}])}\left[\theta_{\mathbf{Y}}^2, \theta_{\mathbf{Y}}^2\right]$ into 2 parts, which will be bounded separately.

First, for $\left\|\theta_{\mathbf{Y}}^2(\theta_{\mathbf{Y}}^2)^\top\right\|_{\mathrm{F}}^2 - 3\left\|\mathbf{Y}\mathbf{Y}^\top - \mathbf{X}^*\right\|_{\mathrm{F}}^2 + 2\langle \mathcal{A}_{\mathbf{n}} - \mathbf{X}^*, \theta_{\mathbf{Y}}^2(\theta_{\mathbf{Y}}^2)^\top \rangle + 4\langle \mathcal{A}_{\mathbf{n}} - \mathbf{X}^*, \mathbf{Y}\mathbf{Y}^\top - \mathbf{X}^* \rangle$, we have

$$\begin{aligned}
&\left\|\theta_{\mathbf{Y}}^2(\theta_{\mathbf{Y}}^2)^\top\right\|_{\mathrm{F}}^2 - 3\left\|\mathbf{Y}\mathbf{Y}^\top - \mathbf{X}^*\right\|_{\mathrm{F}}^2 + 2\langle \mathcal{A}_{\mathbf{n}} - \mathbf{X}^*, \theta_{\mathbf{Y}}^2(\theta_{\mathbf{Y}}^2)^\top \rangle \\
&\quad + 4\langle \mathcal{A}_{\mathbf{n}} - \mathbf{X}^*, \mathbf{Y}\mathbf{Y}^\top - \mathbf{X}^* \rangle \\
&\leq -\left\|\mathbf{Y}\mathbf{Y}^\top - \mathbf{X}^*\right\|_{\mathrm{F}}^2 + 2\langle \mathcal{A}_{\mathbf{n}} - \mathbf{X}^*, \theta_{\mathbf{Y}}^2(\theta_{\mathbf{Y}}^2)^\top \rangle \\
&\quad + 4\langle \mathcal{A}_{\mathbf{n}} - \mathbf{X}^*, \mathbf{Y}\mathbf{Y}^\top - \mathbf{X}^* \rangle & \text{[Equation H.4]} \\
&= -\left\|\mathbf{Y}\mathbf{Y}^\top - \mathbf{X}^*\right\|_{\mathrm{F}}^2 + 2\langle \mathcal{A}_{\mathbf{n}} - \mathbf{X}^*, \theta_{\mathbf{Y}}^2(\theta_{\mathbf{Y}}^2)^\top \rangle + 4\langle \mathcal{A}_{\mathbf{n}} - \mathbf{X}^*, \mathbf{Y}\mathbf{Y}^\top \rangle & [\langle \mathcal{A}_{\mathbf{n}} - \mathbf{X}^*, \mathbf{X}^* \rangle = 0] \\
&\leq -\left\|\mathbf{Y}\mathbf{Y}^\top - \mathbf{X}^*\right\|_{\mathrm{F}}^2 + 6\frac{e_3^2\sigma_{r+1}^2(\mathbf{\Lambda})}{\left|\sigma_r^2(\mathbf{\Lambda}) - e_2 + \sigma_{r+1}^2(\mathbf{\Lambda})\right|^2} & \text{[Equation H.29,Equation H.30]} \\
&\leq -2(\sqrt{2}-1)\sigma_r^2(\mathbf{Y}^*)\left\|\theta_{\mathbf{Y}}^2\right\|_{\mathrm{F}}^2 + 6\frac{e_3^2\sigma_{r+1}^2(\mathbf{\Lambda})}{\left|\sigma_r^2(\mathbf{\Lambda}) - e_2 + \sigma_{r+1}^2(\mathbf{\Lambda})\right|^2} & \text{[Equation H.3]}
\end{aligned}$$

Second, for $\langle \overline{\operatorname{grad} H([\mathbf{Y}])}, \theta_{\mathbf{Y}}^2 \rangle$,

$$\begin{aligned}
\langle \overline{\operatorname{grad} H([\mathbf{Y}])}, \theta_{\mathbf{Y}}^2 \rangle &\leq \|\overline{\operatorname{grad} H([\mathbf{Y}])}\|_{\mathrm{F}} \left\|\theta_{\mathbf{Y}}^2\right\|_{\mathrm{F}} \\
&\leq \alpha\sigma_r^2(\mathbf{Y}^*)\left\|\theta_{\mathbf{Y}}^2\right\|_{\mathrm{F}}^2
\end{aligned}$$

where the last inequality is because $\|\overline{\operatorname{grad} H([\mathbf{Y}])}\|_{\mathrm{F}} \leqslant \alpha\mu\sigma_r^3(\mathbf{Y}^*)/(4\kappa^*)$. According to the definition of $\mathcal{R}_2$ in Equation 3.3, $\mathbf{Y} \in \mathcal{R}_2$ also implies $d([\mathbf{Y}],[\mathbf{Y}^*]) > \mu\sigma_r(\mathbf{Y}^*)/\kappa^*$, then

$$\|\overline{\operatorname{grad} H([\mathbf{Y}])}\|_{\mathrm{F}} \leqslant \alpha d([\mathbf{Y}],[\mathbf{Y}^*])\sigma_r^2(\mathbf{Y}^*)/4 = \alpha\|\theta_{\mathbf{Y}}^2\|_{\mathrm{F}}\sigma_r^2(\mathbf{Y}^*)/4$$

By combining the above three inequalities, we have

$$\overline{\operatorname{Hess} H([\mathbf{Y}])}\left[\theta_{\mathbf{Y}}^2, \theta_{\mathbf{Y}}^2\right] = \left\|\theta_{\mathbf{Y}}^2(\theta_{\mathbf{Y}}^2)^\top\right\|_{\mathrm{F}}^2 - 3\left\|\mathbf{Y}\mathbf{Y}^\top - \mathbf{X}^*\right\|_{\mathrm{F}}^2$$
$$+ 4\left\langle\overline{\operatorname{grad} H([\mathbf{Y}])}, \theta_{\mathbf{Y}}^2\right\rangle + 2\left\langle\mathcal{A}_\mathbf{n} - \mathbf{X}^*, \theta_{\mathbf{Y}}^2(\theta_{\mathbf{Y}}^2)^\top\right\rangle + 4\left\langle\mathcal{A}_\mathbf{n} - \mathbf{X}^*, \mathbf{Y}\mathbf{Y}^\top - \mathbf{X}^*\right\rangle$$

$$\leq (\alpha - 2(\sqrt{2}-1))\sigma_r^2(\mathbf{Y}^*)\left\|\theta_{\mathbf{Y}}^2\right\|_{\mathrm{F}}^2 + 6\frac{e_3^2\sigma_{r+1}^2(\mathbf{\Lambda})}{\left|\sigma_r^2(\mathbf{\Lambda}) - e_2 - \sigma_{r+1}^2(\mathbf{\Lambda})\right|^2}$$

$$\leq \left((\alpha - 2(\sqrt{2}-1))\sigma_r^2(\mathbf{Y}^*) + 6\frac{\alpha^2\sigma_r^4(\mathbf{Y}^*)\sigma_{r+1}^2(\mathbf{\Lambda})/16}{\left|\sigma_r^2(\mathbf{\Lambda}) - e_2 - \sigma_{r+1}^2(\mathbf{\Lambda})\right|^2}\right)\left\|\theta_{\mathbf{Y}}^2\right\|_{\mathrm{F}}^2$$

where the last inequality follows from $\mu\sigma_r(\mathbf{Y}^*)/\kappa^* \leq d([\mathbf{Y}],[\mathbf{Y}^*]) = \|\theta_{\mathbf{Y}}\|_{\mathrm{F}}$ and the definition of $e_3$ in Assumption H.1.

Finally, according to the third assumption in Assumption H.1, one can guarantee the right-hand side of this bound is negative, which implies that $\theta_{\mathbf{Y}}^2$ is the escaping direction in this scenario.

Combining all the discussion, this finishes the proof of this theorem. $\qquad\square$

**Remark H.3.** *Theorem H.1 suggests that if some spectral values of $\mathbf{Y}$ are small, then the descent direction $\theta_{\mathbf{Y}}^1$ should increase them, If all of the spectral values of $[\mathbf{Y}]$ are large enough compared with $\sigma_r(\mathbf{\Lambda})$, then $\theta_{\mathbf{Y}}^2$ should directly point $[\mathbf{Y}]$ to $[\mathbf{Y}^*]$. Theorem H.1 fully characterizes the regime of $\mathbf{Y}$ with respect to different minimum spectral values of $\mathbf{Y}$.*

- *If any spectral value of $\mathbf{Y}\mathbf{Y}^\top$ is smaller than $\frac{\sigma_{r+1}^2(\mathbf{\Lambda})}{2}$, then we have*

$$\overline{\operatorname{Hess} H([\mathbf{Y}])}\left[\theta_{\mathbf{Y}}^1, \theta_{\mathbf{Y}}^1\right] \leq -\frac{\sigma_{r+1}^2(\mathbf{\Lambda})}{2}\|\theta_{\mathbf{Y}}^1\|^2$$

- *When the smallest absolute spectral value of $\mathbf{Y}\mathbf{Y}^\top$ is larger than $\frac{\sigma_{r+1}^2(\mathbf{\Lambda})}{2}$ and smaller than $e_1 + \sigma_{r+1}^2(\mathbf{\Lambda})$, then we have*

$$\overline{\operatorname{Hess} H([\mathbf{Y}])}\left[\theta_{\mathbf{Y}}^1, \theta_{\mathbf{Y}}^1\right] \leq -2\left(\sigma_r^2(\mathbf{\Lambda})\left(1 - \frac{e_1^2}{\left|\sigma_r^2(\mathbf{\Lambda}) - e_1 - \sigma_{r+1}^2(\mathbf{\Lambda})\right|^2}\right) - e_1 - \sigma_{r+1}^2(\mathbf{\Lambda})\right)\|\theta_{\mathbf{Y}}^1\|^2$$

- *If all of the spectral values of $\mathbf{Y}\mathbf{Y}^\top$ is larger than $\frac{\alpha\mu\sigma_r^3(\mathbf{Y}^*)}{2\sqrt{2}\kappa^*\sigma_{r+1}(\mathbf{\Lambda})} + \sigma_{r+1}^2(\mathbf{\Lambda})$, then we have $\overline{\operatorname{Hess} H([\mathbf{Y}])}\left[\theta_{\mathbf{Y}}^2, \theta_{\mathbf{Y}}^2\right]$ is smaller than*

$$\left((\alpha - 2(\sqrt{2}-1))\sigma_r^2(\mathbf{Y}^*) + 6\frac{\alpha^2\sigma_r^4(\mathbf{Y}^*)\sigma_{r+1}^2(\Lambda)/16}{\left|\sigma_r^2(\mathbf{\Lambda}) - e_2 - \sigma_{r+1}^2(\mathbf{\Lambda})\right|^2}\right)\left\|\theta_{\mathbf{Y}}^2\right\|_{\mathrm{F}}^2$$

**Remark H.4.** *The eigengap assumption is crucial in discussing the three regions of the minimum singular value of $\mathbf{Y}$. Without this eigengap assumption and under the current quotient geometry, the third regime cannot lead to a strong convexity result because any span on the eigenspace are all global solutions. We comment that it is possible to change the quotient geometry to show a new strong convexity result when the eigengap assumption does not hold.*

Finally, we look at the last main result. Theorem 3.4 guarantees that when $\mathbf{Y} \in \mathcal{R}_3$, the magnitude of the Riemannian gradient descent is large. The proof of Theorem 3.4 directly follows from the proof of (Luo & García Trillos, 2022) without any modification. Hence, we do not repeat it here. Notice that $\mathbf{Y} \in \mathcal{R}_3$ does not require Assumption 3.1 because $\mathcal{R}_3$ describes the case that $[\mathbf{Y}]$ is far away from the FOSP.

**Theorem 3.4** ((Regions with Large Riemannian Gradient of Equation 1.5).

1. $\|\overline{\operatorname{grad} H([\mathbf{Y}])}\|_{\mathrm{F}} > \alpha \mu \sigma_r^3 \left(\mathbf{Y}^*\right) / \left(4\kappa^*\right), \forall \mathbf{Y} \in \mathcal{R}_3';$

2. $\|\overline{\operatorname{grad} H([\mathbf{Y}])}\|_{\mathrm{F}} \geqslant 2 \left(\|\mathbf{Y}\|^3 - \|\mathbf{Y}\| \|\mathbf{Y}^*\|^2\right) > 2 \left(\beta^3 - \beta\right) \|\mathbf{Y}^*\|^3, \quad \forall \mathbf{Y} \in \mathcal{R}_3'';$

3. $\left\langle \overline{\operatorname{grad} H([\mathbf{Y}])}, \mathbf{Y} \right\rangle > 2(1 - 1/\gamma) \left\|\mathbf{Y}\mathbf{Y}^\top\right\|_{\mathrm{F}}^2, \quad \forall \mathbf{Y} \in \mathcal{R}_3'''.$

*In particular, if $\beta > 1$ and $\gamma > 1$, we have the Riemannian gradient of $H([\mathbf{Y}])$ has large magnitude in all regions $\mathcal{R}_3', \mathcal{R}_3''$ and $\mathcal{R}_3'''$.*

