# OpenReview forum: "Spectral Neural Networks: Approximation Theory and Optimization Landscape"
_ICLR.cc/2024/Conference — Submitted to ICLR 2024_

### Official Review · Reviewer_sU7q · 2023-10-31

**Soundness:** 3 good
**Presentation:** 2 fair
**Contribution:** 3 good
**Rating:** 6
**Confidence:** 2

**Summary:**

This paper theoretically studies several questions regarding spectral neural networks and, more generally, neural networks that are trained to approximate the eigenvalues of specific matrices. The authors prove that multi-layer ReLU NNs can approximate normalized Laplacians, with specific bounds on the error, depth of network, and number of neurons. The authors then show that NNs can approximate eigenvectors, up to rotation. Finally, the authors consider the loss landscape of the optimization and show that, in a quotient geometry, they can decompose the loss landscape into 5 regions (3 that are particularly different).

**Strengths:**

1. This paper provides detailed theoretical results for spectral neural networks, among other neural networks that are trained to approximate the spectra of matrices, a field of growing importance. I believe this work will provide greater insight into this area.

2. The authors provide a detailed overview of existing work that is related.

3. Aspects of the paper were well written and well motivated.

**Weaknesses:**

As a note, I was not able to follow the theoretical results, so my review is limited (as reflected by my confidence score). However, I believe there are several ways in which this paper could be made stronger:

1. Q1 is motivated as being of practical importance - how many neurons are needed to achieve an accuracy of X%. While the theoretical results provide this, they are difficult to parse into practical considerations. It would be helpful to have predictions of the theory on how many neurons are needed (and how many layers) for a given accuracy plotted (for at least some choice of parameters described in Theorem 2.1). A comparison with an actual implementation of SNN would make the theoretical results especially convincing (if they match up).

2. Q1, as phrased in the Introduction, suggests that it is unknown whether it is "possible to approximate eigenvectors of large adjacency matrices with a neural network". However, as noted by the authors, there has been work showing success in this direction already. Perhaps it would be better to phrase Q1 as "are there theoretical guarantees that a neural network can approximate the eigenvectors of large adjacency matrices".

3. Sec. 2.2 is said to be aimed at "constructive ways to approximate $\textbf{Y}^*$, but it is unclear how Theorem 2.2 achieves this. There is not mention of optimization in the theorem, and the number of neurons is set to $N = \infty$. While I understand that the number of neurons can be reduced (in Remark 2.3), this mismatch between aim and result disrupted the flow of the paper.

4. The figures (Fig. 3-5) were mentioned only briefly in the introduction (before many of the details of the paper were introduced and I was under the impression that they would be referenced later in more detail. As it stands, I did not get anything from the figures. Including them later in the text, as the experiments are motivated (e.g., why it is reasonable to consider "Near optimal", "Large gradient", and "Near saddle") would greatly improve their impact.

5. I did not understand what was meant in the Introduction by the "spectral contrastive loss function $\mathcal{l}$ is non-convex in the 'ambient' space' " until the discussion about the need for the quotient geometry in Sec 3. I think making this point more clear earlier in the paper would help the reader understand why this is an interesting and tricky problem.

**Questions:**

My questions can be found in the section above.

---

> ### Author Response · Authors · 2023-11-15
> **Response to Reviewer sU7q**
>
> We thank the reviewer for their interesting questions and suggestions. We are encouraged by their appreciation of our submission regarding its motivation, theoretical soundness, comprehensive literature review, and presentation. Below, we provide answers to the reviewer’s questions. We have added most of these explanatory comments and changes in the updated manuscript in blue text.
>
> > Simulations.
>
> We thank the reviewer for suggesting further simulations exploring the optimality of our theoretical results. This is a good suggestion that can be presented within the context of the broader research inquiries suggested by Reviewer hkrF. Having said this, we want to highlight the merits of our theoretical contributions in the present paper. Indeed, we bring together, in non-trivial and novel ways, a variety of results from several subfields (manifold learning, neural network approximation, Riemannian optimization, matrix factorization problems) that, in our view, have had limited theoretical interaction. We hope this paper motivates systematic theoretical explorations of important applications of neural networks to eigenvalue problems of differential operators on (unknown) manifolds in high dimensional spaces (see, for example, some suggestions by Reviewer hkrF).
>
> > About presentation.
>
> The updated manuscript emphasizes the theoretical nature of our inquiries when describing Q1-Q3. The existing empirical results in the literature were indeed important motivators for our work.
>
> > Constructive nature in Theorem 2.2.
>
> We want to highlight the differences in nature of Theorems 2.1 and 2.2: it is very different from stating that an approximating NN can be found by solving an optimization problem with a **clearly defined objective** (as in Theorem 2.2) than to simply state the existence of an approximating NN with no indication on how even to begin the search for it (as in Theorem 2.1.). While we agree that Theorem 2.2. does not describe an explicit algorithm to find this approximating NN, it is implicit that once the clearly defined target objective has been introduced, one can use any optimization algorithm to search for its minimizers. Because of this, and given our desire to highlight the conceptual differences between Theorems 2.1 and 2.2, we used the word constructive to describe the latter. We would be happy to modify this term if the reviewer believes that there is a more suitable alternative.
>
> > Figures.
>
> We thank the reviewer for their suggestion. In the revised manuscript, we refer to our figures after Remark 3.1 when discussing the theoretical results about the landscape of the objective $\ell$.
>
> > Highlight non-convexity of the loss function $\ell$.
>
> We thank the reviewer for their suggestion. In the revised manuscript, we have highlighted this point in the paragraph mentioned by the reviewer.

---

> > ### Comment · Reviewer_sU7q · 2023-11-15
> >
> > I thank the authors for their time and detailed comments. In reading through their responses, both to my own reviews, as well as those of the other reviewers, I believe the authors have done a good job addresses questions and concerns.
> >
> > I do have a few remaining comments/questions:
> >
> > 1. Is it correct that the authors have not included any simulations relating number of neurons needed to achieve a given accuracy and any comparisons with any implementations of SNNs?
> >
> > 2. The authors mention that Thm. 2.2 is distinct from Thm. 2.1, because it is in terms of a distinct objective. I think it would be helpful, for understanding this point, as well as seeing the connection between Sec. 3 and Thm. 2.2, if this objective was more clearly mentioned and its role in Thm. 2.2 more clearly stated.
> >
> > 3. Having Figures 1 and 2 be subparts of Figure 3 is confusing (because they are easy to miss and it can suggest to readers there are missing figures).

---

> ### Author Response · Authors · 2023-11-17
> **Thank you for your detailed comments**
>
> Thank you for your thoughtful feedback on our manuscript and response.
>
> We have taken your suggestions into consideration and made the following revisions:
> 1. Removed the title of Figure 3.
> 2. Reformatted the objective function to be presented in a single line.
>
> We believe these changes enhance the clarity of our paper.
>
> Regarding the simulation of neuron numbers for accuracy approximation in eigenvector calculations, we acknowledge this as an intriguing area for future research. This aspect, closely tied to the Laplace-Beltrami operator as highlighted in Corollary 1, offers a promising avenue for exploring the relationship between theoretical results and empirical findings. However, our current focus is on the theoretical foundations of Spectral Neural Networks (SNNs). Consequently, our current simulations were confined to the parameterized and ambient optimization landscapes in SNNs.
>
> Future studies could also fruitfully compare SNNs with other neural network models for spectral embedding tasks. While incorporating a detailed discussion on approximations to Partial Differential Equation (PDE) operators would indeed be valuable, we believe it might overburden the scope of this paper. Such topics merit a thorough examination in subsequent research.
>
> We appreciate your consideration of our revisions and suggestions for simulation results. We believe the improvements address your initial concerns, enhancing both the clarity and rigor of our work.  If you find that our revisions meet your expectations and improve the overall quality of the manuscript, we would be grateful if you could reflect this in an updated assessment on rating or confidence.

---

> > ### Comment · Reviewer_sU7q · 2023-11-22
> >
> > I thank the authors for making these changes. I believe they do indeed make the paper stronger.
> >
> > While I understand the focus of the paper is the theoretical foundation of SNNs, I believe that connecting the theory to empirical results would make a clearer paper that strengthens the analytical results. That being said, not having it will not decrease my score (just not increase my score).

---

> > > ### Author Response · Authors · 2023-11-22
> > > **Thank you**
> > >
> > > We express our gratitude for the reviewer's comments. The restriction of a 9-page limit is the factor limiting our capacity to expand the content. Understanding the role of simulation in showcasing the optimal theoretical rate, we have focused on including substantive remarks in the main manuscript. This approach ensures that the significance of the theoretical results is not understated. To comprehensively address the simulation comparison of both discrete and continuum metrics, we have earmarked this aspect for exploration in our future work.

---

### Official Review · Reviewer_BBbU · 2023-11-07

**Soundness:** 2 fair
**Presentation:** 1 poor
**Contribution:** 2 fair
**Rating:** 5
**Confidence:** 3

**Summary:**

The paper makes three main contributions:

* It establishes approximation bounds on the depth and the number of neurons needed for a multi-layer neural network to accurately approximate top eigenvectors of a normalized graph Laplacian matrix constructed from data samples lying on a manifold.

* It shows that by globally optimizing the spectral contrastive loss function, one can provably construct a neural network that approximates the eigenvectors up to rotations.

* Motivated by experiments, the paper analyzes the non-convex optimization landscape of the ambient spectral contrastive loss function and shows it to be benign.

**Strengths:**

* Given the amount of recent interest in using neural networks to approximate eigenfunctions, establishing theoretical guarantees for such algorithms is timely and of interest to the community.

* The proofs of results in section 2 looks sensible to me, except the question mentioned below.

**Weaknesses:**

It should be noted that my review did not cover section 3, as I found it difficult to absorb in a reasonable amount of time given the mathematical depth.

* The theory established in this paper seems far from providing any practical insights on training spectral neural networks besides proving the feasibility of such an approach--at least the authors did not attempt to include argument like this in the paper. Not covering data-dependent kernels as those in HaoChen and most SSL work also reduce the significance of the result.

* The proof of the approximation result (Theorem 2.1) is quite straightforward, combining (known) ReLU network approximation results (Chen et al., 2022) with a Lipschizness-like condition for eigenvectors on manifolds (Calder et al., 2022).  It is not clear whether or not the proof could be useful for future theoretical work in this space.

* The presentation of Theorem 2.2 and its proof needs clarification, e.g., I found the paragraph following Corollary 5 difficult to understand: "Using the fact that bar{U} is invertible, we can easily see that Y_\theta* ....". Can you clarify how to get this result? I assume Y_\theta* is the Y recovered by the optimal neural network that minimizes the spectral contrastive loss. I also find it difficult to see the reasoning behind Remark G.1-G.3 and how they fit into the proof.

* The exposition of the main theorems can be improved. The neural network family constants are never defined in the main text and it makes Theorem 2.1 very hard to read. I strongly encourage the authors to provide intuitive explanations before/after each theorem to aid the understanding of assumptions used, proof ideas, and implications of the result.

**Questions:**

Please see the above question about proof of Theorem 2.2.

---

> ### Author Response · Authors · 2023-11-15
> **Response to Reviewer BBbU**
>
> We thank the reviewer for their comments and questions, which we address below. We are also encouraged by the reviewer’s appreciation for our theoretical contribution to the ML community.
>
> > Significance of approximation result.
>
> We start by highlighting that SNNs are useful beyond SSL and the context of the work of HaoChen et al. 2021 Indeed, our results can justify the use of SNNs to solve eigenvalue problems of differential operators on manifolds. The use of SNN in this context is novel, and in particular the induced training procedure is quite different from popular approaches in the literature such as Physics Informed Neural Networks (PINNS). The use of SNN beyond standard tasks in machine learning is thus of interest to the broader scientific community. See, for example, our response to Reviewer hkrF and the Conclusions section in our paper.
>
> On the other hand, we emphasize that studying the graph constructions in HaoChen et al. 2022 is an interesting research direction and we expect that a similar analysis to the one presented in our paper can be carried out in settings that are more useful for SSL. However, to achieve this, one would first need to generalize some of the theoretical results in Calder et al 2022. This mathematical task seems feasible, but only after a lengthy analysis. This can be explored in future work.
>
> > The proof of Theorem 2.1.
>
> We refer the reviewer to the general response to reviewers.
>
> > The decomposition of $Y_{\theta^*}$.
>
> Recall that $\bar{U}$ is invertible. Hence, let $E = \bar{U}^{-1}Y_{\theta^*}$. We let $E^1$ be the first $r$ rows of $E$ with the rest of rows being equal to 0. Similarly, $E^2$ has the rest of the rows of $E$.
>
> > Remarks G.1 - G.3.
>
> More than being necessary for our proofs, these remarks help convey why our results are not direct consequences of existing results and that careful analysis is needed to deduce them. This is closely connected to the discussion in our general response to reviewers.
>
> > The neural network family that we consider.
>
> We thank the reviewer for their suggestion. The family of approximating functions was defined in detail in (C.2); this was mentioned in the statement of Theorem 2.1. We do not include its (long) definition in the main text because of space limits. Given all the other important things we needed to include in the main body of the paper, we were forced to move some details to the appendix. We also want to highlight that many remarks in the main body of the paper and in the appendix were included to help convey the relevance of our theoretical results.
>
>
> We hope our explanations can help convey that our work is well motivated and that the mathematical problems we study are exciting and non-trivial.

---

### Official Review · Reviewer_SXMx · 2023-11-07

**Soundness:** 2 fair
**Presentation:** 2 fair
**Contribution:** 2 fair
**Rating:** 3
**Confidence:** 3

**Summary:**

This paper considers minimizing the loss function $\|Y(\theta)Y(\theta)^T-A_n\|^2_F$, where the low rank matrix $Y$ is estimated by a neural network. The main claims of the paper are that: (C1) the optimal $Y^*$ can be well approximated by a neural network; (C2) the global minima of the loss function is close to the optimal $Y^*$ (up to a rotation); (C3) the loss function $\|YY^T-A_n\|^2_F$, as a function of $Y$, has nice geometrical properties.

**Strengths:**

The paper explicitly writes out the approximation error bound, requirements on the depth and number of neurons, of the neural network on this spectral approximation problem.

It also showcase a few nice properties of $\|YY^T-A_n\|^2_F$, as a function of $Y$.

**Weaknesses:**

Given the universal approximation theorem of neural networks, it is expected that there exist a neural network that can approximate the optimal matrix $Y^*$. Hence, (C1) should be a natural result. Moreover, most of the techniques seem not new and appeared in prior works, e.g., (Chen et al. 2022).

Given (C1), the second main claim (C2), e.g., Theorem 2.2, should be quite obvious. (C2) does not discuss the solvability of the optimization problem, i.e., how to find the global minima. Hence, I don’t see a “constructive way” to find the approximation. To me, this part (C2) is more like a result of the existence of such a neural network, which highly overlaps with the claim in (C1).

When analyzing the loss landscape, in Section 3, the paper does not consider the loss function as a function of the network parameters. However, it considers it as a function of the network output. More explicitly, it is basically analyzing $\|YY^T-A_n\|^2_F$ as a function of $Y$, not of $\theta$. First of all, this “landscape” is not the optimization landscape we are mostly interested in. One has to compose it with the network function $Y(\theta)$ to have the full optimization loss. As we know, the hard part is the network function $Y(\theta)$. Second, given the simple and symmetrical form of $\|YY^T-A_n\|^2_F$, the results presented in Section 3 are not hard to obtain.

The presentation of the paper can be improved. For example, I had a hard time understanding the notations in the theorems. For example in Theorem 2.1 it is not clear what is $\epsilon$ and $m$. In Eq.(1.2),  $\epsilon$ is used for the “bandwidth” for similarity, however, in the proof of Theorems, $\epsilon$ seems an arbitrarily small positive number. In addition, the meaning of $m$ was not mentioned in the statement of the theorem or its proof.

**Questions:**

no further questions

---

> ### Author Response · Authors · 2023-11-15
> **Response to Reviewer SXMx**
>
> We thank the reviewer for their comments and questions, which we address in what follows.
>
> > The approximation results in Section 2.
>
> Please see the general response to reviewers.
>
> > Constructive approximations in Theorem 2.2.
>
> Theorem 2.2. is not a mere existence result. It is very different to state that an approximating NN can be found by solving an optimization problem with a **clearly defined objective** than to simply state the existence of an approximating NN with no indication on how even to begin the search for it (as in Theorem 2.1.). That is, existence just says there are parameters such that something is true, but gives no indication on how to find these parameters. Of course, once one has a clearly defined objective, one can ask the natural follow-up question: what is the behavior of popular optimization algorithms when trying to optimize the target objective? This is what we explore in Q3.
>
> > The landscape results in Section 3.
>
> As was discussed in paragraphs 2, 3, and 4 on page 4, the optimization landscape is non-trivial because there are two sources of “non-convexity”: 1) the non-linearity of neural networks and 2) the non-convexity of the ambient loss function. With our numerical illustration (Figures 4 and 5), we suggest that the properties of the landscape for the SNN problem are expected to resemble those of the ambient loss landscape, at least in some form of overparameterized regime. A precise mathematical statement along these lines will be explored in the future and deserves its own careful analysis. As we discussed in paragraphs 2, 3, and 4 on page 4, as well as in the last paragraph of the Conclusions section, this problem is different from other analyses in the literature of training dynamics of overparameterized neural networks because in our case, the ambient loss function $\ell$ is non-convex, not because neural networks are nonlinear functions.
>
> > Notation.
>
> We defined $\epsilon$ in the surroundings of Equation (1.2) and defined $m$ in Assumption 2.1. It is the underlying manifold’s intrinsic dimension. $\epsilon$ in this work is always used as the length scale that determines the graph connectivity. $\delta$, on the other hand, is a tunable approximation error. Notice that the size of $\epsilon$ is implicitly controlled by the number of data points available since our error estimates are only meaningful with a likelihood that depends on $n$ and $\epsilon$.
>
> We hope our explanations can help convey that our work is well motivated and that our mathematical contributions are novel, sound, and far from trivial.

---

> > ### Comment · Reviewer_SXMx · 2023-11-22
> >
> > I thank the authors for their detailed response. However, my major concerns remain.
> >
> > 1: Given the universal approximation property of neural networks, it is almost obvious to have theorem 2.1. It seems to me that the novelty is to explicitly write out the expressions of requirement and probability, which is not sufficient to meet the ICLR standard.
> >
> > 2: Given the approximation result, I still think it is trivial to translate it into solving the optimization problem. I could not think it as a contribution of the paper.
> >
> > 3: As for the loss landscape, it is actually just analyzing the $||YY^T-A_n||_F$ as a function of $Y$ instead of $\theta$. Given the simple form of this function, I think it is a course project level problem to analyze. The interesting aspect is still the loss landscape as a function of the model parameters $\theta$.
> >
> > Overall, I think the contributions of the paper are minor. I would like to keep my score.

---

> > > ### Author Response · Authors · 2023-11-22
> > > **Thank you for your comment**
> > >
> > > Thank you for the replies. We invite the reviewer to carefully consider the following replies to their points:
> > >
> > > 1. The universal approximation theory of neural networks does not provide any form of quantitative information. Yes, as the reviewer states, our contribution is precisely in quantifying the approximation error! Without **regularity of graph Laplacian eigenvectors** (which has been ignored in the review) one would not be able to get any non-trivial quantitative information for the approximation (notice also our comment on memorization in Remark F.3.). Suggesting that quantitative information is minor would then suggest that working with polynomials (for which there is a universal approximation result) is as good as working with neural networks. Of course this is not the case!
> > >
> > >
> > > 2. As we stated in our previous replies, the **loss function is not convex,** thus this is not obvious. If the reviewer has a reference that takes the step that we need, we would appreciate it if they can provide this reference.
> > >
> > > 3. This is an unfortunate statement that seems to be based on a simplistic intuition, but that seems to ignore details that are worked out in the paper as well as in multiple papers in the literature that have carried out similar analyses (e.g., see Tarmoun et al (2021) and Luo and Garcia Trillos (2022), and the many other references we mention in our literature review). If for the setting that we are interested in here (PSD matrix with full rank and r less than n), the reviewer has an explicit reference that covers our case, please mention it. Keep also in mind that the relevance of this result has to be interpreted **in-context.** The review isolates our analysis of this problem from the rest of the paper, but in the paper, as well as in previous replies to reviewers, we reiterate that our main goal is to present a layered exploration of the problem of approximation of eigenvectors of large adjacency matrices (or Laplace Beltrami operators on manifolds) using neural networks. The analysis of the ambient optimization problem is motivated by our experiments, and we expect it to be important in future theoretical analysis of training dynamics in the mean-field limit regime (see 2nd paragraph in Page 4 and Section 4).

---

### Official Review · Reviewer_hkrF · 2023-11-10

**Soundness:** 3 good
**Presentation:** 3 good
**Contribution:** 3 good
**Rating:** 8
**Confidence:** 3

**Summary:**

This paper studies the objective of the "spectral neural networks (SNN)".
The objective is defined by the squared Frobenius norm of the approximation error for the kernel matrix (which is a specific graph Laplacian in the current scope) via its low-rank approximation.
By the Eckart--Young--Mirsky theorem, the ambient optimization problem guarantees that its global optimizer recovers the top-$r$ eigenbasis up to an orthogonal transformation. SNN is a neural network that is optimized by this objective, where the neural network outputs parameterize the eigenvectors.
The difficulty in the analysis of the SNN mainly comes from the fact that the ambient optimization problem (i.e., when the optimization is done in the nonparametric way without neural network parameterization) is non-convex.

The paper's contribution is threefold. First, the authors prove that there exists a MLP that can well-approximate the top-$r$ eigen-subspace, with sufficiently large number of neurons, under the manifold assumption (Theorem 2.1).
Second, it is shown that, under the same assumption, if the MLP architecture used in the optimization is sufficiently large, then the global optimizer attained by the architecture closely captures the top-$r$ eigenbasis up to a rotation (Theorem 2.2). Roughly speaking, a good MLP can be "constructed" by optimizing the SNN objective.
Lastly, the authors analyze the optimization landscape of the "ambient" problem, by examining three different regimes (Theorem 3.1-3.4).

The first two theorems constitutes an approximation theory of neural networks for the graph Laplacian matrix. Theorem 2.1 is a general approximation theory, while Theorem 2.2 is a result that specifically applies to the SNN objective.
The results in Section 3 solely cares about the ambient optimization problem being independent of a neural network parametrization, but these results are applicable for any PD matrix with a positive eigengap.

**Strengths:**

Using neural networks to parameterize eigenfunctions of an operator is a promising approach that has a great potential in many applications for large-scale, high-dimensional data.
In particular, the optimization framework of SNN is particularly appealing as an unconstrained problem, in contrast to the existing work such as SpectralNet which considers a constrained optimization problem.
Hence, understanding characteristics of the SNN optimization problem is an important problem.
The results in this paper can serve as a good initial attempt in establishing a theory in this context.

The paper is overall well thought-out, considering the level of technicalities involved in the analysis.
The results are well-motivated with illustrations in the introduction.
I think the paper provides good insights for the subject by carefully putting the recent results in spectral approximation, neural-network approximation, and Riemannian optimization, and thus worth of publication in this venue in general.

**Weaknesses:**

I believe that the manuscript is missing some works in the literature, and adding and discussing these will better guide the reader.

It is a little bit obscure what can be said beyond the graph Laplacian with MLPs under the manifold assumption, considering that there exist other important operators in different applications. For example, decomposing a Hamiltonian operator with neural networks has shown promising results in quantum chemistry, see, e.g., [A].
Also, there exists a recent paper on analyzing the "generalization error" of MLPs in solving the Schrödinger equation [B]. (It would be nice if a generalization error can be analyzed in the current paper, and if so or not, what could be challenge.)

Another line of research missing in the current paper is the recent work on generic NN-based eigensolvers [C], [D] that aim to recover the ordered top-$r$ eigenbasis (i.e., without modulo rotation) unlike the SNN and current work.

[A] Hermann, Jan, Zeno Schätzle, and Frank Noé. "Deep-neural-network solution of the electronic Schrödinger equation." Nature Chemistry 12.10 (2020): 891-897.
[B] Lu, Jianfeng, and Yulong Lu. "A priori generalization error analysis of two-layer neural networks for solving high dimensional Schrödinger eigenvalue problems." Communications of the American Mathematical Society 2.1 (2022): 1-21.
[C] Pfau, David, et al. "Spectral Inference Networks: Unifying Deep and Spectral Learning." International Conference on Learning Representations. 2018.
[D] Deng, Zhijie, Jiaxin Shi, and Jun Zhu. "Neuralef: Deconstructing kernels by deep neural networks." International Conference on Machine Learning. PMLR, 2022.

**Questions:**

- Remark 2.2 is hard to appreciate. Can you explain the reasoning behind this in detail?
- In the paragraph after (eq. 3.3), "Finally" is used twice.

---

> ### Author Response · Authors · 2023-11-15
> **Response to Reviewer hkrF**
>
> We thank the reviewer for their thoughtful comments and their positive feedback. In particular, we thank them for their insightful comments on the use of NNs to solve eigenvalue problems for more general differential operators. Indeed, we agree with the reviewer that it is of interest to provide theoretical guarantees for the use of a SNN-like approach to find the spectrum of other differential operators such as the ones in the Schrödinger equation. As the reviewer mentions, this is of relevance for the broader scientific community.
>
> We hope our edits improve our manuscript’s clarity and also better position our work in the landscape of existing results on similar problems. All references provided by the reviewer were incorporated in the paper. New text has been highlighted in blue. Below we provide answers to the reviewer’s questions.
>
>
>
> > Other differential operators and Generic NN-based eigensolvers.
>
> We have added references to the papers the reviewer mentioned. In Section A.2 we have added a brief discussion on the eigensolvers mentioned by the reviewer.
>
> >On Remark 2.2.
>
> Remark 2.2. aims at highlighting that the $\epsilon^2$ term in our error estimates in Theorem 2.1. is essential for our proof of Theorem 2.2. As discussed in our general response, this $\epsilon^2$ term could not have been obtained by directly fusing the results in Calder et al 2022 and Chen et al 2022. In order to get the term $\epsilon^2$, we needed to first obtain a stronger quantitative estimate for the regularity of the (discrete) graph Laplacian eigenvectors (stronger than the ones explicitly stated in Calder et al 2022) and then apply the results in Chen et al 2022 to find an NN approximator of a suitable extrapolation of the discrete eigenvector to the manifold (different from an associated eigenfunction!, for otherwise the error of approximation would not be $\epsilon^2$ but $\epsilon$). In order to obtain the better discrete regularity for graph Laplacian eigenvectors, we used the fact that graph Laplacian eigenvectors converge in a stronger semi-norm (almost $C^{0,1}$) toward Laplace-Beltrami eigenfunctions.

---

> > ### Comment · Reviewer_hkrF · 2023-11-23
> >
> > I appreciate the authors' response. I read all other reviews and discussion, and learned that there exist some pushback on the contribution of the paper. I still think that the paper's contribution should be judged from the entire story around understanding the theoretical properties of the neural-network-based approach for eigensubspace learning, rather than looking at each result separately. In terms of the entire story, I believe that there is a catch in this paper.
> > That said, I think some discussion on the connection from the literature such as Luo and Garcia Trillos (2022) needs be supplemented, as Reviewer JxCW pointed out. I can see that the work is being mentioned throughout in Section 3, but I think it is proper to mention and discuss this paper in Related Work section to indicate readers that while there is a very similar recent analysis in blah, the current paper has a stronger result.
> >
> > Another comment, though a bit tangential to the focus of the paper, I wish to make is that though the authors argued in one of the response that the ordered eigenbasis can be postprocessed from the Gram-Schmidt like process, in practice the idea may not work well in practice and finding a structured basis from scratch might be a better solution depending on an application at hand. In this perspective, I think it is also important to make it clear that the paper and SNN focuses on the applications such as spectral clustering that might not require ordered eigenbasis, unlike the traditional eigensolvers.
> >
> > All in all, I will keep the score, but decrease my confidence to reflect my agreement on some concerns of other reviewers.

---

### Official Review · Reviewer_JxCW · 2023-11-10

**Soundness:** 3 good
**Presentation:** 2 fair
**Contribution:** 2 fair
**Rating:** 5
**Confidence:** 3

**Summary:**

The paper is a theoretical study of Spectral Neural Networks (SNNs). The problem considered in the paper consists in efficiently approximating an adjacency matrix by a product $\mathbf Y\mathbf Y^{\mathrm T}$. The first main result (Theorem 2.1) gives a bound on the complexity of a neural network that provides an approximate solution of this problem. The second result (Theorem 2.2) shows that a solution provided by the network is close to a global minimizer, up to a rotation. The remaining results (Theorems 3.1-3.4) study the structure of the loss surface by dividing it into several regions with particular properties. The first region is a neighborhood of the optimal solution and the loss is geodesically strongly convex there (Theorems 3.1). Another region is the neighborhood of suboptimal stationary points. These points are described in Theorem 3.2, and Theorem 3.3 shows that near these points there are escape directions so that gradient flow is not trapped. Finally, Theorem 3.4 shows that in the remaining regions the gradient is large. Combined, these results suggest that the considered optimization problem can be efficiently solved by gradient descent.

**Strengths:**

**Contribution, originality, novelty.**  The paper relies very heavily on previous research of matrix factorization and SNNs such as HaoChen et al., 2021 and Luo & Garc´ıa Trillos, 2022. My impression is that the present paper does not bring any fundamentally new ideas compared to previous publications. In particular, the main message expressed in it is that the considered optimization is practically feasible and the loss landscape is benign despite the non-convexity. The same message is found in Luo & Garc´ıa Trillos, 2022 in a similar wording. Moreover, the theorems found in section 3 of the present paper are extremely similar to the theorems in Luo & Garc´ıa Trillos, 2022. The present paper indicates some differences with that earlier paper (e.g., in Remark 3.2), but they are not clearly explained and seem to be rather technical. The paper Luo & Garc´ıa Trillos, 2022 is referred to multiple times in the present paper, but is not mentioned among the related works, which is confusing.

**Writing and clarity.** On the whole, the paper is clearly written and has a big appendix containing details of its several theorems. At the same time, there are various small issues with the exposition (see below).

**Weaknesses:**

In addition to the limited novelty mentioned above, the paper suffers from some lack of clarity.

1. The beginning of the introduction sounds like the goal of the paper is to develop and analyze a neural network-based eigensolver.  My understanding of an eigensolver is that this is an algorithm that produces the full list of eigenvalues and eigenvectors. However, the method considered in the paper gives us much less: first, it is restricted to $r$ largest eigenvalues and, second, the produced matrix $\mathbf Y$ contains eigenvectors only up to an $r\times r$ rotation, so there is still work to be done to extract the eigenvectors and eigenvalues. These points are not discussed; moreover, in the comparison of SNN with traditional eigensolvers only the advantages of the SNN are mentioned.

2. *η is a decreasing, non-negative function* - where do you use that $\eta$ is decreasing?

3. *$D_G$ is the degree matrix associated to $G$* - what exactly is the definition of $D_G$?

4. In Theorem 3.2, the notation for matrices with the subscript $S$ seems unexplained. Also, the matrix $\Lambda$ is not defined (only $\overline{\Lambda}$ is defined).

**Questions:**

I would like the issue with the connection of the present paper to Luo & Garc´ıa Trillos, 2022 to be clarified.

In general, I think that the Related Work section should clearly indicate the papers that are especially closely connected to the current work, and explain the differences and the added value of the current work.

---

> ### Author Response · Authors · 2023-11-15
> **Response to Reviewer JxCW**
>
> We thank the reviewer for their questions and comments, which we address below. We are encouraged by the reviewer’s appreciation of our presentation and theoretical analysis. *We have added most of these explanatory comments and changes in the updated manuscript in blue text.*
>
> > Only recovering eigenvectors up to rotation.
>
> We do not believe this is a limitation. First, notice that for many important downstream tasks, such as spectral clustering and spectral contrastive learning, it is sufficient to generate unions of eigenspaces, and thus, recovering individual eigenvectors may not be necessary for those tasks. On the other hand, if one indeed desires to generate the first K individual eigenvectors of $A_n$, it suffices to simply run the SNN algorithm for $r=1, \ldots , K$ and carry out a Gram-Schmidt like procedure with the outcomes. This pipeline would indeed avoid the use of eigensolvers, as claimed.
>
> > Assumption on $\eta$.
>
> The non-increasing assumption on $\eta$ appears in multiple results in the manifold learning literature. We used some of these results in the proofs of our theorems. Notice that most $\eta$ used in practice (e.g., Gaussian, 0-1) satisfy this assumption.
>
> > The definition of $D_G$.
>
> $D_G$ is the diagonal matrix whose $i$-th diagonal entry is the degree of the point $x_i$. $D_G$ was defined in the Appendix in the paragraph below (A.5). In the updated manuscript, we have added an explicit reference to its definition.
>
> > Notations in Theorem 3.2.
>
> There is a typo in Theorem 3.2. We had not defined $\Lambda$. We thank the reviewer for pointing this out. We have now incorporated it in the updated manuscript.
>
> > About the related work and novelty.
>
> First, we want to point out that the work Luo & Garcia Trillos 2022 that the reviewer mentions makes very different assumptions on the matrix $A_n$ than the ones we make in our paper. The implications of the different assumptions are indicated at multiple points in the paper: in Remark 3.2, the discussion below Theorem 3.2, and Remarks H.1 and H.2.
>
> Our paper studies SNNs comprehensively and combines multiple results from several subfields (such as manifold learning, neural network approximation theory, and Riemannian optimization) in a way that is original and far from trivial. Indeed, as we explain in our general response to reviewers (see also the response to Reviewer hkrF), we had to tackle multiple technical difficulties to bring together existing results in the literature. The paper Chen et al. 2022 that the reviewer mentions is indeed a useful auxiliary result for our work, but by no means are our Theorems 2.1 and 2.2 corollaries of it.

---

> > ### Comment · Reviewer_JxCW · 2023-11-21
> >
> > I thank the authors for their response and clarifying some points that I have raised.
> >
> > However, I have to say that my main issue with the paper - its very high similarity to Luo and Garcia Trillos 2022 and a lack of acknowledgment of this fact - is not resolved by this response (or the general response to reviewers). It is obvious that in section 3 the present paper very closely follows that earlier paper. The idea of "benign landscape" and the Riemannian approach are the same; the domain is decomposed into the same subsets $\mathcal R_1,\mathcal R_2,\mathcal R_3',\mathcal R_3'',\mathcal R_3'''$; theorems 3.1, 3.3, 3.4 describing the loss landscape in particular subsets are very similar to the respective theorems 3, 4, 5 in Luo and Garcia Trillos 2022. The similarities are much more pronounced than the differences. But (in contrast to the differences) the similarities are not acknowledged in the paper. The Related work section mentions all kinds of works, some only marginally relevant, but does not mention Luo and Garcia Trillos 2022.

---

> ### Author Response · Authors · 2023-11-21
> **Thank you for your comments**
>
> Thank you for reviewing our manuscript and providing your feedback. We have now incorporated a brief discussion on the global optimization landscape from Luo & Garcia Trillos 2022 into the related work section of our paper. Please see our revised manuscript, where this is highlighted in blue.
>
> In response to your concerns we would like to highlight the following:
>
> 1. The reviewer’s main concern is that our paper is very much like Luo and García Trillos, yet the reviewer has focused entirely on one single aspect of our paper (more on this below) and did not discuss more than half of our paper, which has little to no overlap with the mentioned paper. In particular, there is no mention of the motivation for our problem, nor to the approximation theory results Theorems 2.1 and 2.2, nor to the connection with learning solutions of PDEs, nor to the prospect of using our ideas to provide a theoretical basis for the use of SNNs in other problems in the sciences (as other reviewers have highlighted). In fact, there is not even a mention to our numerical simulations, which try to convey why studying the landscape of the ambient problem may be relevant for the SNN training problem, which is what motivated some of the open problems we formulated in the paper.
>
> 2. We do mention the extent of the similarities between our section 3 and the paper of Luo and Garcia Trillos. Please see our Remark 3.1 in the main text and the multiple remarks and comments in section H.
>
> 3. The concepts of benign landscape and the Riemannian approach are very broad ideas in the optimization literature, not just ours or Luo & Garcia Trillos 2022’s. A quick online search reveals this.
>
> 4. R1, R2, R3 are indeed similarly defined as in Luo & Garcia Trillos, but, as explained extensively in the paper and in our previous replies to all reviewers,  the challenge is in showing that those regions indeed satisfy the desired properties to describe the landscape of our ambient problem as benign: local strong convexity around global minimizer, escape direction for saddles, large gradient in all other regions. In other words, R1, R2, R3 are ansatz, which are naturally inspired by previous work, but this doesn’t imply the ansatz is trivially correct in our case. There is an enormous difference between 1) guessing what a good answer could be and 2) proving that this guess is indeed a good answer. We thus invite the reviewer to compare the papers in more detail, and not focus exclusively on their superficial similarities. On this last point, we reiterate our discussion in 1. above.

---

> > ### Comment · Reviewer_JxCW · 2023-11-22
> >
> > I thank the authors for their response.
> >
> > 1. I keep my opinion that the paper is misleading in its novelty claims. The paper states as one of its main contributions: "*we begin an exploration of the optimization landscape of SNN and in particular provide a full description of SNN’s associated ambient space optimization landscape*". I don't see how this can be true, since a significant part of this work has obviously been already done in a different publication. I agree that the paper strengthens a previous analogous optimization landscape result. However, this is far from what the paper claims. I believe that the proper way to handle that would be to explain this previous result in the introduction along with the challenges associated with its improvement, and to claim the improvement.
> > 2. Regarding the other results of the paper, I agree that it contains new contributions in the form of motivating experiments and the approximation results in section 2. At the same time, I'm not convinced in the importance of these results. My understanding is that Theorem 2.1 is obtained by applying existing general statistical and approximation frameworks to an existing regularity property for the eigenvectors. There is definitely a new technical contribution here. But conceptually the overall approach seems standard, the result is not surprising, and the importance of the specific bounds established in Theorem 2.1 is not clear to me. As for Theorem 2.2, technical aspects aside, I see it as basically saying that under the spectral gap assumption the approximate minimizer of the ambient functional (1.5) is close to an actual minimizer up to a rotation, which is a standard result from linear algebra. There is definitely a new technical contribution in establishing relevant specific bounds for the network size, probability, approximation error etc. as appear in Theorem 2.2. But again, why these particular bounds are important is not very clear to me.
> >
> > Nevertheless, I admit that I have not studied the proofs and may have missed something, so following this discussion I'm increasing my score.

---

> > > ### Author Response · Authors · 2023-11-22
> > >
> > > 1. The text quoted by the reviewer is actually true and correct, since, as explained multiple times in the paper and in previous replies, we do initiate the exploration of the SNN problem (i.e., the NN parameterized problem,  Luo and García Trillos do not talk about NN at all!), and a first key step is to characterize our ambient problem. In turn, for the reasons we present in the general reply to reviewers and in the paper, the setting of our ambient problem is more involved than the specific instance studied in Luo and Garcia Trillos (no saddles in L & GT, multiple saddles in our case). In summary, no, we don't agree with the reviewer's assessment of our statements at any level.
> > >
> > > 2. It is unclear what the reviewer means by "which is a standard result from linear algebra". What is a result from linear algebra?  Theorem 2.2. provides quantitative information about how one can approximate the space generated by the top eigenvectors of the adjacency matrix with the minimizer, **in the space of neural networks**, of the loss function studied in the paper. By no means this is direct application of a linear algebra result.

---

> > > > ### Comment · Reviewer_JxCW · 2023-11-22
> > > >
> > > > I thank the authors for their response.
> > > > 1. "*we do initiate the exploration of the SNN problem (i.e., the NN parameterized problem, Luo and García Trillos do not talk about NN at all!)*". \
> > > > Again, I do not understand this claim. Section 3 on the loss landscape says nothing about neural networks. It only considers the optimization problem (1.5) in the ambient space, analogous to the problem in an earlier paper.
> > > > 2. "*What is a result from linear algebra?*" \
> > > > I referred to the uniqueness part of the Eckart–Young–Mirsky theorem, which implies that, under the gap assumption, if $l(\mathbf Y)$ is close to the minimal value of the functional $l$, then $\min_{\mathbf O\in\mathbb O_n}\\|\mathbf Y-\mathbf Y^*\mathbf O\\|_F$ is close to 0. As I wrote, I agree that specific quantitative information regarding the approximation (in particular, in the space of neural networks) found in Theorem 2.2 is a new technical contribution of the paper.
> > > >
> > > > I think that it is by now clear that the authors and I have different opinions on some novelty aspects of the paper, so I thank the authors again and conclude this discussion.

---

> > > > > ### Author Response · Authors · 2023-11-22
> > > > > **Thanks again!**
> > > > >
> > > > > We thank the reviewer for engaging in this discussion. We certainly respect their opinions, although we disagree with part of them.
> > > > >
> > > > > For the record, and for the benefit of a future reader of this conversation, we want to highlight that the discussion in section 3 is by no means isolated from the rest of the paper and its motivation can be found in paragraphs 2-4 in page 4 in the paper as well as in the last paragraph in the conclusions section in page 9. See also our last paragraph in the general response to reviewers.
> > > > >
> > > > > Thanks again.

---

### Author Response · Authors · 2023-11-15
**General response to all reviewers**

We thank all reviewers for their comments and questions about our submission.

We start by highlighting the high level aims and outcomes of our paper. Our paper combines, in non-trivial and novel ways, a variety of recent results in subfields such as manifold learning, approximation theory of neural networks, Riemannian optimization, and matrix factorization problems. An important outcome of our work is concrete theoretical backing for using a neural network based approach for learning relevant (spectral) geometric structure from data. This geometric structure can be used in machine learning in multiple downstream applications such as clustering, dimensionality reduction, and contrastive learning. We are confident that our theoretical contributions are novel, non-trivial, and sound. They are also the starting point of a systematic theoretical exploration of the use of NNs for other eigenvalue problems of differential operators on manifolds, which is an important research inquiry for machine learning and the sciences, as also alluded to by some of the reviewers.

Moving on to more detailed points, a common theme among some reviews is the misunderstanding of the merits of our main theorems and their relation to previous results in the literature. We want to make the following clarifications in response to these concerns.

**About our approximation theory results:** We want to highlight that direct fusion of the results in Chen et al. 2022 and Calder et al. 2022 cannot yield Theorem 2.1, contrary to what was alluded to by some reviewers. Indeed, a key highlight of the estimates in Theorem 2.1, which we comment on in Remak 2.2, is that they depend quadratically on the graph connectivity $\epsilon$. If we had used the results in Chen et al. 2022 directly to produce a neural network approximating a manifold eigenfunction and then invoked the manifold learning results from Calder et al. 2022 to say that this neural network must also approximate graph Laplacian eigenvectors, then the error estimates that would take the place of those in Theorem 2.1. would have depended linearly in $\epsilon$. Why would this be an issue for our purposes? Because, as discussed in Remarks 2.2. (in the main text), and in G1, G2, and G3 (in the appendix), with a mere linear dependence on $\epsilon$, we could not have guaranteed that the neural network in Theorem 2.1 has an objective function value smaller than the values of other critical points for the ambient space problem (e.g., matrices $Y$ generated by other eigenvectors of $A_n$ that are not the desired top $r$). In turn, we could not have guaranteed the approximation of the space generated by the top $r$ eigenvectors. Without our version of Theorem 2.1, we would not have been able to prove Theorem 2.2. We highlight that Theorem 2.2. provides concrete theoretical backing for the use of SNNs in practice.

While Theorem 2.2 does indeed use Theorem 2.1, the former is not a trivial consequence of the latter because the loss function $\ell$ is **non-convex**. In particular, controlling the energy gap does not immediately provide a **quantitative** bound for the distance to global minimizers of $\ell$. This is quite different from the strongly convex case (e.g., quadratic loss in supervised settings), where the analog of Theorem 2.2. would indeed follow immediately from a result like Theorem 2.1. For our problem, the relation between the energy gap and the distance to optimizers follows from the careful analysis presented in Section G in the Appendix.

**About our optimization results**: In our work, we assume $A_n$ is full rank and the number of eigenvectors $r$ to be recovered is strictly smaller than $n$. In the work of Luo and Garcia Trillos 2022, the matrix $A_n$ is assumed to be of rank $r$, i.e., it is equal to the number of eigenvectors to be recovered. This difference in setup creates the following non-trivial challenge for us: there are multiple saddle points in our ambient matrix factorization problem, while in the problem in Luo and Garcia Trillos 2022, all critical points are global optimizers in the full rank matrix manifold $R^{n \times r}_{*}$. This point is emphasized **in Remark 3.1, the discussion below Theorem 3.2, Remarks H.1, H.2, and the discussion below H.3**. In summary, we cannot deduce our results from those in Luo and Garcia Trillos 2022, and we need to consider new constructions and computations.

To conclude, Q1-Q3 are logically interconnected and describe a layered exploration of the problem of approximation of eigenvectors of large adjacency matrices using neural networks. Thus, the paper has to be appreciated as a whole, and we encourage the reviewers to look at the entirety of our results to understand our theoretical contributions better. We thank the reviewers again for their careful reading, comments, and suggestions.

---

### Meta-Review · Area_Chair_yaxS · 2023-12-10

**Metareview:**

The reviewers were split about this paper and did not come to a consensus: on one hand they appreciated the overall motivation of the work and some of the technical contributions, their main concerns were with novelty and clarity. After going through the paper and the discussion I have decided to vote to reject based on the above issues. Specifically, the reviewers argued that Theorems 3.1, 3.3, 3.4 in the paper are highly similar to Theorems 3, 4, 5 in Luo and Garcia Trillos, 2022. The authors responded that the reviewer only highlighted one aspect of the paper, and failed to mention any other parts of the paper, which they argue are novel. The authors also say that they do make this comparison in Remark 3.1 and Appendix H. The reviewers responded by arguing that this comparison is not clear from the paper when it states as one of its main contributions that "we begin an exploration of the optimization landscape of SNN and in particular provide a full description of SNN’s associated ambient space optimization landscape", something which has slaredy been done in Luo and Garcia Trillos, 2022. The reviewers further argue that they are not convinced by the importance of the other parts of the paper. They agree there is new technical work, but the utilities of the different  results are either not obvious, or standard. The authors argue that the claim is true because Luo and García Trillos, 2022 do not discuss NNs at all. The reviewer points out that there is no mention of NNs either in Section 3 of the paper. Ultimately the difference seems to be that here A_n is full rank and the number of eigenvectors r to be recovered is strictly smaller than n, whereas in Luo and Garcia Trillos, 2022 A_n is exactly rank r. This work solves a harder issue as there are saddle points. Ultimately, the authors were able to win the argument that there is a technical difference between the results, but in their intensity to make this point they completely dropped the argument about the importance of the result. This point is not fully resolved. However, the bigger point this discussion made, which was also made by multiple reviewers, is that the technical results are not clearly described, and they are not placed into context with respect to prior work. Indeed similar points were made about Theorems 2.1 and 2.2. Constants that are used throughout the paper are not clearly defined and there are not enough intuitions, explanations, or implications provided to make the results easy to understand (as illustrated by the long back and forth above). For these reasons I vote to reject. The reviewers have given extremely detailed feedback and I recommend the authors follow / respond to their comments closely before submitting to a future ML venue. If the authors are able to fix these things it will make a much stronger submission.

**Justification For Why Not Higher Score:**

Authors did a poor job during the dicussion responding directly to the reviewer concerns, sometimes quite rudely. The valid confusions of the reviewers were not resolved.

**Justification For Why Not Lower Score:**

N/A

---

### Decision · Program_Chairs · 2024-01-16

Reject